# The O-glycosyltransferase C1GALT1 promotes *EWSR1::FLI1* expression and is a therapeutic target for Ewing sarcoma

Shahid Banday [1] ✉, Alok K. Mishra [1], Romana Rashid[2], Tianyi Ye [1], Amjad Ali[1], Junhui Li [1], Jason T. Yustein [3], Michelle A. Kelliher [1], Lihua Julie Zhu [1,4], Sara K. Deibler[1], Sunil K. Malonia [1] ✉ & Michael R. Green [1]

Ewing sarcoma (ES) is an aggressive bone cancer driven by the oncogenic fusion-protein EWSR1::FLI1, which is not present in normal cells and is therefore an attractive therapeutic target. However, as a transcription factor, EWSR1::FLI1 is considered undruggable. Factors that promote *EWSR1::FLI1* expression, and thus whose inhibition would reduce EWSR1::FLI1 protein levels and function, are potential drug targets. Here, using genome-scale CRISPR/Cas9 knockout screening, we identify C1GALT1, a galactosyltransferase required for the biosynthesis of many O-glycoproteins, as a factor that promotes *EWSR1::FLI1* expression. We show that C1GALT1 acts by O-glycosylating the pivotal Hedgehog (Hh) signaling component Smoothened (SMO), thereby stabilizing SMO and stimulating the Hh pathway, which we find directly activates *EWSR1::FLI1* transcription. Itraconazole, an FDA-approved anti-fungal agent that is known to inhibit C1GALT1, reduces EWSR1::FLI1 levels in ES cell lines and suppresses growth of ES xenografts in mice. Our study reveals a therapeutically targetable mechanism that promotes *EWSR1::FLI1* expression and ES tumor growth.

Ewing sarcoma (ES) is an aggressive cancer that begins in the bone or soft tissue (such as cartilage), and occurs predominantly in children and young adults[1]. Current treatment of ES involves an intensive chemotherapy regimen followed by radiotherapy and/or surgery. Although outcomes for ES patients with localized disease have improved in recent decades, progress in treating metastatic or recurrent disease has been limited, with 5-year survival rates remaining at a low 20–30%[2]. Notably, there are no approved targeted therapies for ES.

The characteristic feature of ES is a chromosomal translocation that fuses a transactivation domain of the RNA-binding protein EWSR1 with the DNA-binding domain of an ETS proto-oncoprotein, most commonly

FLI1[1]. The resultant EWSR1::FLI1 is an oncogenic fusion-protein that can act as either a transcriptional activator or repressor[3]. Numerous EWSR1::FLI1 target genes have been identified, and it is generally believed that the aberrant transcriptional regulation of these target genes by EWSR1::FLI1 drives ES progression[3]. EWSR1::FLI1 is essential for ES cell survival and tumor growth in mice[4,5]. Given its absence in normal cells, EWSR1::FLI1 is a promising therapeutic target. However, EWSR1::FLI1 lacks enzymatic activity and is considered undruggable by conventional small molecule inhibitors[6]. Therefore, attempts to directly inhibit EWSR1::FLI1 have been largely unsuccessful[7]. As an alternative approach, we sought to identify druggable factors that promote *EWSR1::FLI1* expression; inhibition of such factors would reduce

[1]Department of Molecular, Cell and Cancer Biology, University of Massachusetts Chan Medical School, Worcester, MA 01605, USA. [2]Department of Medicine, University of Massachusetts Chan Medical School, Worcester, MA 01605, USA. [3]Winship Cancer Institute and Aflac Cancer and Blood Disorders Center, Emory University, Atlanta, GA 30322, USA. [4]Program in Molecular Medicine and Department of Genomics and Computational Biology, University of Massachusetts Chan Medical School, Worcester, MA 01605, USA. ✉e-mail: shahidkhursheed.banday@umassmed.edu; sunil.malonia@umassmed.edu

EWSR1::FLI1 levels and thus function and may reveal therapeutically targetable mechanisms for the treatment of ES.

In this study, we describe a fluorescent reporter-based genome-scale CRISPR/Cas9 knockout screening strategy to discover cellular factors that promote the expression of the endogenous *EWSR1::FLI1* gene in ES cells. Using this approach, we identify C1GALT1, an enzyme required for the O-glycosylation of many proteins[8], as a regulator of *EWSR1::FLI1* expression. C1GALT1 is highly expressed in ES, and mechanistically we find that it acts by O-glycosylating and promoting the stabilization of Smoothened (SMO), a key transducer of the Hedgehog (Hh) signaling pathway, which we show is required for *EWSR1::FLI1* expression. Pharmacological inhibition of C1GALT1 using the FDA-approved anti-fungal drug itraconazole decreases *EWSR1::FLI1* expression in human ES cell lines resulting in cell death, and suppresses ES tumor growth in xenograft mouse models. Our findings have therapeutic implications for treating ES.

## Results

### A genome-scale CRISPR/Cas9 screen identifies factors that promote *EWSR1::FLI1* expression

To identify factors that promote *EWSR1::FLI1* expression, we performed a genome-scale CRISPR/Cas9-based screen for genes whose knockout reduces EWSR1::FLI1 protein levels. To facilitate high-throughput screening, we used CRISPR-mediated genome editing to construct a reporter cell line in which the endogenous *EWSR1::FLI1* gene was fused to the fluorescent reporter tdTomato (Fig. 1a and Supplementary Fig. 1a–f). The reporter cell line was derived from the human ES cell line A673, which requires EWSR1::FLI1 to form tumors in mice but not for viability and proliferation in culture and therefore, unlike most other ES cell lines, is not killed following loss of EWSR1::FLI1[9]. The reporter cell line also stably expressed Cas9 and a control fluorophore, enhanced green fluorescence protein (EGFP), enabling us to exclude factors whose knockout leads to a general

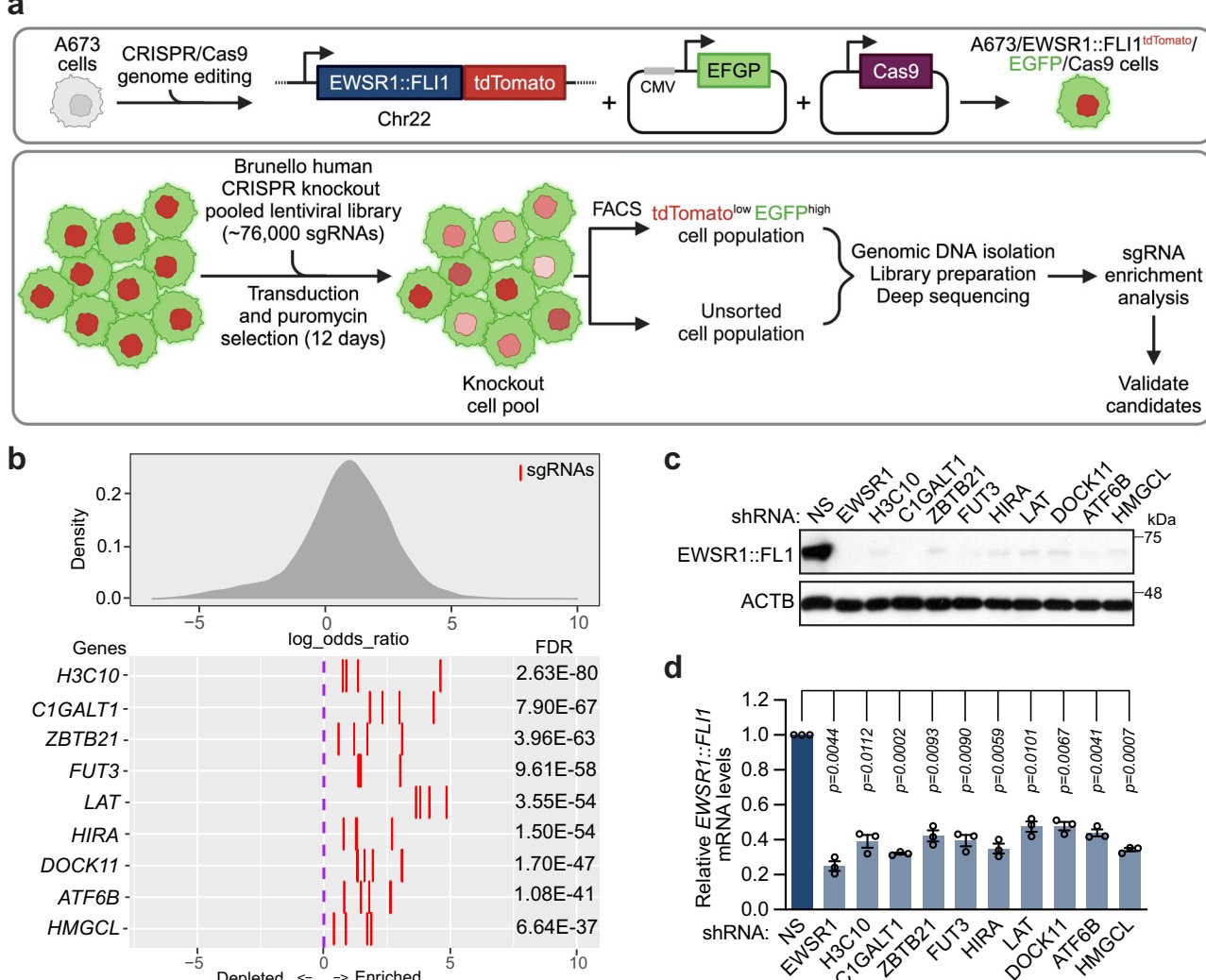

**Fig. 1 | A genome-scale CRISPR/Cas9 screen identifies factors that promote *EWSR1::FLI1* expression. a** Schematic of the reporter cell line construction (top) and genome-scale CRISPR/Cas9-based screening strategy (bottom). Created in BioRender.com[68]. **b** False-discovery rate (FDR) plot showing the distribution of sgRNAs (shown as red lines) targeting candidate genes enriched in the tdTomato^low EGFP^high population. *P*-values were calculated using a two-tailed Fisher's Exact test with Benjamini-Hochberg adjustment to control for FDR in the context of multiple comparisons. **c** Representative immunoblot showing EWSR1::FLI1 protein levels (monitored using an anti-FLI1 antibody) in A673 cells stably expressing an shRNA

targeting each of the nine validated candidates, or as control a NS or EWSR1 shRNA. β-actin (ACTB) was monitored as a loading control. **d** qRT-PCR analysis monitoring relative *EWSR1::FLI1* mRNA levels following knockdown of each of the nine candidates. *EWSR1::FLI1* mRNA was detected using a primer pair spanning the fusion junction and is shown relative to that obtained with an NS shRNA, which was set to 1. Data are presented as mean ± SEM (*n* = 3 biologically independent experiments). *P*-values were calculated using one-way ANOVA with post-hoc Dunnett's multiple comparisons test. Source data are provided as a Source Data file.

reduction in protein levels or cell survival, which would result in decreased EGFP signal. Thus, cells harboring a gene knockout that selectively reduced levels of EWSR1::FLI1 would be identified as tdTomato[low] EGFP[high].

For the screen, A673/EWSR1::FLI1[tdTomato]/EGFP/Cas9 reporter cells were transduced with the Brunello human CRISPR knockout pooled lentiviral library, which consists of ~76,000 sgRNAs targeting ~19,000 genes (~4 sgRNAs per gene)[10]. Cells were puromycin selected for 12 days, tdTomato[low] EGFP[high] cells were isolated by fluorescence-activated cell sorting (Supplementary Fig. 1g) and expanded, and sgRNAs that were enriched relative to the unsorted population were identified by deep sequencing (Fig. 1a and Supplementary Data 1). We identified ~70 genes for which four sgRNAs were significantly enriched in the tdTomato[low] EGFP[high] population (Supplementary Data 2). Based on *P*-value, we selected the highest ranking 20 candidates for validation experiments, in which we knocked down each gene in parental A673 cells using two independent shRNAs and monitored endogenous EWSR1::FLI1 protein levels by immunoblotting. As a positive control, we also knocked down *EWSR1*, which encodes the N-terminal portion of the EWSR1::FLI1 fusion-protein. We considered a candidate validated if both shRNAs (1) decreased EWSR1::FLI1 protein levels and (2) decreased mRNA levels of the target gene compared to that obtained with a control non-silencing (NS) shRNA. This approach enabled us to identify nine factors that promote EWSR1::FLI1 expression (Fig. 1b, c;

Supplementary Fig. 2a, b and Supplementary Table 1). Quantitative reverse transcription PCR (qRT-PCR) analysis revealed that knock-down of each of these factors also reduced *EWSR1::FLI1* mRNA levels (Fig. 1d and Supplementary Fig. 2c). In addition, the primary screen identified several other candidates of interest that had at least two significantly enriched sgRNAs, including: USP14 and USP19, two deubiquitinating enzymes with a known role in EWSR1::FLI1 protein stability[11,12]; HNRNPH1 and SF3B1, two factors previously shown to be required for proper splicing and expression of *EWSR1::FLI1*[13]; and several components of the Hh signaling pathway (discussed further below) (Supplementary Table 1).

## The O-galactosyltransferase C1GALT1 is highly expressed in ES cells and its genetic or pharmacological inhibition reduces EWSR1::FLI1 levels and function

Of the nine validated top-ranking factors, the one we found of particular interest was C1GALT1 (core 1 synthase, glycoprotein-N-acetylgalactosamine 3-β-galactosyltransferase 1), an enzyme that catalyzes a critical step in the O-glycosylation of many proteins[8]. To further validate C1GALT1 as a factor required for *EWSR1::FLI1* expression, we performed several additional experiments. CRISPR/Cas9-mediated knockout of C1GALT1 in A673 cells abolished EWSR1::FLI1 levels (Fig. 2a). Moreover, shRNA-mediated knockdown of the obligate chaperone of C1GALT1, C1GALT1C1 (also known as COSMC)[14], which was

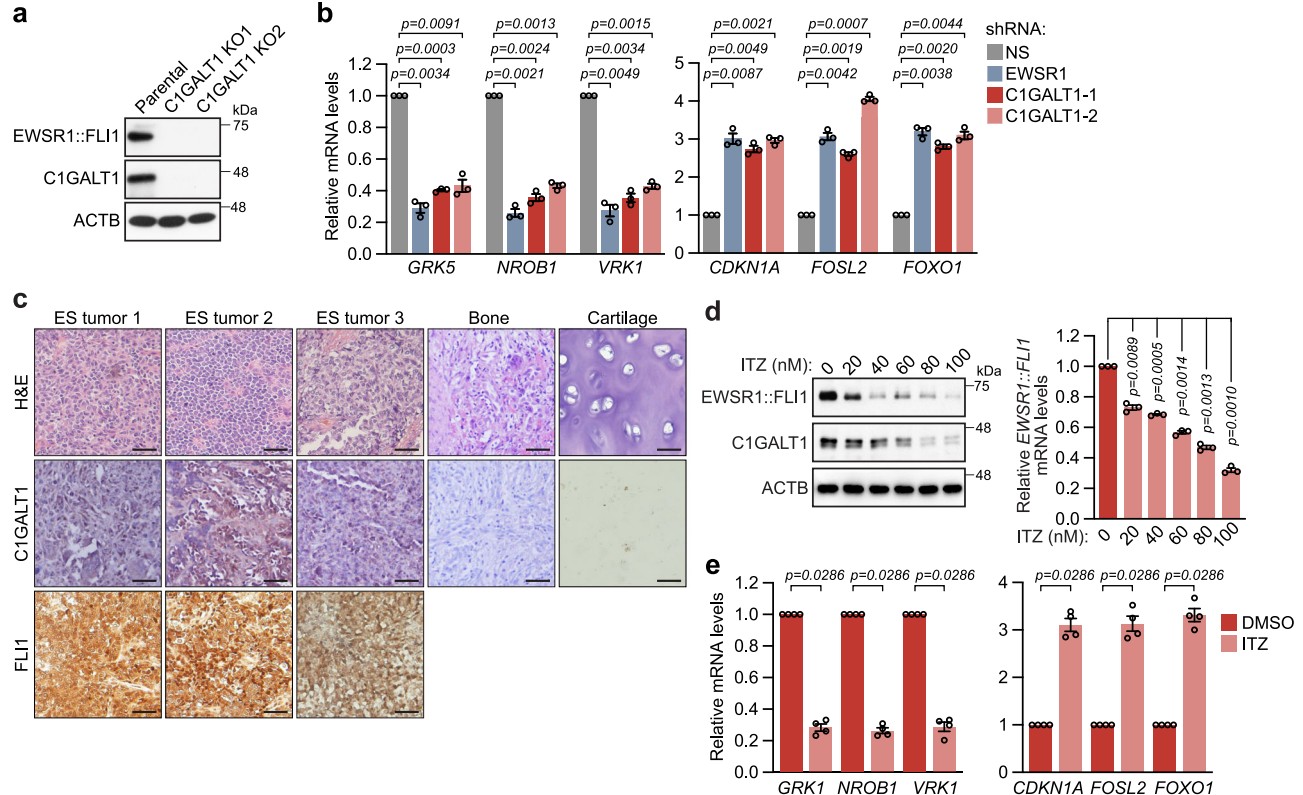

**Fig. 2 | The O-galactosyltransferase C1GALT1 is highly expressed in ES cells, and its genetic or pharmacological inhibition reduces EWSR1::FLI1 levels and function. a** Representative immunoblot showing EWSR1::FLI1 levels (monitored using an anti-FLI1 antibody) in two independent C1GALT1 knockout (KO) A673 clones. **b** qRT-PCR analysis monitoring relative mRNA levels of EWSR1::FLI1 target genes in A673 cells stably expressing an NS, EWSR1, or C1GALT1 shRNA. Data are presented as mean ± SEM (*n* = 3 biologically independent experiments). *P*-values were calculated using one-way ANOVA with post-hoc Dunnett's multiple comparisons test. **c** Representative IHC images showing H&E and C1GALT1 staining in *n* = 3 independent ES patient tumor samples, as well as normal bone and cartilage tissues. FLI1 staining is shown in the ES tumor samples. Scale bars, 50 μm. **d** (Left)

Representative immunoblot showing levels of EWSR1::FLI1 (monitored using an anti-FLI1 antibody) and C1GALT1 in A673 cells treated with DMSO or itraconazole (ITZ) for 72 h. (Right) qRT-PCR analysis monitoring relative *EWSR1::FLI1* mRNA levels in A673 cells treated with DMSO or ITZ for 72 h. The results were normalized to that obtained with DMSO, which was set to 1. Data are presented as mean ± SEM (*n* = 3 biologically independent experiments). *P*-values were calculated using one-way ANOVA with post-hoc Dunnett's multiple comparisons test. **e** qRT-PCR analysis monitoring relative mRNA levels of EWSR1::FLI1 target genes in A673 cells following treatment with DMSO or ITZ (100 nM for 72 h). Data are presented as mean ± SEM (*n* = 4 biologically independent experiments). *P*-values were calculated using a two-tailed Mann–Whitney test. Source data are provided as a Source Data file.

identified in the primary screen (see Supplementary Table 1), also reduced EWSR1::FLI1 levels (Supplementary Fig. 3a, b). Finally, shRNA-mediated knockdown of C1GALT1 reduced expression of genes that are normally positively regulated by EWSR1::FLI1, and increased expression of genes that are normally negatively regulated by EWSR1::FLI1 (Fig. 2b).

C1GALT1 overexpression has been documented in various malignancies, including pancreatic, esophageal, gastric, colorectal, ovarian, prostate, liver, lung, and head and neck cancers, and shown to contribute to the malignant phenotype (e.g., cell migration, proliferation, and metastasis) and chemotherapeutic drug resistance[15–25]. We therefore sought to determine if C1GALT1 was also overexpressed in ES. Immunohistochemistry (IHC) staining revealed high C1GALT1 expression in ES tumors from three primary patient samples, whereas C1GALT1 expression was undetectable in normal bone and cartilage tissue (Fig. 2c and Supplementary Fig. 3c). Consistent with these results, C1GALT1 expression was also elevated in a panel of human ES cell lines (Supplementary Fig. 3d). Notably, Kaplan−Meier analysis of published datasets[26] indicated that high C1GALT1 levels in ES are associated with decreased overall survival (Supplementary Fig. 3e), consistent with observations in other cancers[16,21,23–25].

Previous studies have shown that C1GALT1 is pharmacologically inhibited by itraconazole (ITZ)[16], an FDA-approved anti-fungal agent that acts by inhibiting the fungal cytochrome P450-dependent enzyme lanosterol 14-α demethylase (CYP51A1), which is required for the synthesis of ergosterol, a vital component of fungal cell membranes[27]. ITZ has been shown to inhibit C1GALT1 by directly binding C1GALT1 and promoting its proteasomal degradation[16]. We, therefore, sought to determine whether ITZ, like C1GALT1 knockdown, would reduce EWSR1::FLI1 levels and function. We found that treatment of A673 cells with ITZ led to the dose-dependent reduction of EWSR1::FLI1 mRNA and protein levels (Fig. 2d). Moreover, consistent with the results of Fig. 2b, treatment of A673 cells with ITZ reduced expression of genes that are normally positively regulated by EWSR1::FLI1 and increased expression of genes that are normally negatively regulated by EWSR1::FLI1 (Fig. 2e). Notably, expression of human CYP51A1 or the cytochrome P450 isoenzyme CYP3A4, which is also potently inhibited by ITZ[28], was undetectable in A673 and other ES cell lines (Supplementary Fig. 3f), and treating A673 cells with another FDA-approved anti-fungal CYP51A1 inhibitor, ketoconazole, which does not inhibit C1GALT1[16], had no effect on EWSR1::FLI1 expression (Supplementary Fig. 3g), ruling out the possibility that ITZ reduced EWSR1::FLI1 levels via CYP51A1/CYP3A4 inhibition. Collectively, these results demonstrate that C1GALT1 is highly expressed in ES cells and that its genetic or pharmacological inhibition leads to reduced EWSR1::FLI1 levels and function, suggesting it is a promising therapeutic target in ES.

## EWSR1::FLI1 expression is promoted by Hh signaling

ITZ has been shown to have various anti-cancer activities and, in particular, to inhibit the Hh signaling pathway[29,30], an evolutionarily conserved pathway that plays an essential role in normal embryonic development as well as cancer development and progression[31]. In vertebrates, Hh signal transduction occurs in an organelle called the primary cilia. Canonical activation of the Hh pathway occurs when a Hh ligand binds to the transmembrane receptor PTCH1, thereby relieving its inhibition of SMO, a seven-pass transmembrane G-protein coupled receptor that is the key signal transducer of the Hh pathway. Upon activation, SMO translocates to the primary cilia and initiates a signaling cascade that promotes the release of key effectors of the Hh pathway−the zinc-finger transcription factors GLI1 and GLI2−from their sequestration in the cytoplasm by SUFU, a major negative regulator of Hh signaling. Translocation of GLI1 and GLI2 to the nucleus results in transcriptional activation of Hh-dependent target genes,

which include PTCH1 and GLI1, thereby establishing a series of positive and negative feedback loops. The mechanism by which ITZ inhibits Hh signaling is not well understood. Aberrant activation of Hh signaling is associated with many cancers and can be due to mutations that activate the signaling pathway (such as PTCH1 loss-of-function or a SMO gain-of-function mutations) or overexpression of pathway components[31]. In sarcomas, including ES, aberrant Hh signaling is thought to be due to increased expression of GLI1, SMO, and PTCH1[32]. In particular, GLI1 is a known transcriptional target of EWSR1::FLI1[33]. IHC analysis of ES patient tumor samples confirmed high levels of SMO and GLI1 (Fig. 3a), which correlated with high C1GALT1 expression (see Fig. 2c). Furthermore, high SMO levels were observed in human ES cell lines (see Supplementary Fig. 3d).

Notably, our CRISPR/Cas9 screen identified multiple Hh signaling components, including SMO and GLI1 (see Supplementary Table 1). We, therefore, sought to determine whether Hh signaling promotes EWSR1::FLI1 expression. In A673 cells, knockdown of SMO, GLI1, or GLI2 using two independent siRNAs resulted in reduced EWSR1::FLI1 mRNA and protein levels (Fig. 3b–d). Similarly, treatment of A673 cells with a pharmacological inhibitor of SMO (cyclopamine) or GLI-mediated transcription (GANT-61) led to the dose-dependent reduction of EWSR1::FLI1 mRNA and protein levels (Fig. 3e, f). Conversely, treatment of A673 cells with the Hh activator SAG, a pharmacological agonist of SMO, led to the dose-dependent increase of EWSR1::FLI1 at both the mRNA and protein levels (Fig. 3g).

As mentioned above, Hh signaling culminates in the activation of the transcription factors GLI1 and GLI2, which bind to and activate the expression of target genes. We therefore asked whether GLI1 and GLI2 were bound to the EWSR1 promoter, which drives expression of EWSR1::FLI1. Previous studies have shown that GLI transcription factors bind with high affinity to the consensus sequence GACCACCCA[34], of which the last five residues are the most critical for binding[35]. Bioinformatic analysis identified four candidate GLI-binding sites in EWSR1, three upstream of the transcription start-site (TSS) and one downstream in the second intron (Fig. 3h). Directed chromatin immunoprecipitation (ChIP) experiments in A673 cells showed that GLI1 and GLI2 are bound to all four of the predicted GLI-binding sites in EWSR1 (Fig. 3i). CRISPR/Cas9-mediated deletion of two GLI-binding sites located in the promoter region upstream of the TSS (P2 and P3) resulted in decreased EWSR1::FLI1 expression at the mRNA and protein levels (Fig. 3j). Collectively, these results indicate that in ES cells, Hh signaling directly promotes EWSR1::FLI1 expression. Combined with previous findings showing EWSR1::FLI1 regulates Hh signaling[33,36], our results suggest a feedback loop in which Hh signaling promotes EWSR1::FLI1 expression, which in turn increases Hh signaling.

## C1GALT1 stimulates Hh signaling in ES cells

The results described above, in conjunction with those from previous studies, led us to hypothesize that C1GALT1 promotes EWSR1::FLI1 expression by stimulating Hh signaling. In A673 cells, shRNA-mediated knockdown of C1GALT1 (Fig. 4a) or pharmacological inhibition with ITZ (Fig. 4b and Supplementary Fig. 4a) led to reduced mRNA and protein levels of SMO and GLI1, a common readout of Hh signaling, consistent with previous studies showing ITZ inhibits Hh signaling[37]. To independently confirm this finding, we assessed the effects of C1GALT1 knockdown in a well-characterized reporter assay for monitoring the activity of the Hh signaling pathway, in which luciferase expression is driven by multiple GLI-binding sites. We found that shRNA-mediated knockdown of C1GALT1, or as a control SMO, reduced GLI-driven luciferase expression in A673 cells (Fig. 4c). Consistent with this result, C1GALT1 knockdown (Fig. 4d) or ITZ treatment (Fig. 4e) reduced binding of GLI1 and GLI2 to the endogenous EWSR1 gene in A673 cells.

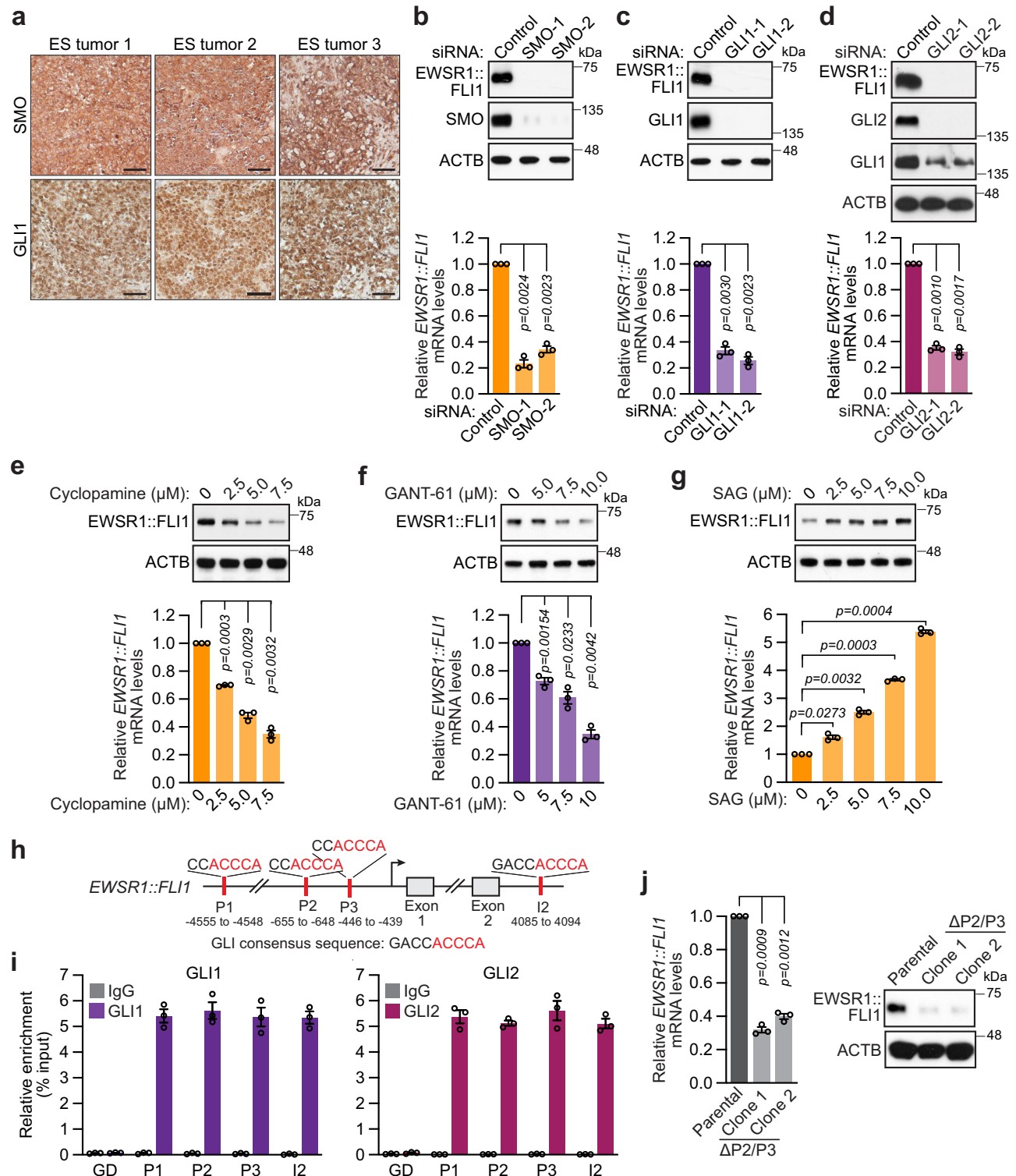

Conversely, we asked whether ectopic expression of C1GALT1 could promote Hh signaling. These experiments were performed in NIH 3T3 cells, which exhibit a low, basal level of Hh signaling[38] and are a commonly used experimental model for studying Hh signaling regulation. Similar to treatment with the Hh activator SAG, ectopic expression of C1GALT1 (Supplementary Fig. 4b) resulted in increased levels of SMO and GLI1 (Fig. 4f), indicative of increased Hh signaling. Ectopic expression of C1GALT1 also increased GLI-driven luciferase expression in NIH 3T3 cells, similar to ectopic expression of SMO (Fig. 4g and Supplementary Fig. 4c).

Collectively, these findings suggest that high levels of C1GALT1 stimulate Hh signaling.

## C1GALT1 promotes O-glycosylation and stabilization of SMO in ES cells

We next sought to determine the mechanistic basis by which C1GALT1 stimulates Hh signaling. C1GALT1 plays a critical role in the glycosylation of a class of O-glycoproteins that are post-translationally modified by the addition of N-acetylgalactosamine (GalNAc)-linked glycans[8]. O-GalNAc glycans vary in length from a

**Fig. 3 | *EWSR1::FLI1* expression is promoted by Hh signaling. a** Representative IHC images showing SMO and GLI1 staining in $n = 3$ independent ES patient tumor samples. Scale bars, 50 µm. **b–d** Representative immunoblot (top) and qRT-PCR (bottom) analysis monitoring EWSR1::FLI1 protein (using an anti-FLI1 antibody) and relative mRNA levels, respectively, in A673 cells expressing a control siRNA or one of two independent SMO (**b**), GLI1 (**c**) or GLI2 (**d**) siRNAs. Data are presented as mean ± SEM ($n = 3$ biologically independent experiments). *P*-values were calculated using one-way ANOVA with post-hoc Dunnett's multiple comparisons test. **e–g** Representative immunoblot (top) and qRT-PCR (bottom) analysis monitoring EWSR1::FLI1 protein (using an anti-FLI1 antibody) and relative mRNA levels, respectively, following treatment of A673 cells with cyclopamine (**e**), GANT-61 (**f**) or SAG (**g**) for 48 h. Data are presented as mean ± SEM ($n = 3$ biologically independent experiments). *P*-values were calculated using one-way ANOVA with post-hoc

Dunnett's multiple comparisons test. **h** Schematic showing the GLI consensus sequence (of which the last five residues, shown in red, are the most critical for binding) and locations and sequences of GLI-binding motifs in the promoter (P1, P2, or P3) or intron 2 (I2) of *EWSR1*. **i** ChIP assay in A673 cells monitoring binding of GLI1 and GLI2 to *EWSR1* promoter regions containing GLI-binding motifs or, as a negative control, a gene desert (GD) region. IgG was used as an isotype control for ChIP. Data are presented as mean ± SEM ($n = 3$ biologically independent experiments). **j** qRT-PCR analysis (left) and representative immunoblot (right) monitoring EWSR1::FLI1 relative mRNA levels and protein (using an anti-FLI1 antibody), respectively, in A673 cells in which the GLI-binding sites P2 and P3 have been deleted. Data are presented as mean ± SEM ($n = 3$ biologically independent experiments). *P*-values were calculated using one-way ANOVA with post-hoc Dunnett's multiple comparisons test. Source data are provided as a Source Data file.

single GalNAc monosaccharide to more than 20 saccharides, which are added to the target protein through sequential glycosyltransferase reactions that occur in the Golgi[39]. Following the initial addition of a GalNAc monosaccharide to serine or threonine (forming so-called Tn antigen), the emerging glycan structure can diverge into one of eight different core extension pathways; C1GALT1 is the only glycosyltransferase that catalyzes the addition of galactose to GalNAc to synthesize the core 1 structure (also called T antigen), the most common O-GalNAc core[39]. Previous large-scale glycoproteomic studies in kidney and T cells have identified SMO as an O-GalNAc glycoprotein[40] harboring glycans with a core 1 structure[41], indicating it is a C1GALT1 target. We therefore considered the possibility that the mechanism by which C1GALT1 stimulates Hh signaling is through O-glycosylation of SMO, which may enhance SMO activity.

To determine whether in ES cells SMO harbors the T antigen moiety produced by C1GALT1, we performed a lectin pull-down assay. In brief, A673 cell extracts were incubated with peanut agglutinin (PNA) lectin beads, which bind strongly to T antigen, and the eluate from the beads was assayed for SMO by immunoblotting; in a reciprocal experiment, extracts were incubated with an anti-SMO antibody, and the presence of T antigen in the immunoprecipitate was detected by immunoblotting using biotinylated PNA. Figure 5a shows that SMO bound to PNA lectin and that binding was substantially reduced following treatment with O-glycosidase, which catalyzes the removal of core 1 Gal-GalNAc disaccharides from O-glycoproteins, indicating that SMO is O-glycosylated (harboring Gal-GalNAc disaccharides) in ES cells. Similar results were obtained using another T antigen-binding lectin, jacalin (Supplementary Fig. 5a). Immunoblotting of whole-cell extracts revealed that O-glycosidase treatment also resulted in a decrease in the apparent molecular weight of SMO (Fig. 5a and Supplementary Fig. 5a). Consistent with the idea that SMO is a C1GALT1 target, co-immunoprecipitation experiments in A673 cells revealed that SMO and C1GALT1 interact (Fig. 5b).

We next sought to determine the consequences of the loss of C1GALT1 activity on SMO O-glycosylation and function. Because SMO is undetectable in C1GALT1 knockdown A673 cells (see Fig. 4a), we performed these experiments by inhibiting C1GALT1 with ITZ, which enabled us to monitor SMO O-glycosylation at early time points following loss of C1GALT1. Treatment of A673 cells with ITZ led to the rapid decrease of O-glycosylated SMO, as assessed by PNA lectin pull-down assay, with a subsequent loss of total SMO levels, as well as GLI1 and EWSR1::FLI1 (Fig. 5c). Similarly, treatment of A673 cells with Ac$_5$GalNTGc, an inhibitor of O-glycosylation that prevents elongation of O-glycans beyond the Tn antigen[42] and therefore effectively blocks O-glycan elongation at the same step as C1GALT1 inhibition, also reduced SMO O-glycosylation, as well as the levels of SMO, GLI1 and EWSR1::FLI1 (Supplementary Fig. 5b).

The reduced levels of SMO observed upon genetic or pharmacological inhibition of C1GALT1 suggest that loss of C1GALT1-mediated O-glycosylation of SMO reduces its stability. Consistent with this

possibility, cycloheximide chase analysis revealed that the half-life of SMO was substantially reduced in the presence of ITZ (Fig. 5d). Notably, the reduction in SMO levels observed upon C1GALT1 knockdown or inhibition by ITZ was reversed by addition of the proteasome inhibitor MG132 or the lysosome inhibitor bafilomycin A1 (Fig. 5e). Using MG132 or bafilomycin A1 treatment allowed us to circumvent the issue of SMO undetectability in C1GALT1 knockdown cells, and we performed a PNA lectin pull-down assay, which revealed that SMO O-glycosylation was substantially reduced following shRNA-mediated knockdown of C1GALT1 (Fig. 5f).

The above results suggested that SMO mutants that are unable to be O-glycosylated by C1GALT1 would have reduced stability. We, therefore, performed site-directed mutagenesis to generate SMO derivatives in which two O-GalNAc glycosylation sites identified through glycoproteomic studies, Thr-55 or Thr-500[41], were mutated to Ala, and stably expressed the mutants as FLAG-tagged versions in parallel with wild-type SMO in A673 cells. The immunoblot of Fig. 5g shows that the SMO(T55A) and SMO(T500A) mutant proteins were expressed at low or undetectable levels, which could be restored by the addition of MG132 or bafilomycin A1. Similar results were obtained by immunofluorescence microscopy (Supplementary Fig. 5c). These results suggest that the SMO mutants are degraded through a proteasomal- and/or lysosomal-dependent mechanism. To test this idea, we performed a ubiquitination assay in A673 cells stably expressing either FLAG-SMO, FLAG-SMO(T55A) or FLAG-SMO(T500A). As shown in Fig. 5h, the SMO mutants were polyubiquitinated, suggesting that their reduced stability was due, at least in part, to ubiquitin-mediated protein degradation. As expected, the steady-state mRNA level of the SMO mutants was unaffected by treatment with MG132 or bafilomycin A1 (Supplementary Fig. 5d). Collectively, these results suggest that C1GALT1-mediated O-glycosylation of SMO enhances its stability in ES cells.

## C1GALT1 is required for EWSR1::FLI1-mediated transformation

We next performed a series of experiments to investigate the role of C1GALT1 in cellular transformation. As mentioned above, A673 ES cells do not require EWSR1::FLI1 for proliferation in cell culture but are dependent on EWSR1::FLI1 for the transformed phenotype[9]. Similar to shRNA-mediated knockdown of EWSR1, knockdown of SMO or, notably, C1GALT1 substantially reduced the ability of A673 cells to form colonies in soft agar (Fig. 6a and Supplementary Fig. 6a). To test the tumor-forming ability of these cells, we subcutaneously injected them into the flank of NSG mice and monitored tumor growth. Figure 6b shows that similar to shRNA-mediated knockdown of EWSR1, knockdown of SMO or C1GALT1 substantially reduced the ability of A673 cells to form tumors in mice. IHC analysis of the residual tumors derived from C1GALT1 knockdown cells revealed substantially reduced levels of FLI1, SMO, and GLI1 (Fig. 6c and Supplementary Fig. 6b), confirming that the reduction in EWSR1::FLI1, SMO, and GLI1 expression following C1GALT1 knockdown that we observe in cultured cells also occurs in vivo. In addition, the tumors displayed reduced

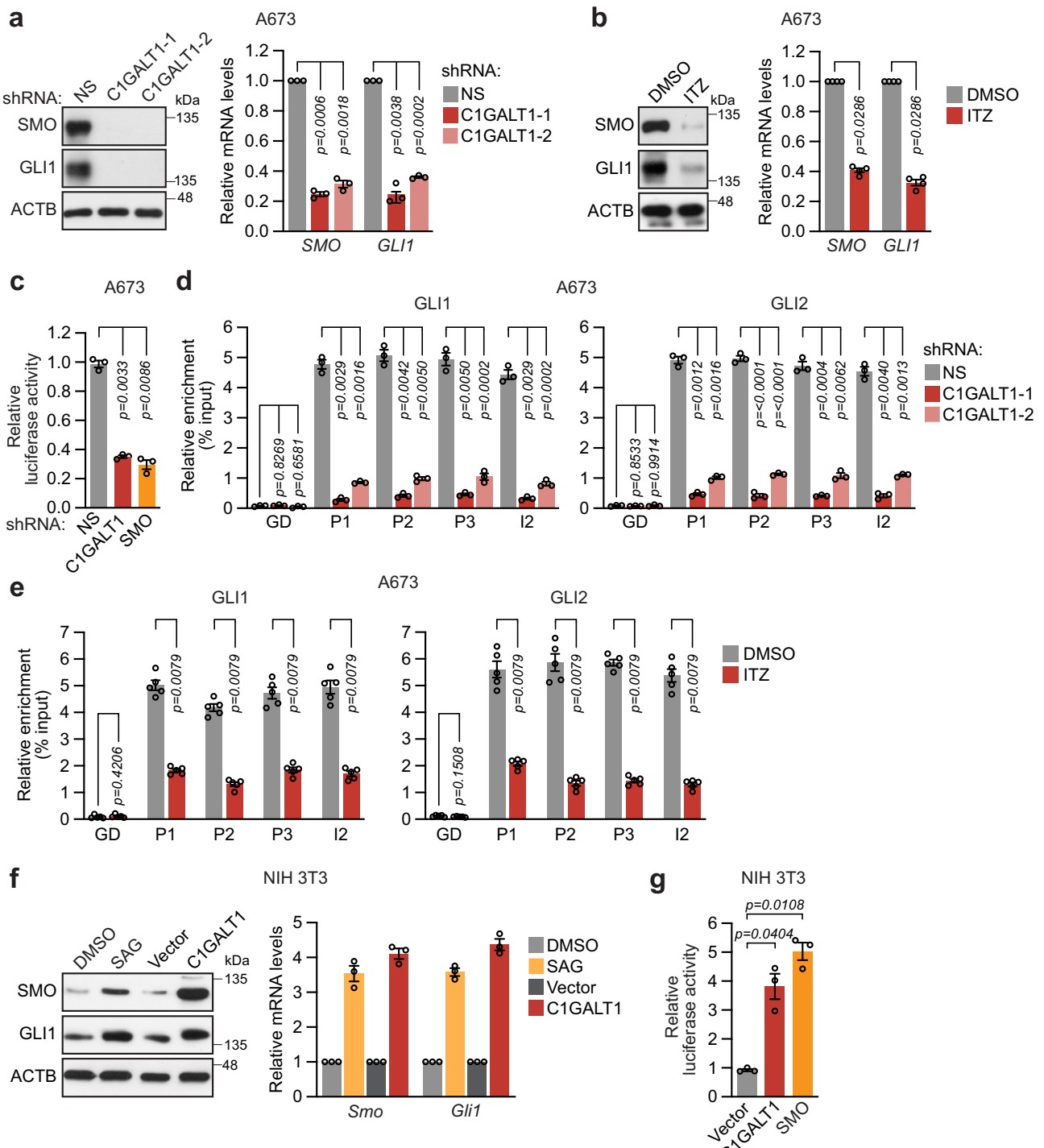

**Fig. 4 | C1GALT1 stimulates Hh signaling in ES cells. a, b** Representative immunoblot (left) and qRT-PCR (right) analysis monitoring SMO and GLI1 protein and relative mRNA levels, respectively, in A673 cells stably expressing an NS or one of two independent C1GALT1 shRNAs (**a**) or treated with DMSO or ITZ (100 nM for 72 h) (**b**). qRT-PCR data are presented as mean ± SEM (*n* = 3 [**a**] or 4 [**b**] biologically independent experiments). *P*-values were calculated using one-way ANOVA with post-hoc Dunnett's multiple comparisons test (**a**) or two-tailed Mann–Whitney test (**b**). **c** GLI-driven luciferase reporter assay. Relative luciferase activity in A673 cells transfected with a luciferase reporter plasmid driven by 10 GLI-binding sites and stably expressing a NS, C1GALT1, or SMO shRNA. Data are presented as mean ± SEM (*n* = 3 biologically independent experiments). *P*-values were calculated using one-way ANOVA with post-hoc Dunnett's multiple comparisons test. **d, e** ChIP analysis monitoring binding of GLI1 and GLI2 to the *EWSR1* promoter and intron 2 regions

containing GLI-binding motifs or, as a negative control, a GD region, in A673 cells stably expressing a NS or C1GALT1 shRNA (**d**) or treated with DMSO or ITZ (100 nM for 72 h) (**e**). Data are presented as mean ± SEM (*n* = 3 [**d**] or 5 [**e**] biologically independent experiments). *P*-values were calculated using two-way ANOVA with post-hoc Tukey's multiple comparisons test (**d**) or two-tailed Mann–Whitney test (**e**). **f** Representative immunoblot (left) and qRT-PCR (right) analysis monitoring SMO and GLI1 protein and relative mRNA levels, respectively, in NIH 3T3 cells following treatment with DMSO or SAG, or expression of vector or C1GALT1. Data are presented as mean ± SEM (*n* = 3 biologically independent experiments). **g** GLI-driven luciferase reporter assay in NIH 3T3 cells expressing vector, C1GALT1 or SMO. Data are presented as mean ± SEM (*n* = 3 biologically independent experiments). *P*-values were calculated using one-way ANOVA with post-hoc Dunnett's multiple comparisons test. Source data are provided as a Source Data file.

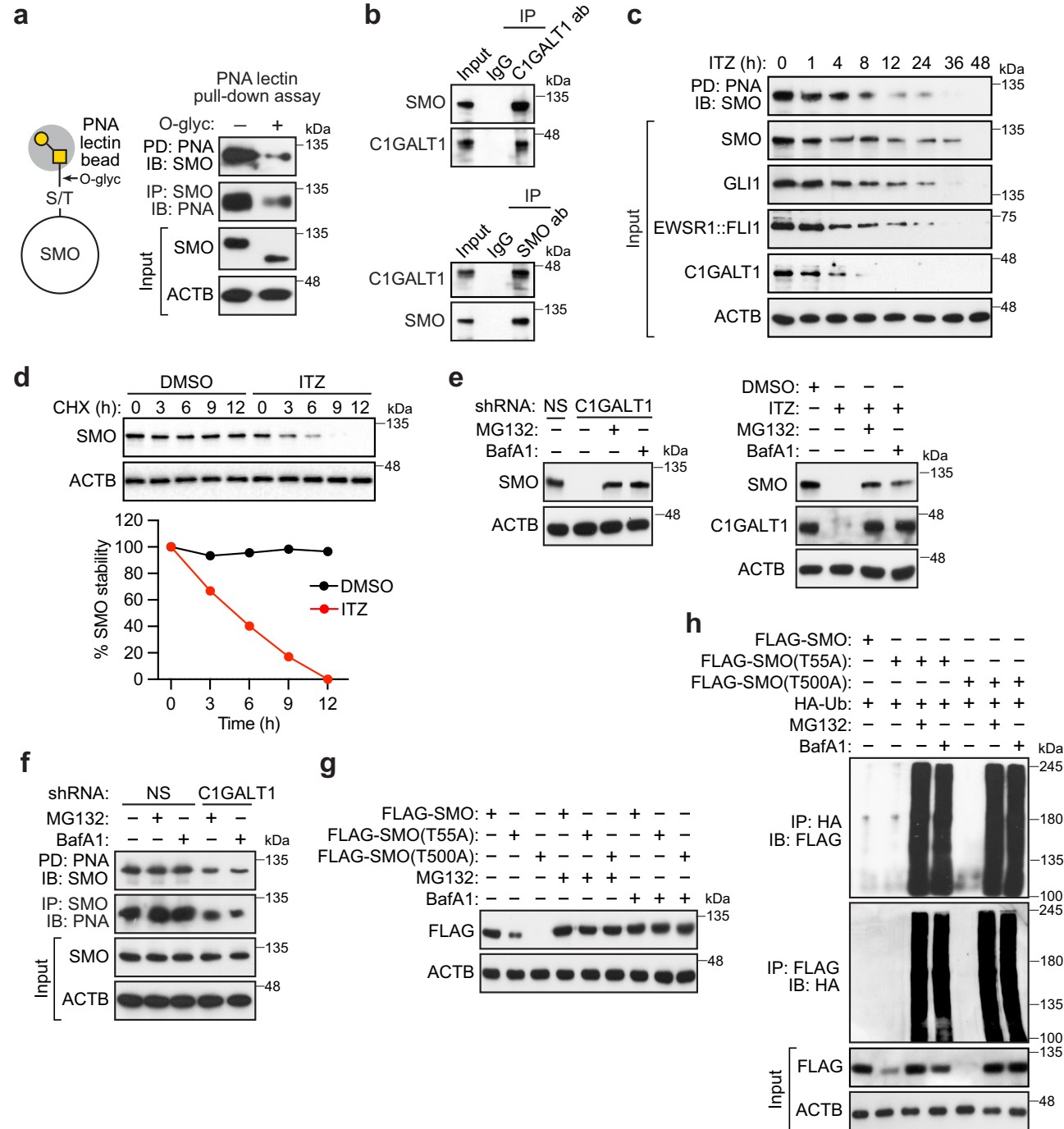

**Fig. 5 | C1GALT1 promotes O-glycosylation and stabilization of SMO in ES cells.**
**a** PNA lectin pull-down assay. (Left) Schematic showing a PNA lectin bead (gray) bound to T antigen, comprising GalNAc (yellow square) and Gal (yellow circle), which is attached to SMO through serine or threonine (S/T). The O-glycosidase (O-glyc) cleavage site is shown. (Right) Representative immunoblot showing the level of O-glycosylated SMO, detected by either PNA lectin pull-down (PD) followed by SMO immunoblot (IB) analysis or SMO immunoprecipitation (IP) followed by PNA IB, in A673 cell lysates treated with O-glycosidase and α2-3,6,8 neuraminidase, which removes terminal sialic residues that inhibit lectin binding and O-glycosidase cleavage. **b** Representative co-IP assay in A673 cells monitoring the presence of SMO in a C1GALT1 IP, performed using a C1GALT1 antibody (ab), or vice versa.
**c** Representative immunoblot showing levels of O-glycosylated SMO, total SMO, GLI1, EWSR1::FLI1 (monitored using an anti-FLI1 antibody), and C1GALT1 in A673 cells treated with 100 nM ITZ. **d** (Top) Representative immunoblot showing SMO levels in

a cycloheximide (CHX) chase assay performed in A673 cells treated with DMSO or 100 nM ITZ. (Bottom) Quantification; time 0 in the presence of DMSO or ITZ was set to 100%. The plot shows the average SMO levels from two independent experiments. **e** Representative immunoblot showing SMO levels in A673 cells stably expressing an NS or C1GALT1 shRNA (left) or treated with DMSO or 100 nM ITZ for 72 h (right) and treated in the presence or absence of MG132 or bafilomycin A1 (BafA1).
**f** Representative immunoblot showing the level of O-glycosylated SMO in A673 cells stably expressing an NS or C1GALT1 shRNA and treated in the presence or absence of MG132 or BafA1. **g** Representative immunoblot showing FLAG-SMO levels in A673 cells stably expressing FLAG-tagged wild-type SMO, SMO(T55A) or SMO(T500A) in the presence or absence of MG132 or BafA1. **h** Representative ubiquitination assay showing the levels of polyubiquitinated SMO(T55A) or SMO(T500A) in A673 cells expressing HA-tagged ubiquitin (HA-Ub) and treated in the presence or absence of MG132 or BafA1. Source data are provided as a Source Data file.

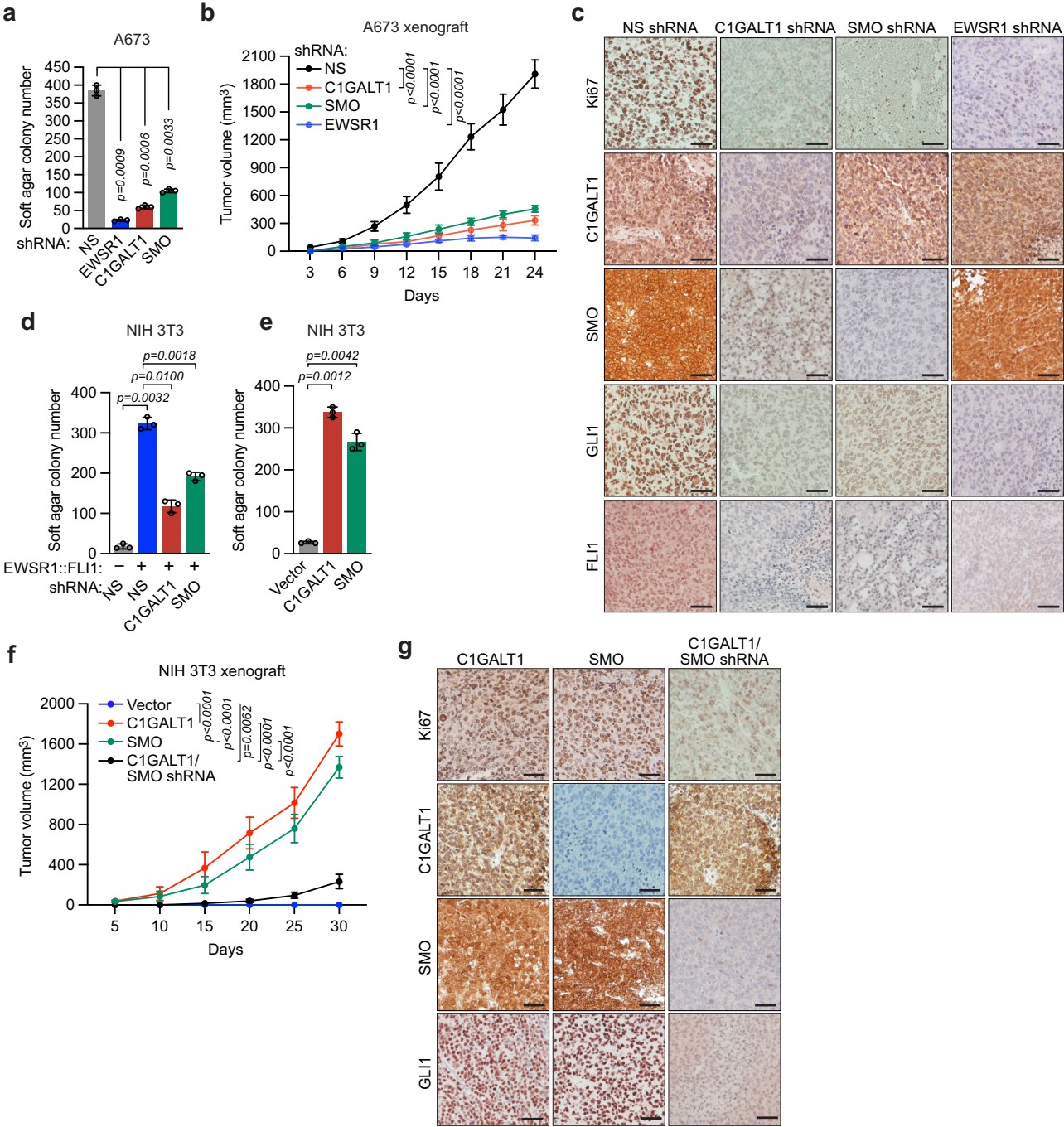

**Fig. 6 | C1GALT1 is required for EWSR1::FLI1-mediated transformation. a** Soft agar assay monitoring colony formation of A673 cells stably expressing a NS, EWSR1, C1GALT1, or SMO shRNA. Data are presented as mean ± SD (*n* = 3 biologically independent experiments). *P*-values were calculated using one-way ANOVA with post-hoc Dunnett's multiple comparisons test. **b** Tumor formation of A673 cells stably expressing a NS, C1GALT1, SMO, or EWSR1 shRNA following subcutaneous injection into NSG mice. Tumor dimensions were monitored every 3 days starting at day 3, when tumors became palpable. Data are presented as mean ± SD (*n* = 6 mice per group). *P*-values were calculated at day 24 using two-way ANOVA with post-hoc Tukey's multiple comparisons test. **c** Representative IHC images showing Ki67, C1GALT1, SMO, GLI1, and FLI1 staining in tumors derived from mice in (b) at day 24. Scale bar, 50 μm. **d** Colony formation of NIH 3T3 cells expressing vector or EWSR1::FLI1 and an NS, C1GALT1, or SMO shRNA. Data are presented as mean ± SD

(*n* = 3 biologically independent experiments). *P*-values were calculated using one-way ANOVA with post-hoc Tukey's multiple comparisons test. **e** Colony formation of NIH 3T3 cells expressing vector, SMO or C1GALT1. Data are presented as mean ± SD (*n* = 3 biologically independent experiments). *P*-values were calculated using one-way ANOVA with post-hoc Dunnett's multiple comparisons test. **f** Tumor formation of NIH 3T3 cells ectopically expressing SMO or C1GALT1, or cells ectopically expressing C1GALT1 and an SMO shRNA, following subcutaneous injection into NSG mice. Tumor dimensions were monitored every 5 days starting at day 5, when tumors became palpable. Data are presented as mean ± SD (*n* = 5 mice per group). *P*-values were calculated at day 30 using two-way ANOVA with post-hoc Tukey's multiple comparisons tests. **g** Representative IHC images showing Ki67, C1GALT1, SMO, and GLI1 staining in tumors derived from mice in (**f**) at day 30. Scale bar, 50 μm. Source data are provided as a Source Data file.

K167 staining, indicative of reduced tumor cell proliferation. Collectively, these results indicate that Hh signaling and C1GALT1 are required for the transformed phenotype of A673 cells.

To independently confirm these results, we asked whether Hh signaling and C1GALT1 were required for the ability of EWSR1::FLI1 to transform NIH 3T3 cells. Consistent with previous studies[43], ectopic expression of EWSR1::FLI1 (Supplementary Fig. 6c) enabled NIH 3T3 cells to form colonies in soft agar (Fig. 6d and Supplementary Fig. 6d), which was substantially reduced by knockdown of SMO or C1GALT1 (Fig. 6d). Collectively, these results indicate that C1GALT1 and Hh signaling are required for EWSR1::FLI1-mediated cellular transformation.

Finally, we investigated whether high levels of C1GALT1 were sufficient for cellular transformation. Previous studies have shown that NIH 3T3 cells can be transformed by increased Hh signaling[44]. We therefore asked whether ectopic expression of C1GALT1, which increases Hh signaling, could also transform NIH 3T3 cells. Ectopic expression of either C1GALT1 or, as a positive control, SMO enabled NIH 3T3 cells to form colonies in soft agar (Fig. 6e and Supplementary Fig. 6d) and tumors in mice (Fig. 6f and Supplementary Fig. 6e–h). Tumors derived from C1GALT1-expressing NIH 3T3 cells displayed high levels of SMO and GLI1, and active proliferation, as evidenced by positive Ki67 staining (Fig. 6g). Notably, the ability of ectopic C1GALT1 expression to transform NIH 3T3 cells was abrogated by knockdown of SMO (Fig. 6f, g). Based on these results, and on those of Fig. 4f–g, we conclude that C1GALT1 transforms NIH 3T3 cells by stimulating Hh signaling in a SMO-dependent manner.

### ITZ inhibits the proliferation of human ES cell lines in culture and suppresses the growth of ES xenografts in mice

Finally, we analyzed the effect of ITZ on the proliferation of ES cells in culture and the growth of ES xenografts in mice. These experiments were performed in a panel of ES cell lines that express high levels of C1GALT1 (Supplementary Fig. 3d): TC-32 and TC-71, which require EWSR1::FLI1 for viability and proliferation in culture[45], and TC-106, which expresses another ES fusion gene, EWSR1::ERG[46], that shares the *EWSR1* promoter and is therefore expected to be transcriptionally regulated by the same mechanism as *EWSR1::FLI1*. Consistent with our results in A673 cells, we found that Hh signaling was required for expression of EWSR1::FLI1/ERG in these cell lines (Supplementary Fig. 7). We next performed several key experiments to confirm in these ES cell lines that EWSR1::FLI1/ERG expression is promoted by the same C1GALT1-mediated Hh signaling mechanism that we elucidated in A673 cells. We found that C1GALT1 was required for viability of these cell lines (Supplementary Fig. 8a), precluding us from deriving stable C1GALT1 knockdown derivatives, and therefore subsequent experiments were performed by transducing cells with high titer C1GALT1 shRNA lentivirus followed by short-term drug selection; under these conditions, we observed efficient reduction of C1GALT1 levels (Supplementary Fig. 8b) without a marked decrease in cell viability (Supplementary Fig. 8c). Consistent with our results in A673 cells, knockdown of C1GALT1 or ITZ treatment reduced both Hh signaling and EWSR1::FLI1/ERG expression (Supplementary Fig. 8b, d–f). C1GALT1 knockdown or ITZ treatment also abrogated the binding of GLI1 and GLI2 to *EWSR1* (Supplementary Fig. 8g, h). Moreover, SMO was O-glycosylated (Supplementary Fig. 9a, b), which was reduced by ITZ treatment (Supplementary Fig. 9b), and O-glycosylation-defective SMO mutants had reduced stability (Supplementary Fig. 9c). Finally, C1GALT1 knockdown substantially reduced the ability of these cells to form colonies in soft agar (Supplementary Fig. 9d) and tumors in mice (see below).

As expected, ITZ treatment led to a dose-dependent reduction in ES cell viability (Fig. 7a and Supplementary Fig. 10a), which was due, at least in part, to the induction of apoptosis, as evidenced by an increase in PARP1 cleavage (Fig. 7b and Supplementary Fig. 10b). Notably, in TC-71 cells, ITZ displayed substantially higher efficacy compared to the FDA-approved SMO antagonists sonidegib and vismodegib, which required much higher concentrations (50-200 times that of ITZ) to achieve a comparable decrease in cell viability (compare Supplementary Fig. 10c to Fig. 7a), Hh signaling and *EWSR1::FLI1* expression (compare Supplementary Fig. 10d to Supplementary Fig. 8d, f). These results suggested the possibility that ITZ may exert its effects in ES cells through pathways in addition to inhibition of the C1GALT1-Hh signaling-EWSR1::FLI1 axis. However, two lines of evidence argue against this possibility. First, the ITZ-mediated decrease in TC-71 cell viability was counteracted by ectopic expression of C1GALT1 or EWSR1::FLI1 under the control of a constitutive CMV promoter (Fig. 7c), indicating that the reduced levels of C1GALT1 and EWSR1::FLI1 following ITZ treatment were responsible for the proliferation defect. Second, at the highest dose of ITZ used in our experiments (100 nM), we did not observe inhibition of other signaling pathways known to be targeted by ITZ, including AKT/mTOR and Wnt/β-catenin[47–49] (Supplementary Fig. 10e). Collectively, these results suggest that ITZ exerts its effects on *EWSR1::FLI1* expression and cell viability primarily through inhibition of C1GALT1-mediated Hh signaling.

Sonidegib and vismodegib, as well as cyclopamine, act by directly binding to a pocket in SMO that is enclosed by transmembrane helices and extracellular loops[50]. We, therefore, hypothesized that the relative insensitivity of ES cells to these conventional SMO inhibitors could be due to C1GALT1-induced glycosylation of SMO, which may block the ability of the SMO inhibitors to bind to the drug-binding pocket. To test this model, we ectopically expressed C1GALT1 (or, as a control, empty vector) in NIH 3T3 cells, treated them with cyclopamine, sonidegib, or vismodegib, and monitored GLI1 mRNA and protein levels. We first confirmed that ectopic expression of C1GALT1 increased SMO O-glycosylation, as expected (Supplementary Fig. 11a), consistent with the increase in SMO levels we observe (see Fig. 4f). We found that ectopic expression of C1GALT1 abrogated the decrease in GLI1 observed upon treatment of NIH 3T3 cells with cyclopamine, vismodegib or sonidegib (Supplementary Fig. 11b–d), strongly supporting the notion that C1GALT1-mediated increase in SMO O-glycosylation renders cells resistant to the conventional SMO inhibitors. We observed that ectopic expression of C1GALT1 also abrogated the decrease in GLI1 levels observed upon ITZ treatment (Supplementary Fig. 11e), consistent with the results presented in Fig. 7c.

Finally, we tested the effect of ITZ treatment on the growth of ES xenografts in mice. In brief, ES cells were injected subcutaneously into the flank of NSG mice, and once tumors were established, mice were treated daily with 100 mg/kg ITZ by oral gavage. Growth of TC-71, TC-32, or TC-106 xenografts was substantially suppressed by ITZ treatment (Fig. 7d and Supplementary Fig. 12a, b). IHC analysis revealed substantially reduced levels of FLI1, C1GALT1, SMO, and GLI1, as well as reduced Ki67 staining, in tumors from ITZ-treated mice (Fig. 7e and Supplementary Fig. 12c, d), confirming the reduction in EWSR1::FLI1 levels and Hh signaling following ITZ treatment in vivo. The dosage of ITZ used in these experiments (100 mg/kg/d) was comparable to that used in previous studies and has shown to be non-toxic and well tolerated[29,30,51], which we also found, as evidenced by the absence of weight loss in ITZ-treated mice (Supplementary Fig. 12e).

Consistent with the results in cultured cells, the ability of ITZ to suppress TC-71 xenograft growth was counteracted by ectopic expression of EWSR1::FLI1 (Fig. 7d), indicating that the reduced levels of EWSR1::FLI1 following ITZ treatment were responsible for the reduction in tumor growth. Furthermore, the ability of ITZ to suppress the growth of TC-71 or TC-32 xenografts was comparable to that observed upon knockdown of C1GALT1, and was not further exacerbated by C1GALT1 knockdown (Fig. 7f and Supplementary Fig. 12f), indicating that ITZ exerts its effects on ES tumor growth by inhibiting C1GALT1.

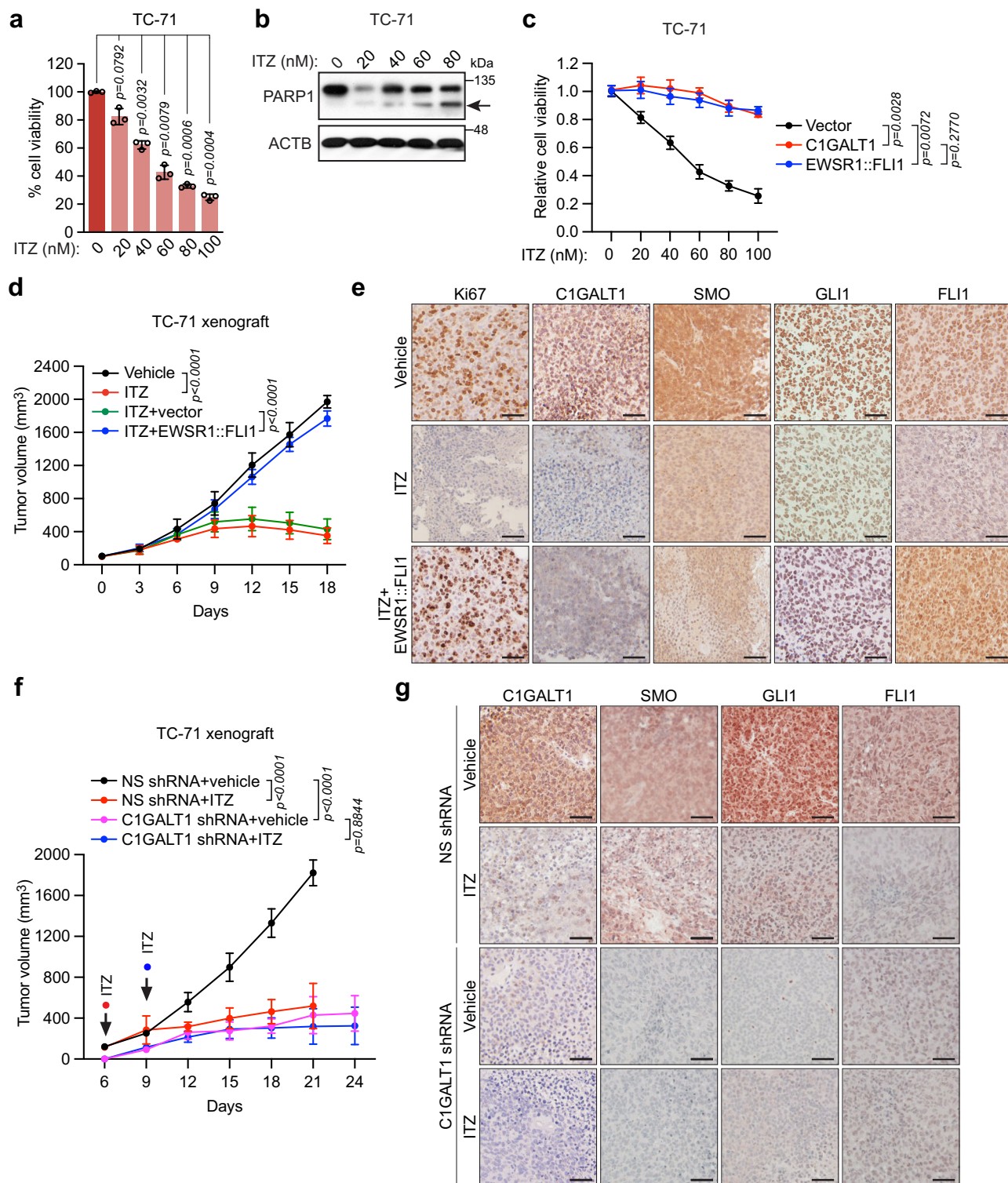

## Discussion

In this study, using a genome-scale CRISPR/Cas9 screen, we identified C1GALT1 as a factor that promotes the expression of *EWSR1::FLI1* in ES cells. Based on our collective results, a model for the relationship between C1GALT1, Hh signaling and *EWSR1::FLI1* expression is presented in Fig. 8. In ES cells, high levels of C1GALT1 stimulate the Hh signaling pathway in a non-canonical (i.e., ligand-independent) manner through O-glycosylation of SMO, which blocks proteasomal/lysosomal degradation of SMO and results in its stabilization. Our results are reminiscent of a previous study showing that in the absence of ligand-

induced Hh pathway activation, SMO undergoes ubiquitin-mediated degradation and removal from the primary cilia; and that blocking ubiquitination of SMO, using either an E1 ligase inhibitor or by mutating lysine residues required for ubiquitination, causes SMO to aberrantly accumulate in primary cilia without Hh pathway activation[52]. Furthermore, we find that the Hh signaling effectors GLI1 and GLI2 directly bind *EWSR1* and promote *EWSR1::FLI1* expression. Notably, previous studies have found that EWSR1::FLI1 promotes *GLI1* expression and thus Hh signaling[33,36], thereby establishing a positive feedback loop. Inhibition of C1GALT1 using a genetic or

**Fig. 7 | ITZ inhibits the proliferation of human ES cell lines in culture and suppresses the growth of ES xenografts in mice. a** Viability TC-71 cells treated with DMSO or ITZ for 72 h. Data are presented as mean ± SD ($n = 3$ biologically independent experiments). *P*-values were calculated using one-way ANOVA with post-hoc Dunnett's multiple comparisons test. **b** Representative immunoblot showing PARP levels in TC-71 cells treated with DMSO or ITZ for 48 h. Cleaved PARP is indicated by the arrow. **c** Viability of TC-71 cells expressing vector, C1GALT1 or EWSR1::FLI1 and treated with DMSO or ITZ for 72 h. Data are presented as mean ± SD ($n = 3$ biologically independent experiments). *P*-values were calculated at 100 nM ITZ using two-way ANOVA with post-hoc Tukey's multiple comparisons test. **d** Tumor formation. TC-71 cells, either parental or expressing vector or EWSR1::FLI1, were injected subcutaneously into NSG mice and when tumors reached ~100 mm³ (denoted as day 0), mice were treated daily with vehicle or ITZ

(100 mg/kg), and tumor dimensions were monitored every 3 days. Data are presented as mean ± SD ($n = 6$ mice per group). *P*-values were calculated on day 18 using two-way ANOVA with post-hoc Tukey's multiple comparisons tests. **e** Representative IHC images showing Ki67, C1GALT1, SMO, GLI1 and FLI1 staining in TC-71 xenografts at day 18. Scale bar, 50 μm. **f** Tumor formation. TC-71 cells expressing a NS or C1GALT1 shRNA were injected subcutaneously into NSG mice, and when tumors reached ~100 mm³ (day 6 for NS shRNA-expressing tumors or day 9 for C1GALT1 shRNA-expressing tumors), mice were treated daily with vehicle or ITZ. Data are presented as mean ± SD ($n = 6$ mice per group). *P*-values were calculated following 15 days of treatment using mixed-effect analysis with post-hoc Tukey's multiple comparisons tests. **g** Representative IHC images showing C1GALT1, SMO, GLI1, and FLI1 staining in tumors derived from mice in (**f**). Scale bar, 50 μm. Source data are provided as a Source Data file.

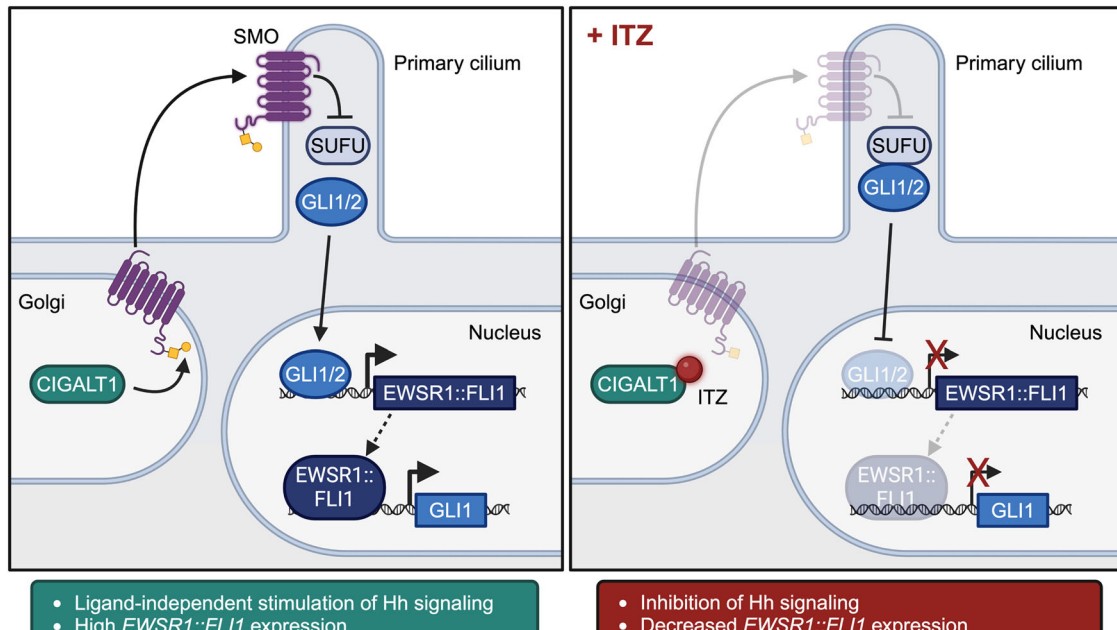

**Fig. 8 | Model for C1GALT1-mediated regulation of Hh signaling and EWSR1:: FLI1 expression.** In ES cells, C1GALT1 (a resident Golgi protein) O-glycosylates SMO (as indicated by the yellow square and circle), resulting in SMO stabilization and release of GLI1/2 from sequestration by SUFU. Translocation of GLI1/2 to the nucleus promotes EWSR1::FLI1 expression and drives ES cell survival and tumor

formation. GLI1 is also a direct target of EWSR1::FLI1. Inhibition of C1GALT1 by ITZ results in destabilization of SMO and de-activation of the Hh signaling pathway, which in turn decreases *EWSR1::FLI1* expression and reduces ES cell viability and tumor growth. Created in BioRender.com[69].

---

pharmacological inhibitor, such as ITZ, reduces SMO O-glycosylation, Hh signaling, and *EWSR1::FLI1* expression, resulting in decreased ES cell viability and tumor growth in mice. Our results are applicable to other oncogenic ES fusion-proteins driven by *EWSR1*, as genetic or pharmacological inhibition of C1GALT1 also decreases cell viability and tumor growth of ES cells that harbor an EWSR1::ERG fusion.

Our findings suggest that ES cells depend on continuous C1GALT1-mediated glycosylation and stabilization of SMO, which is required for their growth, proliferation, and ability to form tumors. However, given our proposed model that there is a reciprocal positive feedback loop between EWSR1::FLI1 and GLI1 that is SMO independent, it is not clear why ES cells would continuously require SMO. One possible explanation is that SMO may be required to inactivate SUFU to allow translocation of the GLI transcription factors to the nucleus (see Fig. 8).

Previous studies have highlighted roles for C1GALT1 in cancer development and/or progression through diverse mechanisms. For example, C1GALT1 has been shown to promote cancer by O-glycosylating receptor tyrosine kinases, such as EGFR in head and neck squamous carcinoma[16] and prostate cancer[19], FGFR in colorectal cancer[21] and EPHA2 in gastric adenocarcinoma[15], resulting in inhibition

of these oncogenic signaling pathways. In addition, C1GALT1 has been shown to target integrins, including integrin $\alpha_v$ in pancreatic ductal adenocarcinoma[17] and integrin β1 in hepatocellular carcinoma[18], thereby enhancing cell-extracellular matrix adhesion and promoting cell invasiveness. Interestingly, a very recent study found that in osteosarcoma, C1GALT1 is highly expressed and promotes cell proliferation and tumor growth by maintaining ERK signaling[53]. Our results strongly suggest that in ES, the primary target of C1GALT1 is SMO. However, we cannot exclude the possibility that C1GALT1 may promote ES cell viability and tumor growth, in part, through O-glycosylation of other substrates, such as integrins or receptor tyrosine kinases.

Due to its pivotal role in Hh signaling and its presence on the cell membrane, SMO has garnered significant interest as a target for the development of small molecule inhibitors of the Hh pathway[50]. To date, two SMO inhibitors, vismodegib, and sonidegib, have received FDA approval for the treatment of basal cell carcinoma[54]. However, concerns about the emergence of SMO mutants in the drug-binding pocket that are resistant to vismodegib and sonidegib[50], as well as the limited demonstrated efficacy of vismodegib and sonidegib across

broad tumor types, have highlighted the limitations of these SMO antagonists as Hh pathway inhibitors for clinical use.

Our results uncovered a modality for targeting SMO through inhibition of C1GALT1 by ITZ. Previous work showed that ITZ acts on SMO through a mechanism that is distinct from cyclopamine, but which at the time was not well defined[29]. In this study, we have shown that ITZ acts on SMO activity indirectly by inhibiting C1GALT1. Although previous studies had shown that ITZ is an inhibitor of both C1GALT1[16] and Hh signaling[29], to our knowledge the link between C1GALT1 and Hh signaling has not been previously reported, and thus our results reveal C1GALT1 as a target for inhibiting Hh signaling.

Although previous reports have described similar IC50s for ITZ and the SMO inhibitor cyclopamine in NIH 3T3 cells (800 nM and 300 nM, respectively)[29], we observed much higher IC50s for cyclopamine, as well as sonidegib and vismodegib, in ES cells (5-20 μM). We speculated that the relative insensitivity of ES cells to these conventional SMO inhibitors is due to C1GALT1-induced O-glycosylation of SMO, which may block the ability of the inhibitors to bind to the drug-binding pocket of SMO. Indeed, we found that in NIH 3T3 cells, increasing C1GALT1-mediated SMO O-glycosylation confers resistance to cyclopamine, sonidegib, and vismodegib (Supplementary Fig. 11). Our results establish O-glycosylation-dependent stabilization of SMO as a potential resistance mechanism to FDA-approved conventional SMO inhibitors.

Our results suggest that ITZ is a potential therapeutic agent for ES. Notably, a previous study has shown that the dosage of ITZ typically used in animal experiments to inhibit tumor growth (100 mg/kg/d) results in serum levels in mice that are comparable to those reported in patients undergoing high-dose ITZ anti-fungal therapy[29], indicating the dosage is clinically achievable. Moreover, the results from an open-label exploratory phase II clinical trial of oral ITZ in human basal cell carcinoma demonstrated that administration of ITZ at standard therapeutic doses ranging from 100-200 mg twice a day for one month reduced tumor proliferation and Hh pathway activity[55], indicating that the dosage of ITZ used as an anti-fungal agent is effective for treating Hh-dependent cancer. Interestingly, the combination of ITZ and arsenic trioxide, an FDA-approved drug for refractory or relapsed acute promyelocytic leukemia[56] that has been shown to non-specifically antagonize Hh signaling by targeting GLI[57,58], displays inhibitory activity against SMO mutants that are resistant to other SMO antagonists[51]. In light of our findings, clinical trials focusing on the use of ITZ, either alone or in combination with other Hh pathway antagonists, may be more efficacious than currently available SMO antagonists.

## Methods

### Ethics statement
All experiments were performed following protocols approved by the Institutional Biosafety Committee (IBC) at UMass Chan Medical School. Animal experiments were performed in accordance with the Guide for the Care and Use of Laboratory Animals from NIH, and a protocol (PROTO202000105) approved by the UMass Chan Medical School Institutional Animal Care and Use Committee (IACUC).

### Cell lines and culture
A673 (CRL-1589; female), SK-N-MC (HTB-10; female), IMR-90 (CCL-186; female), NIH 3T3 (CRL-1658), and HepG2 (HB-8065; male) cell lines were obtained from the American Type Culture Collection (ATCC), and TC-32 (female), TC-71 (male) and TC-106 (male) were obtained from The Childhood Cancer Repository at Texas Children's Hospital. A673 and NIH 3T3 cells were cultured in Dulbecco's Modified Eagle's Medium (DMEM) (HyClone, Cat# SH30022.01). IMR90, SK-N-MC, and HepG2 cells were cultured in Eagle's Minimum Essential Medium (EMEM) (ATCC, Cat# 30-2003). TC-71 cells were cultured in Iscove's

Modified Dulbecco's Medium (IMDM) (HyClone, Cat# SH30228.01) supplemented with 1x Insulin-Transferrin-Selenium (ITS; 5 μg/ml insulin (Gibco, Cat# 12585014), 5 μg/ml transferrin (Sigma, Cat# 10652202001), 5 ng/ml selenous acid (Sigma, Cat# 211176)). TC-32 and TC-106 cells were grown in RPMI-1640 Medium (ThermoFisher, Cat # 12633012). All media were supplemented with 10% (v/v) fetal bovine serum (FBS) (Atlanta Biologicals, Cat# S11550), 100 units/ml penicillin, 100 μg/ml streptomycin (Gibco, Cat# 15140122), 1X MEM non-essential amino acids solution (Gibco, Cat# 11140050). All cell lines were cultured at 37 °C and 5% $CO_2$. All cell lines were mycoplasma-negative and routinely tested using a mycoplasma detection kit (ATCC, Cat# 30-1012).

NIH 3T3 cells stably expressing EWSR1::FLI1 were generated by transduction with the lentiviral expression plasmid pCDH-puro-EWS-FLI1 (a gift from Jialiang Wang; Addgene plasmid #102813) or, as a control, pCDH-CMV-MCS-EF1a-Puro (System Biosciences, Cat# CD510B-1) and cells were selected with puromycin for 5 days. NIH 3T3 cells stably expressing SMO or C1GALT1 were generated by transduction of pHAGE-SMO (a gift from Gordon Mills and Kenneth Scott; Addgene plasmid #116792) or as a control pHAGE_puro (a gift from Christopher Vakoc; Addgene plasmid #118692) or a C1GALT1-expression plasmid (GeneCopoeia, Cat#: EX-W0494-Lv120) (or a control plasmid; GeneCopoeia, Cat#: EX-EGFP-Lv120), respectively. A673 and TC-71 cell lines stably expressing FLAG-tagged wild-type SMO, SMO(T55A), or SMO(T500A) used lentiviral plasmids synthesized by GenScript in the pGenLenti vector. TC-71 cells stably expressing EWSR1::FLI1 or C1GALT1 were generated by transduction with pCDH-puro-EWS-FLI1 or the GeneCopoeia C1GALT1-expression plasmid, respectively.

### Generation of the A673/EWSR1::FLI1-tdTomato/EGFP/Cas9 reporter cell line
To generate the reporter cell line using CRISPR/Cas9-mediated homology-directed repair, a donor plasmid was constructed as follows. First, tdTomato cDNA was excised from plasmid pCSCMV:tdTomato (a gift from Gerhart Ryffel; Addgene plasmid #30530) and ligated into pGEM7 (Promega, Cat# P2391), followed by insertion of a DNA fragment encoding T2A (generated by annealing overlapping primers) downstream and in-frame with tdTomato. A Neomycin-SV40 polyA cassette was created by PCR amplification of the Neomycin cDNA from plasmid PGKneobpA (a gift from Philippe Soriano; Addgene plasmid #13342) and PCR amplification of the SV40 polyA terminator from plasmid pSV-β-Galactosidase (Promega, Cat# E1081); the PCR products were denatured and annealed, PCR amplified, digested and ligated downstream of tdTomato-2A to create pTV-tdTomato-2A-Neo-polyA. The sequence correctness of the plasmid was verified by Sanger sequencing. Lastly, 500 bp of the C-terminal end coding sequence (exon 9) and the 3' UTR region of the FLI1 gene were PCR-amplified (using primer pairs FLI1 C-term and FLI1 3'UTR, respectively; see Supplementary Data 3) and subcloned into pTV-tdTomato-2A-Neo-polyA using AscI/NheI and XhoI/NotI sites, respectively.

The donor plasmid (1 μg) was co-transfected into A673 cells, together with 0.2 μg of plasmid pX330-U6-Chimeric_BB-CBh-hSpCas9 (a gift from Feng Zhang; Addgene plasmid #42230) containing a 20 bp sgRNA targeting the EWSR1::FLI1 3' UTR region (FLI1 3'UTR sgRNA, 5'-GGAGGACCAAATTCAGTGG-3'), using Lipofectamine RNAiMAX Transfection Reagent (Invitrogen, Cat# 13778100). After 48 h of co-transfection, cells were selected with 300 μg G418 for 10 days, and tdTomato-positive cells were isolated using a BD FACSAria II cell sorter (BD Biosciences) and seeded in 96-well plates by serial dilution to isolate single-cell clones. The correct insertion site (i.e., the C-terminus of endogenous FLI1 fused to tdTomato) in the knock-in clones was verified by performing RT-PCR and genomic DNA PCR assays using a forward primer in FLI1 exon 9 and a reverse primer in tdTomato (FLI1

Ex9 and tdTomato, respectively; Supplementary Data 3), and a single clone (clone 2) was selected for the reporter cell line. To confirm that knockdown of EWSR1::FLI1 would reduce tdTomato fluorescence, A673/EWSR1::FLI1tdTomato cells ($1 \times 10^5$ cells/well in a 6-well plate) were infected with lentiviral particles expressing a NS, EWSR1 or FLI1 shRNA (Supplementary Data 3), selected with 2 μg/ml puromycin for 5 days, and analyzed by flow cytometry using a BD LSRII flow cytometer (BD Biosciences) using FlowJo v10 (BD Biosciences) software. A673/EWSR1::FLI1tdTomato cells were then infected with lentiviral particles expressing the plasmid lentiCas9-Blast (a gift from Feng Zhang; Addgene plasmid #52962) and selected with 3 μg/ml blasticidin for 7 days. Finally, A673/EWSR1::FLI1tdTomato/Cas9 cells were infected with lentiviral particles expressing the plasmid pLenti CMV EGFP Blast (a gift from Eric Campeau and Paul Kaufman; Addgene plasmid #17445). Cas9 expression was confirmed in A673/EWSR1::FLI1tdTomato/EGFP/Cas9 cells by immunoblotting.

## CRISPR/Cas9 screen

For the genome-wide CRISPR/Cas9 knockout screen, the A673/EWSR1::FLI1tdTomato/EGFP/Cas9 reporter cell line was transduced with the Human CRISPR Knockout Pooled Library (Brunello) (Addgene, Cat# 73178)[10], consisting of ~76,441 sgRNAs targeting 19,114 genes at multiplicity-of-infection (MOI) of 0.3. Cells were selected with 2 μg/ml puromycin for 12 days, followed by 3 days of recovery in the absence of puromycin. At least $1 \times 10^8$ cells were FACS sorted using a BD FACSAria II cell sorter, and the population with tdTomatolow EGFPhigh expression (~7% of the total population) was isolated and expanded. Total genomic DNA was extracted from the sorted and unsorted populations, and 20 μg was used to prepare a next-generation sequencing (NGS) library[59]. In brief, a series of 24 PCR reactions were prepared—each consisting of 10 μl HF Buffer (NEB), 1 μl 10 μM forward primer, 1 μl 10 μM reverse primer, 0.5U Phusion Hot Start II polymerase (NEB, Cat # M0530L), and 1 μl 10 mM dNTPs, with milli Q water and template added to 50 μl—and subjected to the following cycling conditions: 2 min at 98 °C, 20 cycles of 30 s at 98 °C, 30 seconds at 60 °C, and 30 s at 72 °C, and 5 min at 72 °C (primers are listed in Supplementary Data 3). The products of all reactions were pooled, and 2 μl of this PCR1 product was used in a subsequent PCR2 reaction using primers containing adapters for NGS. The same PCR protocol was used but for 15 instead of 20 cycles. PCR products were purified using standard PCR purification columns, DNA concentrations were measured, and samples were equimolarly pooled. NGS was performed using an Illumina MiSeq System and the quality of the raw reads was assessed using FastQC (version 0.11.5). The 20-bp sequences immediately following the pre-guide RNA sequence GGCTTTATATATCTTGTGGAAAGGACGAAACACCG were extracted. These sequences were then mapped to the sgRNA sequences in the human Brunello library using Bowtie[60] (version 1.2.2) with the default parameter settings, except for the options -m 1, −best, and -v 3. Genes were selected for validation and subsequent analyses if at least four sgRNAs were significantly enriched in the tdTomatolow EGFPhigh population; the sgRNAs were considered significantly enriched if they had an odds ratio of at least 2 and a Benjamini-Hochberg (BH)-adjusted $P$-value < 0.05, as determined by a two-tailed Fisher's Exact test. The BH adjustment was applied to control for the false-discovery rate in the context of multiple comparisons[61]. Statistical analysis and figure generation were performed using the *CRISPRscreen* R package, available at https://github.com/LihuaJulieZhu/CRISPRscreen, and deposited at Zenodo[62].

## ShRNA and siRNA knockdown

For stable shRNA knockdowns in A673 cells, $1 \times 10^5$ cells per well were seeded in 6-well plates and transduced with 500 μl lentivirus particles expressing an shRNA (listed in Supplementary Data 3 and obtained from Open Biosystems/Thermo Scientific through the UMass Chan Medical School RNAi Core Facility) in a total volume of 1 ml of appropriate medium supplemented with 10 μg/ml polybrene. Medium was replaced after overnight incubation to remove polybrene and viral particles and cells were subjected to puromycin selection (2 μg/ml) for 3 days. For C1GALT1 shRNA knockdown in TC-32, TC-71, and TC-106 cell lines, $7 \times 10^5$ cells seeded in 6-well plates were transduced with a concentrated, high titer lentivirus for 24 h, selected with puromycin for 12–16 h, and then cultured for 24 h in the drug-free medium; under these conditions >80% cells were viable. For siRNA knockdown, $6 \times 10^5$ A673 cells, at 60% confluency and cultured in Opti-MEM Reduced-Serum Medium (Gibco Cat# 31985-070), were transfected with the following siRNAs (obtained from Dharmacon (Colorado, USA)) at 100 nM using Lipofectamine RNAiMAX Transfection Reagent: GLI1-1 (J-003896-05-0005), GLI1-2 (L-003896-00-0005), GLI2-1 (A-006468-14-0005), GLI2-2 (A-006468-15-0005), SMO-1 (A-005726-13-0005), SMO-2 (L-005726-00-0005), or a control pool (D-001810-10-05).

## Quantitative RT-PCR

Total RNA was isolated from cells using Trizol (Invitrogen, Cat# 15596). cDNA was synthesized using Proto Script II reverse transcription kit (NEB, Cat# E6560) and real-time PCR reactions were performed and analyzed using a Quant Studio 3 real-time PCR system (Applied Biosystems by Thermo Scientific) using primer sequences listed in Supplementary Data 3. Gene expression was normalized to that of *GAPDH*.

## Immunoblot analysis

Cells lysates were prepared in RIPA buffer (50 mM Tris pH 7.5, 1% Triton-X-100, 0.1% SDS, 0.5% deoxycholic acid, 10% glycerol) containing 1X cOmplete EDTA-free protease inhibitor cocktail (Roche, Cat# 11873580001) and 1 mM PMSF. Total cell lysates (30 μg) were subjected to SDS-PAGE and transferred to nitrocellulose membrane, which were blocked with 5% non-fat dry milk, and incubated with the following antibodies overnight at 4 °C: anti-C1GALT1 (clone F-31) (1:1000 dilution; Santa Cruz Biotechnology, Cat# sc-100745), anti-EWSR1 (1:1000 dilution; Sigma-Aldrich, Cat# HPA051771), anti-FLI1 (clone EPR4646) (1:2000 dilution; Abcam, Cat# ab133485), anti-GLI1 (1:1000 dilution; Cell Signaling Technology, Cat# 2553), anti-GLI2 (clone OTI1F9) (1:500 dilution; Abcam, Cat# ab187386), anti-SMO (1:1000 dilution; Abcam, Cat# ab236465), anti-PARP1 (clone F-2) (1:500 dilution; Santa Cruz Biotechnology, Cat# sc-8007), anti-CYP51A1 (clone N6-P2H5*G8) (1:1000 dilution; EMD Millipore, Cat# MABS1259), anti-CYP3A4 (clone HL3) (1:500 dilution; Santa Cruz Biotechnology, Cat# sc-53850), anti-pAKT (Ser473) (1:1000 dilution; Cell Signaling Technology, Cat# 9271 T), anti-AKT (1:1000 dilution; Cell Signaling Technology, Cat# 9272S), anti-pS6K(Thr389) (clone 108D2) (1:1000 dilution; Cell Signaling Technology, Cat#9234 T), anti-S6K (1:1000; Cell Signaling Technology, Cat# 9202S), anti-β-catenin (clone D10A8) (1:1000 dilution, Cell Signaling, Cat# 8480 T), anti-β-actin (clone AC-74) (1:2000 dilution; Sigma, Cat# A2228), anti-GAPDH (clone 6C5) (1:2000 dilution; Abcam, Cat# ab8245), and anti-FLAG (clone M2) (1:1000 dilution; Sigma, Cat# F1804). Blots were washed with TBST (1X Tris-Buffered Saline, 0.1% Tween) and incubated with rabbit IgG, HRP-linked whole Ab (1:5000 dilution; Cytiva, Cat# NA934V) or mouse IgG, HRP-linked whole Ab (1:4000 dilution; Cytiva, Cat# NA931V) for 1 h at room temperature. Following three washes with TBST, blots were incubated with SuperSignal West Pico Chemiluminescent Substrate (Thermo Scientific, Cat# 34080), and signals were detected by exposing them to X-ray film, which was developed using a Kodak X-OMAT 2000A film processor, with the exception of the cycloheximide chase experiment (Fig. 5d) and ubiquitination experiment (Fig. 5h), which were imaged using a ChemiDoc Imaging System (Bio-Rad) equipped with Image Lab software (version 5.0).

## Small molecule inhibitor treatment

Cells were treated with itraconazole (Sigma-Aldrich, Cat# 16657) at a concentration of 20–100 nM (as specified in the figure or legend) for

72 h (with the exception of Supplementary Fig. 11e where itraconazole was used at a concentration of 1–2.5 μM for 48 h), with cyclopamine (Selleckchem, Cat# S1146), GANT-61 (Selleckchem, Cat# S8075) or SAG (Selleckchem, Cat# S7779) at a concentration of 1-10 μM (as specified in the figure) for 48 h, or with sonidegib (Selleckchem, Cat# S2151) or vismodegib (Selleckchem, Cat# S1082) at a concentration of 1–20 μM (as specified in the figure) for 72 h (with the exception of Supplementary Fig. 11c-d where cells were treated with sonedegib and vismodegib for 48 h). Ketoconazole (Sigma, Cat# PHR1385) was used at a concentration of 10 μM for 72 h. MG132 (Selleckchem, Cat# S2619) was used at a concentration of 10 μM for 8 h, and bafilomycin A1 (Selleckchem, Cat# S1413) was used at a concentration of 100 nM for 8 h. Ac5GalNTGc (MedChemExpress, Cat# HY-160109) was used at a concentration of 50 μM or 100 μM for 48 h. DMSO (0.1% v/v) was used as a vehicle control.

## CRISPR gene editing
To generate C1GALT1 knockout cell lines, a C1GALT1 sgRNA (sgRNA-1, 5′-GCAACACTTTGTTACAACGC-3′, or sgRNA-2, 5′-GAGGTATTC-TAACTCATACA-3′ was cloned into lentiCRISPR v2 (gift from Feng Zhang; Addgene plasmid #52961). Lentivirus particles expressing the sgRNAs were packaged and used to infect A673 cells. After 48 h of infection, cells were selected with 2 μg/ml puromycin for 4 days. C1GALT1 knockout was confirmed by immunoblot analysis and Sanger sequencing. For the deletion of GLI1-binding motifs in the *EWSR1* promoter, two independent sgRNAs (5′-CGAGACCCTATCCCCGGTAA-3′ and 5′-AACAACTGCTGACTAATCCG-3′) were designed targeting the flanking regions of GLI1-binding sites P2 and P3 and were subcloned into the plasmid lentiCRISPRv2. Lentivirus particles expressing the sgRNAs were packaged and used to infect A673 cells. After 48 h of infection, cells were selected with 2 μg/ml puromycin for 4 days and were seeded in 96-well plates by limiting dilution to isolate single-cell clones, which were confirmed by Sanger sequencing.

## Chromatin immunoprecipitation assays
A673 or TC-71 cells (either untreated, treated with 80 nM itraconazole or DMSO, or expressing an NS or C1GALT1 shRNA) were crosslinked with 1% of formaldehyde. Fixed cells were suspended in ChIP lysis buffer (50 mM Tris–HCl pH 8.1, 1% SDS, 5 mM EDTA, 1X protease inhibitor cocktail) followed by sonication using a Bioruptor device (Diagenode). Sonicated chromatin was diluted 3-fold in ChIP dilution buffer (10 mM Tris–HCl pH 8.1, 100 mM NaCl, 1 mM EDTA, 0.01% SDS, 1% Triton X-100, 1X protease inhibitor cocktail) and incubated with 5 μg anti-GLI1 antibody (R&D Systems, Cat#: AF3324) or anti-GLI2 antibody (R&D Systems, Cat#: AF3526) overnight at 4 °C with constant rotation. In the ChIP experiment shown in Fig. 3i, normal goat IgG (R&D Systems, Cat# AB-108-C) or normal sheep IgG (R&D Systems, Cat# 5-001-A) was used as an isotype control antibody for experiments monitoring GLI1 or GLI2, respectively. Immunoprecipitated protein-DNA complexes were captured by incubating with magnetic Protein G beads (Promega, Cat# G7471) for 1 h at 4 °C with constant rotation. Beads were then washed with wash buffer (20 mM Tris–HCl pH 8.1, 500 mM NaCl, 2 mM EDTA, 0.1% SDS, 1% Triton X-100), LiCl buffer (20 mM Tris–HCl pH 8.0, 1 mM EDTA, 250 mM LiCl, 0.5% NP-40, 0.5% Na-deoxycholate) and TE buffer (10 mM Tris–HCl pH 8.0, 0.1 mM EDTA). Protein-DNA complexes were eluted by resuspending the beads in elution buffer (1% SDS, 0.1 M NaHCO$_3$) and incubating at room temperature for 30 min with rotation. Eluted samples were reverse cross-linked by incubating at 65 °C for 4 h. DNA was extracted using Tris-phenol and chloroform followed by ethanol precipitation. The extracted DNA was then subjected to qPCR using primers listed in Supplementary Data 3. A gene desert region on chromosome 2 (88,349–88,548) was used as a negative control. The relative enrichment at each target site was determined using the percent input method.

## GLI reporter assay
NIH 3T3 or A673 cells were co-transfected with 500 ng PGL4.26-10X-GLI1-Luc (generated by Genescript by inserting 10 GLI-binding sites [CCACCCA, corresponding to the sequence found in the *EWSR1* promoter] into the NheI/XhoI sites of vector pGL4.26-Luc (Promega, Cat# E844A)) and 50 ng of Renilla luciferase control plasmid (pGL4.72; Promega, Cat# E690A). After 48 h of co-transfection, cells were lysed with Passive Lysis buffer (Promega, Cat #E1941). Firefly luciferase and Renilla activities were measured using a GlowMax plate reader (Promega) using the Dual-Luciferase Reporter Assay System (Promega, Cat# E1910). Raw data (fluorescence readings) were collected from the plate reader and analyzed in Microsoft Excel (version 16.91).

## O-glycosylation assays
A673 or TC-71 whole-cell lysates (600 μg) were pre-treated with 100 U α2-3,6,8 neuraminidase (New England BioLabs, Cat# P0720S) in the presence or absence of 800 U O-glycosidase (New England BioLabs, Cat# P0733S) for 2 h at room temperature, and then incubated with PNA- or jacalin-conjugated agarose beads (Vector Laboratories, Cat# AL-1073 or Cat# AL-1153-10, respectively) overnight at 4 °C. The beads were washed three times with PBS and heated at 95 °C for 10 min to elute bound proteins. Eluted proteins were resolved by SDS-PAGE and immunoblotted using an anti-SMO antibody. For reciprocal pull-down experiments, SMO was immunoprecipitated using an anti-SMO antibody, and immunoprecipitates were subjected to SDS-PAGE and membrane transfer. The membranes were incubated in Carbo-Free Blocking Solution (Vector Laboratories, Cat# SP-5040-125) for 30 min at room temperature, followed by incubation in a solution containing 2 μg/ml biotinylated PNA (Vector Laboratories, Cat# B-1075-5) or Jacalin (Vector Laboratories, Cat # B-1155-5) for 30 min at room temperature. Membranes then were washed with TBST and incubated with HRP Streptavidin (1:6000 dilution; Biolegend, Cat# 405210) for 30 min at room temperature. Blots were developed as described above in the Immunoblot analysis.

## Co-immunoprecipitation
A673 cell lysate (300 μg) was incubated with 5 μg anti-SMO antibody, anti-C1GALT1 antibody, or control rabbit or mouse IgG antibody overnight at 4 °C. Immune complexes were captured with magnetic Protein G beads (Promega, Cat# G7471) for 1 h with gentle rocking at 4 °C. Beads were washed three times with lysis buffer. Immunoprecipitated proteins were eluted by resuspending the beads in 2X sample buffer (2% w/v SDS, 2 mM DTT, 4% v/v glycerol, 0.04 M Tris-HCL, pH 6.8 and 0.01% w/v Bromophenol blue) and heating at 95 °C for 5 min, followed by analysis by immunoblotting with an anti-SMO or anti-C1GALT1 antibody. Input lanes represent 10% of the total cell lysate used for immunoprecipitation.

## Cycloheximide chase assay
A673 cells were seeded at a density of $8 \times 10^5$ cells per well in a 6-well plate and grown in DMEM. On the day of the experiment, the medium was replaced with fresh DMEM, and cycloheximide (30 μg/ml) (ThermoFisher Scientific, Cat# J66004.XF) was added in the presence or absence of 100 nM ITZ for 3, 6, 9 or 12 h. DMSO-treated cells served as control. Cells were then harvested, and total cell lysates were prepared as described above and subjected to immunoblotting with an anti-SMO antibody. Blots were imaged using a ChemiDoc Imaging System (Bio-Rad) and the densitometry of the bands was performed using ImageJ software version 1.47 v (NIH)[63].

## Immunofluorescence assay
A673 cells stably expressing FLAG-SMO, FLAG-SMO(T55A), or FLAG-SMO(T500A) were treated with or without 10 μM MG132 for 8 h or 100 nM bafilomycin A1 for 8 h and fixed with 4% PFA (ThermoFisher Scientific, Cat# J61899.AK) for 10 min. Cells were washed with PBS and

permeabilized using buffer (0.5% Triton-X100 in PBS) for 10 min. After permeabilization, cells were incubated with blocking buffer (5% BSA in PBS) for 1 h, and then stained with anti-FLAG (clone M2) antibody (1:500 dilution; Sigma, Cat# F1804) overnight. The next day, cells were washed and incubated with F(ab')2-Goat anti-Mouse IgG (H + L) Cross-Adsorbed Secondary Antibody, Alexa Fluor 647 (1:1000 dilution, ThermoFisher Scientific, Cat# A21237), and mounted with Vectashield antifade mounting medium with DAPI (Vector Laboratories, Cat# H-1200-10). Stained cells were imaged at 63x magnification using a ZEISS LSM 900 confocal microscope equipped with a Plan-Apo chromat 63x/1.40 oil objective and processed using ZEISS ZEN imaging software (version 3.0).

## Ubiquitination assay

A673 cells stably expressing FLAG-tagged wild-type SMO, FLAG-SMO(T55A) or FLAG-SMO(T500A) were transiently transfected with a plasmid expressing HA-tagged ubiquitin (HA-Ub; a gift from Edward Yeh; Addgene plasmid #18712) and treated in the presence or absence of 10 $\mu$M MG132 for 8 h or 100 nM bafilomycin A1 for 8 h. Cell lysates (500 $\mu$g) were incubated with anti-HA magnetic beads (Sigma, Cat#-SAE0197) or anti-FLAG M2 magnetic beads (Sigma, Cat# M8823) overnight. The next day, the beads were pelleted using a magnetic separation rack and washed five times with 500 $\mu$l of 1X RIPA lysis buffer. Immunoprecipitated proteins were eluted by resuspending the beads in 2X sample buffer (2% w/v SDS, 2 mM DTT, 4% v/v glycerol, 0.04 M Tris–HCl, pH 6.8 and 0.01% w/v Bromophenol blue) and heating at 95 °C for 5 min. Eluted proteins were immunoblotted with an anti-FLAG (clone M2) (1:1500 dilution; Sigma, Cat# F1804) or anti-HA (clone 6E2) (1:1500 dilution; Cell Signaling Technology, Cat# 2367) antibody.

## Soft agar colony formation assay

A673 cells (1×10$^4$) stably expressed an NS, EWSR1, C1GALT1 or SMO shRNA; NIH 3T3 cells (5×10$^3$) stably expressed empty vector or EWSR1::FLI1 and a NS, C1GALT1 or SMO shRNA; NIH 3T3 cells (5×10$^3$) stably expressing empty vector, C1GALT1 or SMO; or TC-71 or TC-32 cells (1×10$^4$) following transduction with high titer NS, C1GALT1 or SMO shRNA lentivirus (see ShRNA and siRNA knockdown above) were resuspended in 2 ml of top agar (DMEM containing 0.3% Difco Noble agar (BD Biosciences, Cat# 214220)) pre-warmed to 40 °C. The cell suspension was layered onto 2 ml of set bottom agar (DMEM containing 0.7% Noble agar)) in a 6-well plate. The following day, 1 ml of DMEM was added and was changed every other day. After 3–4 weeks, visible colonies were counted at 20X magnification using a Zeiss Axio Observer 7 inverted microscope equipped with LD Plan-Neofluar 20x/0.40 objective and Axiocam 503 mono camera with ZEISS ZEN imaging software (version 2.3).

## Cell viability assays

TC-71, TC-32, and TC-106 cells were seeded in a 96-well plate (5×10$^3$ cells per well), and the following day ITZ was added at a concentration of 20–100 nM (as specified in the figure or legend) for 72 h. DMSO (0.1% v/v) was used as a control. For the experiment in Fig. 7c, TC-71 cells stably expressing C1GALT1 or EWSR1::FLI1 under the control of the CMV promoter were treated with 20–100 nM ITZ for 72 h. Cells were then incubated with PrestoBlue Cell Viability Reagent (Invitrogen, Cat# A13261) according to the manufacturer's protocol and fluorescence was quantified using a GlowMax Microplate Reader (Promega). For the C1GALT1 knockdown experiment in Supplementary Fig. 8a, TC-71 cells transduced with a lentivirus expressing an NS or C1GALT1 shRNA and selected with puromycin for 2 days were incubated with PrestoBlue Cell Viability Reagent. For the C1GALT1 knockdown experiments of Supplementary Fig. 8c, TC-71, TC-32 or TC-106 cells transduced with a high titer NS or C1GALT1 shRNA lentivirus (see shRNA and siRNA knockdown above) were stained with Trypan Blue

Solution, 0.4% (ThermoFisher Scientific, Cat# 15250061) and the number of viable (unstained) cells was determined using a Countess 3 FL Automated Cell Counter (Invitrogen, Cat# AMQAF2000) with Countess 3 FL software (version 1.1.505.429).

## Animal experiments

All mice used in this study were kept under specific pathogen-free conditions, housed in a 12-h light/dark cycle, and provided with ad libitum access to chow and water. Mice were randomly allocated to each group. No blinding was done as animal groups were identified by tagging and labeling the cages with the cells injected. An experiment was terminated, and mice were euthanized when the tumor volume reached ≥2000 mm$^3$, the maximal tumor size permitted by the UMass Chan Medical School IACUC. Sex was not considered in the study design; ES affects both males and females. and therefore both male and female mice were used.

A673 cells (5×10$^6$) stably expressed an NS, EWSR1, C1GALT1, or SMO shRNA; NIH 3T3 cells (5×10$^6$) stably expressed empty vector, SMO, C1GALT1, C1GALT1, and a SMO shRNA, or FLAG-SMO, FLAG-SMO(T55A) or FLAG-SMO(T500A); or TC-71 or TC-32 cells (5×10$^6$) following transduction with high titer NS or C1GALT1 shRNA lentivirus (see ShRNA and siRNA knockdown above) were resuspended in PBS and Matrigel (Corning Cat #356234) in a 1:1 ratio, and subcutaneously injected into the right flank of 6–7-week-old male or female NOD-*scid* IL2Rgamma$^{null}$ (NSG) mice (Jackson Laboratory, Strain# 005557). Tumor length (L) and width (W) were measured every 3-5 days using a caliper, and the tumor volume was calculated using the formula: [*L* x *W*$^2$]/2.

For ITZ treatment, TC-32, TC-71, or TC-106 cells (3×10$^6$), resuspended in PBS and Matrigel in a 1:1 ratio were subcutaneously injected in the right flank of NSG mice, when the tumor size reached ~80–100 mm$^3$, mice were administered with either vehicle (5% DMSO + 40% PEG300 + 5% Tween-80 + 45% Saline (PBS) or ITZ (100 mg/kg/day) by oral gavage for the duration of the experiment as indicated.

## Immunohistochemistry

All histology studies were carried out by the Morphology Core Facility at UMass Chan Medical School. For human samples, bone and cartilage tissue arrays containing formalin-fixed paraffin-embedded (FFPE) 5 $\mu$m sections of three ES patient tumor samples (Biomax Inc., Cat#T264a, [tissue ID: Lbn030013 and Lbn040062] and Cat#BO801 [tissue ID: Lbn060070]) and one sample each of normal bone (OriGene Cat# CS711511) and cartilage tissue (Biomax, Cat#T261b [tissue ID; Lca06N023]) were obtained, along with H&E stained slides. For the mouse tumor immunohistochemistry analyses, tumors were harvested, fixed in 10% buffered formalin phosphate (Fisher Scientific, Cat# SF1004) for 48 h, and then embedded in paraffin blocks. Blocks were sectioned at 5 $\mu$m thickness and mounted onto slides. For both human and mouse specimens, FFPE sections were first deparaffinized with xylene and rehydrated through graded alcohol washes, followed by a wash with PBS containing 1% horse serum (Vector Labs, Cat# S-200-20). Heat-induced antigen retrieval was performed using Retrievagen A working solution (pH 6.0) (BD Biosciences, Cat# 550524). Endogenous peroxidase activity was blocked with 3% hydrogen peroxide (Fisher Scientific, Cat #H324-500), followed by additional washes with PBS containing 1% horse serum. Specimens were then incubated with the following primary antibodies: anti-FLI1 (clone EPR4646) (1:50 dilution; Abcam, Cat# ab133485), anti-EWSR1 (1:50 dilution; Sigma-Aldrich, Cat# HPA051771), anti-Ki67 (1:250 dilution; Sino Biological, Cat# 100130-MM22), anti-C1GALT1 (clone F-31) (1:100 dilution, Santa Cruz Biotechnology, Cat# sc-100745), anti-SMO (1:25 dilution, Abcam, Cat# ab236465), anti-GLI1 (clone OTI2E1) (1:25 dilution, ThermoFisher Scientific, Cat# MA5-26639), and anti-FLAG (clone 9A3) (1:250

dilution; Cell Signaling Technology, Cat#8146 T). Following incubation with primary antibodies for 1 h, sections were incubated with Dako EnVision Dual-link System-HRP (DAKO, Cat# K4061) for 1 h, followed by two washes with PBS containing 1% BSA. Sections were then incubated with DAB Chromogen (DAKO, Cat# K3468) for 2 min and DAB Enhancer (DAKO, Cat# S1961) for 5 min. Slides were subsequently counterstained with hematoxylin, dehydrated, and mounted with a mounting medium. All stained sections were analyzed at 20X magnification using an Olympus BX40 microscope.

## Bioinformatic analyses
In silico GLI motif analysis for the *EWSR1* gene was done using the Broad Institute's Integrative Genomics Viewer (IGV) genome browser (https://software.broadinstitute.org/software/igv/) using the GLI consensus sequence GACCACCCA[34,35]. For Ewing sarcoma data analysis and Kaplan–Meier plots, datasets GSE17679 and GSE63157_core_transcriptom were downloaded from the R2 platform [R2:GenomicsAnalysis and Visualization Platform (http://r2.amc.nl)] and analyzed using a custom script available at https://github.com/JunhuiLi1017/kaplan_meier_GSE17679_GSE63157, deposited at Zenodo[64]. All samples in each dataset were grouped into two groups based on the value of the median, average, first quartile, last quartile, first_vs_last_quartile of the expression of *C1GALT1*. In the order of expression of *C1GALT1*, a cutoff with the highest value from the log-rank test was employed to split all samples into two groups (scan method). Analysis of overall survival between high and low *C1GALT1* expression groups was performed with the R package survival[65], and Kaplan–Meier curves were plotted with the R packages survminer[66] and patchwork[67]. *P*-values were calculated using the log-rank test.

## Statistics and reproducibility
Data for all qRT-PCR, ChIP, luciferase reporter, cell viability, and colony formation assays were collected from 3-5 independent biological replicates as stated in the figure legends (and each independent experiment included at least three technical replicates of the sample). Data are represented as mean ± SEM or ± SD as indicated in the figure legends. Statistical analysis of quantitative data was performed using GraphPad Prism (version 10.2.3). Differences between two groups were assessed using the two-tailed Mann–Whitney test, and differences between three or more groups were assessed using one-way analysis of variance (ANOVA) with post-hoc Dunnett's multiple comparisons test or one-way or two-way ANOVA with post-hoc Tukey's multiple comparisons tests, with the exception of Fig. 7f and Supplementary Fig. 12f, which were assessed using mixed-effect analysis with post-hoc Tukey's multiple comparisons tests. The statistical analysis used for each experiment is stated in the figure legends. *P* < 0.05 was considered significant.

For the main figures, the majority of the experiments involving immunoblot analysis, including the PNA lectin pull-down assay (Fig. 5a), co-immunoprecipitation assay (Fig. 5b), and ubiquitination assay (Fig. 5h), were conducted three independent times and the results from one representative experiment are shown. The cycloheximide chase assay of Fig. 5d was performed two independent times, and the immunoblot results from one representative experiment are shown. For the Supplementary Figs., the number of times the immunoblot experiment was performed is stated in the figure legend. For all IHC analyses, at least three sections of each tumor sample were analyzed, and the results from one representative section are shown.

## Reporting summary
Further information on research design is available in the Nature Portfolio Reporting Summary linked to this article.

## Data availability
The raw sequencing data from the CRISPR/Cas9 screen have been deposited at the NCBI Gene Expression Omnibus under accession number GSE287721. This paper analyzed previously published RNA-seq data deposited at the NCBI Gene Expression Omnibus under accession numbers GSE17679 and GSE63157. The remaining data supporting the findings of this study are available within the Article, Supplementary Information, or Source Data. Source data are provided in this paper.

## Code availability
For the CRISPR screen, statistical analysis and figure generation were performed using the *CRISPRscreen* R package, available at https://github.com/LihuaJulieZhu/CRISPRscreen, deposited at Zenodo[62]. For Ewing sarcoma data analysis and generation of Kaplan–Meier plots, datasets GSE17679 and GSE63157 were analyzed using a custom script available at https://github.com/JunhuiLi1017/kaplan_meier_GSE17679_GSE63157, deposited at Zenodo[64].

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

## Acknowledgements

This work is dedicated to the memory of Michael R. Green, a visionary scientist and exceptional mentor. His profound insight and enduring support were crucial to this work. We thank Junhao Mao for critically reading the manuscript; W. Rod Hardy for providing the tdTomato targeting plasmid; Ritesh P. Thakare for assistance with confocal microscopy; The Childhood Cancer Repository at Texas Children's Hospital for providing ES cell lines; Lynn Chamberlain for technical assistance; the University of Massachusetts Chan Medical School (UMCMS) RNAi Core Facility for providing shRNAs; the UMCMS Flow Cytometry Core Facility; the UMCMS Morphology Core for histology services; and the UMCMS Deep Sequencing Core Facility for high-throughput sequencing.

## Author contributions

Conceptualization, S.B., S.K.M., S.K.D., M.R.G.; Methodology, S.B., A.K.M., R.R., T.Y., A.A., J.T.Y., M.A.K., S.K.M.; Investigation, S.B., A.K.M., R.R., T.Y., A.A., S.K.M.; Formal analysis, S.B., S.K.D., S.K.M., M.R.G.; Bioinformatic analysis, J.L., L.J.Z.; Writing, reviewing, and editing, S.B., S.K.D., S.K.M., M.R.G.; Supervision, S.B., S.K.D., S.K.M., M.R.G. All authors read and approved the final manuscript.

## Competing interests

S.B. and M.R.G. are listed as inventors on a PCT application filed by the University of Massachusetts Chan Medical School on the identification of regulators (including C1GALT1) and inhibitors thereof (including itraconazole) of EWSR1::FLI1 expression for use as a method to treat Ewing sarcoma. The remaining authors declare no competing interests.
