## [Transparent Peer Review file · Nature Communications]

The O-glycosyltransferase C1GALT1 promotes EWSR1::FLI1 expression and is a therapeutic target for Ewing sarcoma

Corresponding Author: Dr Shahid Banday

Version 0:

Reviewer comments:

Reviewer #1

(Remarks to the Author)

In the proposed manuscript entitled “The O-glycosyltransferase C1GALT1 promotes EWS-FLI1 expression and is a therapeutic target for EwS”, Banday et al. provide an insightful look into the O-glycosyltransferase C1GALT1 and its association with expression of EWSR1::FLI1, the oncogenic driver of Ewing sarcoma (EwS), and a previously undescribed mechanistic link to the Hedgehog (Hh) pathway.

The paper is well-written and follows logically through the initial CRISPR/Cas9 screen for targets that promote expression of the EWSR1::FLI1 fusion, to validation of the O-glycosylation of Hh pathway targets, to the in vivo significance of targeting C1GALT1/Hh pathway in EwS xenografts. The experiments are thorough, examining the mRNA and protein level of the targets, validating with genetic and pharmacological means of inhibiting, and testing on multiple EwS lines and xenografts. The paper is of high quality, relevance to the field and novelty and is generally recommended for publication. However, there are some aspects that the authors should address before publication:

Major aspects:

- the micrographs for OS (Ext. Data Fig 5) appears to show fibroadipose tissue. As a pathologist, this reviewer cannot recognize tumor cells on the provided images. The authors are advised to critically check these figures. Eventually new tissue sections should be taken and stained.
- qPCR and ChIP graphs are shown as one run with the error being that of the technical replicates. This does not allow readers to assess the true biological variance of the effects – it is recommended to publish results of the true independent runs and the error to reflect the biological (not technical) error.
- n=3 is a very small number of mice per group and may be underpowered to see true variation in response.
- The authors should address/discuss whether the used doses for itraconazole in their in vitro and in vivo experiments would correspond to clinically achievable doses as they suggest that these drugs could be employed to treat EwS.
- The authors declare in the statistics section that they have employed the student's t-test. Yet, most experiments were done n=3 times. It should be noted that the t-test as a parametric test can only be applied if normal distribution of the data can be demonstrated or mathematically be proven, which is not possible in case of n=3. Thus, application of the student's t-test is obsolete, and the authors should employ (if they desire to do statistics) a non-parametric test, such as the Mann-Whitney test. This, however, would require at least n=4 replicates. Hence, the authors are advised to either omit inappropriate statistics or to use the appropriate statistics based on a sufficiently powered number of independent replicates. In the current case, this would mean to add independent biological replicates for key experiments of the paper.

Minor aspects:

- The authors should consider including negative control loci for the ChIP assay with GLI and the fusion
- This reviewer recommends updating the nomenclature for the fusion from – to ::, i.e. EWSR1::FLI1 following the new official HUGO nomenclature
- In the immunoblot images, EWS-FLI1 is stated as the antibody target but the antibody listed in the method is only FLI1, not the fusion. As this can be misleading, recommend changing immunoblot labels to FLI1. Also, it is recommended to provide an indication of the observed molecular weights of the corresponding bands for all immunoblot images.
- Citation should be provided for using ImageJ.
- The authors describe at some point the usage of the TC-106 cell line, which is driven by EWSR1::ERG. This is very interesting as these data suggest that their findings are not restricted to the type of EWSR1::ETS fusion (which is somewhat expected since the Hh pathway regulates these fusions via the common EWSR1 promoter). The authors may wish to

explicitly emphasize this aspect.

Reviewer #2

(Remarks to the Author)

Bandy et al present data that identifies C1GALT1 as a regulator of EWS-FLI1 in Ewing Sarcoma (ES) cell line from a CRISPR screen. They further go on to demonstrate that C1GALT1 binds to SMO and possibly glycosylates SMO, leading to its stabilization. SMO stabilization presumably activates the GLI transcription factors and the manuscript demonstrates that GLI1 binds to the promoter and intron 2 regions of EWS-FLI1 to promote its transcription. Itraconazole (ITZ) is known to inhibit C1GALT1. The authors demonstrate that ITZ has potent effects on EWS-FLI1 expression and tumor cell growth in culture and in vivo. The authors further show that high expression of C1GALT1 correlates with poor survival in ES patients.

The connection C1GALT1, SMO glycosylation and stability to drive EWS-FLI1 is an interesting and novel finding. The sensitivity of ES cells to ITZ is striking, leading to tumor regression in vivo, which is rarely seen. The data presented are solid but as noted below, other experiments may more directly test the conclusions and model of the manuscript. Also, there are some major concerns. The two most concerning issues are (1) the majority of the mechanistic studies have been performed in EWS-FLI1 independent A673 cell lines which are not reflective of the majority of ES patients and thus loses significant clinical relevancy and (2) the very high sensitivity of ES cells to ITZ but not to more potent SMO inhibitors raises the question of whether the ES cells are truly driven by SMO activation and whether ITZ's potent effects are due to other mechanisms besides the C1GALT1-SMO axis. These concerns are noted in greater detail below with other significant concerns in the order of importance.

1. The CRISPR screen and all of the mechanistic studies were performed in one ES line, A673, which does not depend on EWS-FLI1 for survival and growth. While this is beneficial for identification of regulators of EWS-FLI1, such as C1GALT1, it does not demonstrate that EWS-FLI1 dependent ES cells require C1GALT1 and SMO to regulate EWS-FLI1 expression, its tumor growth in culture or in vivo. This would be more clinically relevant for Ewing Sarcoma therapy. ITZ was used in TC-32 cells to test for EWS-FLI1 mRNA expression in Ext. Data Fig 7a,b. However, ITZ has many effects beyond SMO inhibition, including as noted in the manuscript, on C1GALT1 and EWS-FLI1. Therefore, the mechanistic experiments to demonstrate the connection between C1GALT1, SMO and EWS-FLI1 should also be performed in at least one or two ES cell lines that depend on EWS-FLI1 as was done for A673.

2. A673 cells seem to be unusually sensitive to ITZ, beyond what would be expected by its inhibition of SMO. The ITZ IC50 of SMO inhibition in NIH-3T3 cells is roughly 800 nM. Cyclopamine has an IC50 of SMO inhibition of 200-300 nM in NIH-3T3 cells. Fig. 2c shows that cyclopamine IC50 of EWS-FLI1 expression is 5 micromolar, ~80x less potent than ITZ (Fig. 1e). ITZ is a very promiscuous drug with numerous targets and there is a concern here that inhibition of other targets besides C1GALT1-SMO may be causing such potent responses. GLI1 and PTCH1 mRNA expression, as a readout of Hh pathway activation, should be assayed here with increasing ITZ concentration to better assess if ITZ's effect is through SMO or through other mechanisms.

3. As a corollary to the previous comment, there seems to be an incongruity with regards to TC-71. Similar to A673, TC-71 seems to be unusually sensitive to itraconazole (Fig. 5) whereas the LD50 for the SMO inhibitors sonidegib and vismodegib is ~20 micromolar (Ext. Fig. 7c), almost 1000x their IC50 for SMO inhibition in NIH-3T3 cells. If the primary effect of ITZ in TC-71 cells is through C1GALT1 and SMO, then the cells should also be much more sensitive to sonidegib and vismodegib. These SMO antagonists are ~50x more potent than ITZ to inhibit SMO. These results raise significant questions on whether TC-71 cells use a SMO-dependent mechanism for survival, presumably through EWS-FLI1 (since expression of EWS-FLI1 mRNA with increasing concentrations of sonidegib and vismodegib is not shown). Experiments that show levels of PTCH1 and GLI1 mRNA expression (as reporters of Hh pathway activity) with increasing doses of ITZ, sonidegib and vismodegib used in the viability studies would be helpful to distinguish between the effects mediated by SMO inhibition versus inhibition of other targets that ITZ may be affecting. mRNA expression of EWS-FLI1 with increasing doses of sonidegib and vismodegib would also be helpful.

4. The model of this manuscript is that C1GALT1 stabilizes SMO through glycosylation and that stabilization activates SMO that in turn leads to GLI1 activation and EWS-FLI1 expression. Overexpression of SMO in Hh ligand responsive NIH-3T3 cells without any Hh ligands induce only a modest activation of GLI1 activity. Thus, it's unclear how much SMO driven Hh pathway activity is in the ES cells. An immunoblot of A673 cells and an EWS-FLI1 dependent cell line demonstrating loss of GLI1 and EWS-FLI1 expression treated with and without O-glycosylase would give more direct evidence that it is the stabilization of SMO in the absence of Hh ligands that leads to EWS-FLI1 expression.

5. In the O-glycosylation studies of Fig. 3 f-j, IP of PNA lectin with IB of SMO in the context of C1GALT1 knock down or knock out in A673 cells and a EWS-FLI1 dependent ES cells would more directly demonstrate that C1GALT1 is required for SMO glycosylation.

6. The authors test GLI1 as the only transcription factor downstream of SMO. However, GLI2 is the primary effector of SMO activity and GLI1 is a target gene of GLI2 activity. GLI1 transcription is often used as a readout of Hh pathway activity because it is a target gene of GLI2. Thus, Fig 2b should also include knock-downs for GLI2 with readouts for GLI1 and EWS-FLI1 to make the connection that SMO regulates EWS-FLI1.

7. Similarly, for the ChIP analyses, the GLI consensus binding sequence is the same for both GLI1 and GLI2. Thus, presumably, GLI2 also binds to the promoter and intron 2 regions of EWS-FLI1 to activate its transcription. Demonstration that GLI2 binds to the promoter and intron 2 regions of EWS-FLI1 would further solidify the conclusions of this figure although not absolutely necessary.

8. The western blot of NIH-3T3 cells in Fig. 3d does not show any basal level of SMO with DMSO treatment. This is odd as NIH-3T3 cells respond to SHH and IHH ligands and thus, there must be some baseline levels of SMO to initiate the pathway. Furthermore, since SAG binds to and activates SMO, there must be some baseline level of SMO.

9. Page 5 – The description of Hh signaling pathway needs amendments. GLI2, not GLI1, is the primary effector of the pathway after SMO activation. GLI1 is a target gene of GLI2 and acts as an amplifier of the pathway. Thus, GLI2 knock out mice are embryonically lethal whereas GLI1 knockout mice show no developmental defects and as far as we know, is not required for development. Also, SUFU sequesters GLI1/2 from the nucleus but not in the primary cilia. Without activation of SMO, GLI1/2 are not found in the primary cilia, which is implied in the paragraph.

Reviewer #3

(Remarks to the Author)

Ewing sarcoma (ES) is an aggressive pediatric bone cancer driven by EWS-FLI1, which is an attractive drug target. Using CRISPR/Cas9 screening, the authors identified C1GALT1 as a critical factor to promote EWS-FLI1 expression. Mechanistically, the authors find that C1GALT1 O-glycosylates and stabilizes SMO to enhance EWS-FLI1 expression and signaling, leading to ES growth. Itraconazole, a known C1GALT1 inhibitor, reduces EWS-FLI1 and suppresses ES tumor growth in mice. Although this finding is interesting, there are still several concerns.

Major concerns

1. In a panel of ES cell lines, the evidence showing that C1GALT1 knockdown or knockout decreases SMO protein levels was only demonstrated in A673 cells. Deng et al (Cell Death and Disease, 2020, 11:539) showed that itraconazole did not decrease SMO protein levels in SW480 and HCT116 colon cancer cell lines. In addition, Hu et al (Journal of Experimental & Clinical Cancer Research, 2017, 36:50) showed that itraconazole did not affect SMO levels in MKN45 and AGS gastric cancer cells. However, itraconazole still inhibits the hedgehog signaling pathway. These results suggest that itraconazole can exert its inhibitory effect without alteration of SMO protein levels.
2. The mechanism by which C1GALT1 O-glycosylates SMO to stabilize SMO protein remains elusive. The authors showed that FLAG-SMO (T55A) and (T500A) mutants cannot be stably expressed in A673 cells. In C1GALT1 deficient cells, proteins still have Gal-GalNAc (T antigen) or sialyl T expression on all O-glycosites, which is quite different from the mutants with a complete loss of the O-glycan at certain O-glycosites. Moreover, O-glycosites on the proteins in different cell types may be different. To identify the correct O-glycosites on SMO in Ewing sarcoma, authors need to perform mass spectrometry using SMO from ES cell lines or ES tumors. Overall, although C1GALT1 can O-glycosylate SMO, whether C1GALT1-mediated glycosylation can stabilize SMO protein stability remains unclear.
3. In ES clinical samples, the quality for C1GALT1 immunohistochemistry is poor. Moreover, there is deficiency in the correlation between C1GALT1 expression and SMO.
4. The effect of C1GALT1 overexpression and knockdown (or knockout) on tumor growth should be done using multiple ES cell lines. Moreover, the effect of itraconazole on tumor growth using C1GALT1 knockdown or knockout cells should be analyzed. This approach will provide insights into whether itraconazole predominantly exerts its effects by inhibiting C1GALT1 or through alternative pathways not explored in the current study.
5. C1GALT1 has been known to regulate several integrins and receptor tyrosine kinases. How can the authors rule out the possibility that C1GALT1 regulate ES tumor growth through the regulation of these proteins.

Specific comments:

Figure 1. Molecular weight markers should be added to all western blots.

Figure 3.

In 3d, SMO, GLI, and ACTB in a mouse cell line should be labeled with Smo, Gli, and Actb, respectively. The relative changes in Smo and Gli1 using qRT-PCR analysis are not comparable to those observed on western blots.

In 3f, the molecular weight of SMO was decreased after O-glycanase treatment; however, PNA pulldown SMO did not show the change in molecular weight. In addition to PNA, the authors need to use VVA and jacalin to perform pull-down assays. Moreover, to validate the glycan changes, immunoprecipitation of SMO and then immunoblotting with PNA and VVA also should be done.

In 3j, SMO degradation could be dependent on both proteasome and lysosome, the authors also need to add a lysosome inhibitor and assess the mutant SMO levels. In addition, the mRNA levels of mutant SMO need to be analyzed using qRT-PCR. Ubiquitination of SMO should be checked. To monitor and validate the degradation of SMO mutants, it is recommended to perform immunofluorescence microscopy specifically for FLAG-SMO (T55A) and FLAG-SMO (T5800A) in cells treated with or without proteasome and lysosome inhibitors. This additional experiment will provide valuable insights into the subcellular localization and stability of the SMO mutants under different conditions, helping to elucidate the mechanisms involved in their degradation.

The experiments presented in Figure 3 are crucial for gaining insights into the underlying mechanism. To enhance the robustness and generalizability of the findings, it is strongly recommended to conduct experiments in panels 3f and 3j using multiple ES cell lines. Additionally, considering the space constraints in the main figures, the data from these experiments involving multiple cell lines can be appropriately presented in the extended data section.

Figure 4.

In 4a, the C1GALT1 staining should be localized in the Golgi rather than being observed throughout the entire cell. Additionally, it will be of importance to stain SMO, EWSFL1, and GLI1, and then analyze the correlation of C1GALT1 expression with the expression of these proteins in ES clinical samples.

In 4f, it is necessary to transfect FLAG-SMO (T55A) and FLAG-SMO (T500A) into NIH3T3 cells and analyze their effect on tumor growth. Additionally, to establish a more causal relationship, it is recommended to knock down the endogenous Smo in C1GALT1 overexpressing NIH3T3 cells and then perform tumor xenograft assays using these cell lines.

Figure 5.

In addition to A673, it is necessary to analyze the effect of itraconazole on C1GALT1 levels and SMO mRNA and protein levels in all EC cell lines used in this study, including SK-N-MC, TC-71, TC-106, and TC-32.

In 5e, C1GALT1 knockdown TC-71 cells should be included in the animal model. Moreover, the effect of itraconazole treatment on the tumor growth of C1GALT1 knockdown TC-71 cells need to be assessed. The tumors should be harvested to analyze C1GALT1, SMO, and FLI1 expression using immunohistochemistry.

Figure 6. The sugars are expected to be located in the extracellular region of SMO. However, it is crucial to note that the authors currently only possess evidence indicating the presence of disaccharides Gal-GalNAc, as determined by PNA lectin binding, instead of pentasaccharides.

Extended Data Figure 3.

c. The label C1GALTC1 should be corrected to C1GALT1C1.

Extended Data Figure 5.

a. SMO expression should also be analyzed in all cell lines.

b. C1GALT1 staining is not localized in the Golgi and the quality of tumor samples is poor. Nothing can be concluded from these images.

Extended Data Figure 7.

The animal model used in this study lacks the specific manipulations of C1GALT1, such as knockdown or knockout.

Incorporating C1GALT1 knockdown or knockout in the animal model is essential for a more targeted investigation into the role of C1GALT1 in the observed outcomes. This experimental modification will provide a clearer understanding of the direct contribution of C1GALT1 to the studied processes in the context of the animal model.

Extended Data Figure 8.

All representative western blots should be shown. This result is recommended to be described in the Result section instead of Discussion section.

In Discussion section

The authors should consider and discuss the possibility that C1GALT1 may exert its functional effects through other protein substrates such as integrins and receptor tyrosine kinases (RTKs), which are well known C1GALT1 substrates.

Other minor concerns

Line 104: References should include the original papers instead of one review article.

Line 230 and 232: six instead of seven ES samples.

Line 330: For pancreatic cancer, Oncogene. 2021 Feb;40:1242-1254 should be cited.

Version 1:

Reviewer comments:

Reviewer #1

(Remarks to the Author)

The authors have significantly improved the manuscript and the paper became much stronger.

There are only two remaining minor concerns of my previous concerns that should be addressed:

1) While I agree that bone and cartilage may be a better control than osteosarcoma (Fig. 2c), the new image on IHC for C1GALT1 in bone does not show bone like in the corresponding H&E image, but univacuolar fat tissue. The fat tissue should be replaced by bone.

2) I commend the authors for providing better statistics and more replicates for important experiments. Yet, I disagree that inappropriate t-tests should be employed in n=3 experiments even though this practice has been done in other papers published in the same journal or other high-impact journals. In short: inappropriate statistics cannot be justified by the mistakes of others. Science is not helped by perpetuating inappropriate statistics. In my opinion, statistics should be either appropriate or not provided. The authors could opt in for the latter option as in case of strong and consistently observed differences between groups, analytical statistics is not necessary.

Reviewer #2

(Remarks to the Author)

The authors have updated the manuscript with significant new data to address the concerns of previous comments by me and the other reviewers. The manuscript is much improved. However, I still have some major concerns and a minor one that need to be addressed.

1. The presented data, particularly those of SMO shRNA data from Fig. 6 and Supp Fig 7 and SMO glycosylation mutations in Supp Fig 6, suggest that Ewing Sarcoma cells depend on continuous SMO stabilization by C1GALT1 for growth and proliferation, presumably through transcription of FLI1 fusion proteins by activated GLI1/2 since GLI1/2 binds to promoter regions of EWS-FLI1 (Fig 3, 4). As noted in the model Fig 8a and also previous literature, EWS-FLI1 then induces further transcription of GLI1. Given this model, a circular loop of upregulation has been induced between EWS-FLI1 and GLI1 such that this loop should be independent of SMO. Given the model, how can EWS-FLI1 cells still depend on SMO and C1GALT1 esp. if the primary target of C1GALT1 is SMO in ES cells? The reciprocal activation loop between EWS-FLI1 and GLI1 must stop in order for the ES cells to continue to depend on C1GALT1 and SMO. Or perhaps EWS-FLI1 does not induce further GLI1 expression in these cells and thus require GLI1/2 activation by SMO to maintain ESW-FLI1 expression. Data are needed to reconcile this seeming contradiction as currently presented.

2. The authors posit that C1GALT1 glycosylation of SMO may be the reason for the insensitivity of ES cells to SMO inhibition by standard SMO inhibitors, cyclopamine, vismodegib and sonidegib, that share the same binding SMO binding pocket whereas ITZ is potent due to its ability to inhibit C1GALT1. Thus, ITZ may be more potent than SMO inhibitors in ES cells than in NIH-3T3 cells where the SMO inhibitors are more potent than ITZ. This hypothesis should be easily testable with the reagents that the authors have already presented. Overexpression of C1GALT1 in NIH-3T3 cells as in Fig. 6E should result in increased glycosylation and protein stabilization of SMO and thus GLI1. The induction of GLI1 protein expression and GLI1 mRNA transcription in the context of C1GALT1 overexpression should be relatively resistant to cyclopamine, vismodegib, sonidegib and perhaps other drugs that bind in the same SMO pocket but sensitive to ITZ. This is important as it would establish upregulation and glycosylation of SMO as a potential new resistance mechanism to the FDA-approved vismodegib/sonidegib and possibly glasdegib as well.

3. In response to my previous comment, the authors note that PNA lectin study with C1GALT1 knock down was difficult due to resultant SMO instability and therefore, used ITZ as a C1GALT1 inhibitor to demonstrate loss of SMO glycosylation. However, as noted previously, ITZ has numerous other effects including glycosylation of other proteins and thus, these new results, while suggestive, are still less definitive. If SMO stability is an issue, then treating cells with proteasome and lysosome inhibitors with C1GALT1 shRNA (as was done in Fig. 5e, f) should solve the SMO instability issue and allow for PNA lectin pull down.

4. In line 360, the authors note that Hh pathway activation drives other cancers such as ovarian, pancreatic, and lung cancers. However, this definitive statement is incorrect. Canonical Hh signaling pathway in tumors of epithelia derived from the endoderm (e.g. lung, pancreas, colon, prostate, bladder) have been shown to be tumor suppressive through Hh pathway activation in stroma and Hh pathway inhibition, genetically and pharmacologically, accelerated tumor growth. These preclinical results are consistent with the negative human clinical trials of SMO inhibitors in these tumor types. Thus, antagonizing the Hh pathway (at least with therapeutic SMO inhibitors) for these tumor types has been abandoned. Whether canonical Hh pathway activation in these tumors does indeed drive tumor growth is controversial in light of these data demonstrating tumor suppression by paracrine Hh pathway. Perhaps, as the authors note, subsets of these cancers with high C1GALT1 may be amenable to SMO inhibition by C1GALT1 inhibitors esp. if these inhibitors do not lead to stromal SMO antagonism. However, this would not explain the failure of SMO inhibitors in clinical trials of colon, pancreas and other solid tumors without Hh pathway mutations unless the vast majority of these tumors had overexpression of C1GALT1 or other reasons to cause SMO to be resistant to vismodegib and sonidegib.

4. Minor: Line 147-148: Activation of SMO indeed induces GLI1/2 activation. However, SMO activation suppresses GLI3R, not activate it as is implied in the current text.

Reviewer #3

(Remarks to the Author)

The revised manuscript has been significantly improved. However, there are still some concerns that should be resolved.

The immunohistochemistry (IHC) of C1GALT1 in the whole manuscript did not show specific staining signals. This is not acceptable. These IHC data include Fig. 2c, 6c, 6g, 7e, 7g, Supplementary Fig. 6f, Supplementary Fig. 11c and 11d. In addition, the IHC of FLAG in the Supplementary Fig. 6f indicates nuclear staining patterns as hematoxylin counterstain, which is not the expected localization of FLAG-SMO, FLAG-SMO(T55A), and FLAG-SMO(T500A).

Fig. 6f: It is necessary to show C1GALT1, SMO, and GLI1 levels in the stable lines of NIH3T3 cells in the same western blot because the IHC data are not convincing. There are four groups in Fig. 6b and 6f. However, only three groups of IHC data were shown in Fig. 6c and 6g.

In several figures, such as Fig. 4a, 4b, 4f, and Supplementary Fig. 8e, C1GALT1 or ITZ can significantly regulate SMO

mRNA levels. From these results, it appears that most of SMO expression is determined by its transcriptional regulation. However, in the Fig. 5e (left panel), either a proteasome inhibitor or lysosome inhibitor can almost completely reverse the SMO expression to the normal level. Do the authors have explanations for this phenomenon? Which is the major mechanism by which C1GALT1 regulates SMO levels?

Fig. 8: In the left panel, the multiple transmembrane domains of SMO should be located on the Golgi membrane instead inside the Golgi.

Version 2:

Reviewer comments:

Reviewer #1

(Remarks to the Author)

The authors have adequately addressed all my concerns.

Reviewer #2

(Remarks to the Author)

The authors have responded well to my previous comments. I have one further suggestion that should be added to the Discussion section as noted below. Otherwise, the manuscript is good to be published.

Comment:

In response to my concern that the previous model did not explain how SMO can affect the positive loop between GLI1 and EWS-FLI1, the authors noted in their rebuttal that SUFU must be inactivated by SMO in order for the GLI transcription factors to be translocated to the nucleus. However, without experimental evidence in EWS-FLI1 cells, this explanation is unsatisfactory. GLI1/GLI1 amplifications are a known mechanisms for sonidegib and vismodegib resistance (Buonamici et al., *Sci Transl Med*, 2010; Sharpe et al., *Cancer Cell*, 2015; Atwood et al., *Cancer Cell*, 2015). aPKC- λ is another vismodegib resistance mechanism through phosphorylation and activation of GLI1 (Atwood et al., *Nature*, 2013). In both of these scenarios, according to the reasoning presented in the rebuttal, overexpression of GLI1/2 and, in particular, activation of physiological levels of GLI1 by aPKC- λ should not be able to overcome SMO inhibitors since SMO is inactivated by the SMO antagonists and SUFU should still active to sequester GLI1/2 from the nucleus.

At this point, I believe it would be more accurate to note in the Discussion that the reason for the dependency of EWS-FLI1 cells on SMO activation, despite the positive feedback loop between GLI1 and EWS-FLI1, is not clear and may be due to the inactivation of SUFU to allow translocation of the GLI transcription factors to the nucleus given the previously noted examples of resistance mechanisms of SMO antagonists.

Reviewer #3

(Remarks to the Author)

In the revised figures, including Figures 2c, 6c, 6g, 7e, and 7g, as well as Supplementary Figures 6g, 12c, and 12d, the IHC staining for C1GALT1 did not display a specific Golgi staining pattern. Additionally, the background signals vary considerably within the same image data set.

Point-by-Point Response to Reviewers' Comments

Reviewer #1 (Remarks to the Author):

In the proposed manuscript entitled “The O-glycosyltransferase C1GALT1 promotes EWS-FLI1 expression and is a therapeutic target for EwS”, Banday et al. provide an insightful look into the O-glycosyltransferase C1GALT1 and its association with expression of EWSR1::FLI1, the oncogenic driver of Ewing sarcoma (EwS), and a previously undescribed mechanistic link to the Hedgehog (Hh) pathway. The paper is well-written and follows logically through the initial CRISPR/Cas9 screen for targets that promote expression of the EWSR1::FLI1 fusion, to validation of the O-glycosylation of Hh pathway targets, to the in vivo significance of targeting C1GALT1/Hh pathway in EwS xenografts. The experiments are thorough, examining the mRNA and protein level of the targets, validating with genetic and pharmacological means of inhibiting, and testing on multiple EwS lines and xenografts. The paper is of high quality, relevance to the field and novelty and is generally recommended for publication. However, there are some aspects that the authors should address before publication.

Major aspects:

1. The micrographs for OS (Ext. Data Fig 5) appears to show fibroadipose tissue. As a pathologist, this reviewer cannot recognize tumor cells on the provided images. The authors are advised to critically check these figures. Eventually new tissue sections should be taken and stained.

In the original manuscript, we performed immunohistochemistry (IHC) staining for C1GALT1 in Ewing sarcoma (ES) patient tumor samples; in these experiments, osteosarcoma (OS) patient tumor samples were included as a negative control. In response to a related comment by Reviewer 3 (regarding the quality of the ES patient samples), we obtained high-quality, commercially available tumor tissue arrays containing malignant tumor samples from ES patients, as well as normal bone and cartilage samples. We feel that the normal bone and cartilage samples serve as a more appropriate control than OS patient samples for the IHC experiments. In the revised manuscript, we have therefore removed the OS patient samples and instead show normal bone and cartilage samples, which do not express C1GALT1 (Fig. 2c).

2. qPCR and ChIP graphs are shown as one run with the error being that of the technical replicates. This does not allow readers to assess the true biological variance of the effects – it is recommended to publish results of the true independent runs and the error to reflect the biological (not technical) error.

We thank the reviewer for alerting us to this error in the manuscript. All qRT-PCR and ChIP experiments shown in the original manuscript did, in fact, present data from three independent biological replicates, and each independent experiment included at least three technical replicates of the sample. In the revised manuscript, we have corrected the text in the Methods and figure legends to state that results are shown for biological replicates rather than technical replicates. In addition, we have revised all figures to show individual data points to illustrate the biological variance.

3. n=3 is a very small number of mice per group and may be underpowered to see true variation in response.

In the revised manuscript, we conducted additional tumor xenograft experiments to increase the sample size of mice to n=6 per group (for experiments with ES cell lines) and n=5 per group (for experiments with NIH 3T3 cells), which has increased the statistical power of these experiments.

4. The authors should address/discuss whether the used doses for itraconazole in their in vitro and in

vivo experiments would correspond to clinically achievable doses as they suggest that these drugs could be employed to treat EwS.

In the mouse xenograft experiments, we administered itraconazole (ITZ) at a dosage of 100 mg/kg/day via oral gavage. This dosage was selected based on its established anti-tumor efficacy and safety in preclinical mouse models (see, for example, Kim et al. 2010, *Cancer Cell* 17:388; Kim et al. 2013 *Cancer Cell* 23:23; Deng et al. 2020, *Cell Death Dis* 11:539). Notably, a previous study showed that this dosage results in serum levels of ITZ in mice that are comparable to those reported in patients undergoing high-dose (600-900 mg/day) ITZ anti-fungal therapy (Kim et al. 2010, *Cancer Cell* 17:388), indicating the dosage is clinically achievable. Moreover, the results from an open-label exploratory phase II clinical trial of oral ITZ in human basal cell carcinoma demonstrated that administration of ITZ at standard therapeutic doses ranging from 100 to 200 mg twice a day for one month reduced tumor proliferation and Hh pathway activity (Kim et al. 2014, *J Clin Oncol* 32:745), indicating that the dosage of ITZ used as an anti-fungal agent is effective for treating Hh-dependent cancer. We have revised the Discussion to mention these key points.

Given the low IC₅₀ of ITZ against ES cells in vitro, we hypothesize that ITZ could be effective against human ES tumors when administered according to standard treatment regimens or in combination with chemotherapy. However, this hypothesis remains to be tested until clinical trials of ITZ in ES are conducted.

5. The authors declare in the statistics section that they have employed the students t-test. Yet, most experiments were done n=3 times. It should be noted that the t-test as a parametric test can only be applied if normal distribution of the data can be demonstrated or mathematically be proven, which is not possible in case of n=3. Thus, application of the student's t-test is obsolete, and the authors should employ (if they desire to do statistics) a non-parametric test, such as the Mann-Whitney test. This, however, would require at least n=4 replicates. Hence, the authors are advised to either omit inappropriate statistics or to use the appropriate statistics based on a sufficiently powered number of independent replicates. In the current case, this would mean to add independent biological replicates for key experiments of the paper.

We thank the reviewer for bringing this issue to our attention and appreciate the concern regarding the use of the Student's t-test with a sample size of n=3. In response to the reviewer's comment, we have increased the number of independent biological replicates and performed statistical analysis using the Mann-Whitney test for several key experiments (Fig. 2e and Supplementary Fig. 11a, b, e).

For several of the remaining experiments in our manuscript where a t-test is used with a sample size of n=3, we note that the results of many of these experiments are corroborated by independent evidence. For example, the qRT-PCR analyses of Fig. 4b and 4f and Supplementary Fig. S12a are independently validated by accompanying immunoblot analyses confirming increased or decreased expression of the factor of interest. Moreover, the ChIP experiment of Fig. 4d, showing reduced binding of GLI transcription factors to *EWSR1* the upon C1GALT1 knockdown, is independently corroborated by a GLI-driven luciferase reporter assay (Fig. 4c).

Although we are aware that using a parametric t-test with a sample size of n=3 is not ideal, we would like to point out that is routinely done for these types of experiments in published studies, including those recently published in *Nature Communications*. For example, in Hughes et al. 2023 (*Nat Commun* 14:4357), a t-test was used in experiments with n=3 for qRT-PCR (see Figs. 1a, 5a, 5b) and soft agar assays (see Figs. 1j, 1l). In Wang et al. 2024 (*Nat Commun* 15:4914), a t-test was used in experiments with n=3 for qRT-PCR (see Figs. 4d, 4e, 6g, 7g, 8c, 9c), ChIP-qPCR (see Figs. 3e, 3g, 3h, 3m, 5h) and cell growth/viability assays (see Fig. 8d). In Chouhan et al. 2024 (*Nat Commun* 15:5629), a t-test was used in experiments with n=3 for qRT-PCR (see Figs. 4c, 4d, 4i, 4j, 8d, 8e) and ChIP-qPCR (see Figs. 3d, 8c, 8g). In Patterson et al. 2024 (*Nat Commun* 15:5809), a t-test was used in experiments with n=3 for qRT-PCR (see Figs. 1e, 2b, 2e, 2h, 3j, 3l, 4i, 5a, 5f, 8a), ChIP-qPCR (Fig. 3n), soft agar

assays (see Fig. 4e, 4h, 5e, 5j, 6d), cell growth/viability assays (see Fig. 4f, 4k, 5c, 5h, 6b, 8b) and luciferase reporter assays (Fig. 2c, 2f, 2i, 3b, 3d, 3g, 3h, 3i, 3k).

Minor aspects:

6. The authors should consider including negative control loci for the ChIP assay with GLI and the fusion

In the original manuscript, the ChIP assay did in fact, include a negative control locus (labeled “Control”), which is a gene desert region on chromosome 2. As expected, at this negative control locus, we did not detect enrichment of GLI1/2 binding. In response to the reviewer’s comment, we have labeled the ChIP graphs in the manuscript (Figs. 3i, 4d and 4e and Supplementary Figs. 8g, h) with “GD” (for “gene desert”) instead of “Control” for clarity and have described the coordinates of the gene desert region analyzed in the Methods section.

7. This reviewer recommends updating the nomenclature for the fusion from – to ::, i.e. EWSR1::FLI1 following the new official HUGO nomenclature

We thank the reviewer for bringing this recent nomenclature change to our attention. As suggested, we have updated the nomenclature for the fusion protein to EWSR1::FLI1 throughout the revised text and figures.

8. In the immunoblot images, EWS-FLI1 is stated as the antibody target but the antibody listed in the method is only FLI1, not the fusion. As this can be misleading, recommend changing immunoblot labels to FLI1. Also, it is recommended to provide an indication of the observed molecular weights of the corresponding bands for all immunoblot images.

As recommended, we have provided molecular weight marker information for all immunoblot images. However, we respectfully disagree with the reviewer's recommendation to change the immunoblot labels to FLI1. In theory, the FLI1 antibody would detect both the wild-type FLI1 protein (51 kDa) and the EWSR1::FLI1 fusion-protein (68 kDa). However, in the cell lines we use, we only detect the 68 kDa EWSR1::FLI1 fusion protein, which is consistent with a previous report showing wild-type FLI1 is not expressed in human ES cell lines (Smith et al. 2006, *Cancer Cell* 9:405). Therefore, we think it would be inaccurate to label the blots with “FLI1” when they clearly show the EWSR1::FLI1 fusion protein, not wild-type FLI1. Moreover, coupled with the newly added molecular weight marker information, we feel it could be potentially confusing to the reader if we label a blot “FLI1” but show a band corresponding to a molecular weight that is greater than FLI1. Finally, we note that labeling immunoblots with “EWSR1::FLI1” (or EWS-FLI1) is widely done in the field (see, for example, Seong et al. 2021, *Cancer Cell* 39:1262; Selvanathan et al. 2015, *PNAS* 112:E1307; Toretsky et al. 2006, *Cancer Res* 66:5574; Castellero et al. 2005, *Cancer Res* 65:8698; Chan et al. 2003, *B J Cancer* 88:137).

The use of an FLI1 antibody to detect EWSR1::FLI1 was mentioned at first use in the legend to Fig. 1c; prompted by the reviewer’s comment, we have clearly indicated an anti-FLI1 antibody was used in the legends for all relevant figure panels. We note that to detect EWSR1::ERG in TC-106 cells, we used an anti-EWSR1 antibody; we labeled these blots with “EWSR1” and indicated in the legend for all relevant panels that an anti-EWSR1 antibody was used (Supplementary Figs. 7d, 8b and 8f).

9. Citation should be provided for using ImageJ.

The citation for ImageJ is included in the revised manuscript.

10. The authors describe at some point the usage of the TC-106 cell line, which is driven by EWSR1::ERG. This is very interesting as these data suggest that their findings are not restricted to the

type of EWSR1::ETS fusion (which is somewhat expected since the Hh pathway regulates these fusions via the common EWSR1 promoter). The authors may wish to explicitly emphasize this aspect.

We thank the reviewer for this recommendation. As suggested, we have mentioned in the Discussion section of the revised manuscript that our findings are not restricted to EWSR1::FLI1, and that we find C1GALT1-mediated induction of Hh signaling also promotes expression of EWSR1::ERG.

Reviewer #2 (Remarks to the Author):

Bandy et al present data that identifies C1GALT1 as a regulator of EWS-FLI1 in Ewing Sarcoma (ES) cell line from a CRISPR screen. They further go on to demonstrate that C1GALT1 binds to SMO and possibly glycosylates SMO, leading to its stabilization. SMO stabilization presumably activates the GLI transcription factors and the manuscript demonstrates that GLI1 binds to the promoter and intron 2 regions of EWS-FLI1 to promote its transcription. Itraconazole (ITZ) is known to inhibit C1GALT1. The authors demonstrate that ITZ has potent effects on EWS-FLI1 expression and tumor cell growth in culture and in vivo. The authors further show that high expression of C1GALT1 correlates with poor survival in ES patients.

The connection C1GALT1, SMO glycosylation and stability to drive EWS-FLI1 is an interesting and novel finding. The sensitivity of ES cells to ITZ is striking, leading to tumor regression in vivo, which is rarely seen. The data presented are solid but as noted below, other experiments may more directly test the conclusions and model of the manuscript. Also, there are some major concerns. The two most concerning issues are (1) the majority of the mechanistic studies have been performed in EWS-FLI1 independent **A673 cell** lines which are not reflective of the majority of ES patients and thus loses significant clinical relevancy and (2) the very high sensitivity of ES cells to ITZ but not to more potent SMO inhibitors raises the question of whether the ES cells are truly driven by SMO activation and whether ITZ's potent effects are due to other mechanisms besides the C1GALT1-SMO axis. These concerns are noted in greater detail below with other significant concerns in the order of importance.

1. The CRISPR screen and all of the mechanistic studies were performed in one ES line, A673, which does not depend on EWS-FLI1 for survival and growth. While this is beneficial for identification of regulators of EWS-FLI1, such as C1GALT1, it does not demonstrate that EWS-FLI1 dependent ES cells require C1GALT1 and SMO to regulate EWS-FLI1 expression, its tumor growth in culture or in vivo. This would be more clinically relevant for Ewing Sarcoma therapy. ITZ was used in TC-32 cells to test for EWS-FLI1 mRNA expression in Ext. Data Fig 7a,b. However, ITZ has many effects beyond SMO inhibition, including as noted in the manuscript, on C1GALT1 and EWS-FLI1. Therefore, the mechanistic experiments to demonstrate the connection between C1GALT1, SMO and EWS-FLI1 should also be performed in at least one or two ES cell lines that depend on EWS-FLI1 as was done for A673.

In response to this comment, we performed several key mechanistic experiments in three additional ES cell lines: TC-32 and TC-71, which require *EWSR1::FLI1* for viability and proliferation in culture, and TC-106, which expresses another ES fusion gene, *EWSR1::ERG*, that shares the *EWSR1* promoter and is therefore expected to be transcriptionally regulated by the same mechanism as *EWSR1::FLI1*.

We first performed a series of experiments to confirm in these ES cell lines that *EWSR1::FLI1/ERG* expression is promoted by the same C1GALT1-mediated Hh signaling mechanism that we elucidated in A673 cells. We found that C1GALT1 was required for viability of these cell lines (Supplementary Fig. 8a), precluding us from deriving stable C1GALT1 knockdown derivatives, and therefore these experiments were performed by transducing cells with concentrated, high titer C1GALT1 shRNA lentivirus followed by short-term drug selection (12-16 hours) and recovery in the absence of drug (see Methods section for details); under these conditions, we observed efficient reduction of C1GALT1 levels (Supplementary Fig. 8b) without a marked decrease in cell viability (Supplementary Fig. 8c). Consistent with our results in A673 cells, we found that Hh signaling (i.e., SMO, GLI1 and GLI2) was required for expression of *EWSR1::FLI1/ERG* (Supplementary Fig. 7), and that knockdown of C1GALT1 reduced Hh signaling, as evidenced by reduced levels of SMO and GLI1 (Supplementary Fig. 8b). As expected, knockdown of C1GALT1 also reduced expression of *EWSR1::FLI1/ERG* (Supplementary Fig. 8b) and reduced binding of GLI1 and GLI2 to the *EWSR1* promoter and intron 2

regions (Supplementary Fig. 8g). Finally, C1GALT1 knockdown substantially reduced the ability of TC-71, TC-32 or TC-106 cells to form colonies in soft agar (Supplementary Fig. 9d) and tumors in mice (Fig. 7f and Supplementary Fig. 11f).

Consistent with our results in A673 cells, we also found that in EWSR1::FLI1/ERG-dependent ES cell lines, ITZ treatment reduced Hh signaling and EWSR1::FLI1/ERG expression (Supplementary Fig. 8d-f) and reduced binding of GLI1 and GLI2 to the *EWSR1* promoter (Supplementary Fig. 8h). Moreover, we showed that SMO was O-glycosylated in TC-71 cells (Supplementary Fig. 9a, b) and that SMO O-glycosylation was reduced by ITZ treatment (Supplementary Fig. 9b). Furthermore, in TC-71 cells O-glycosylation-defective SMO mutants had reduced stability (Supplementary Fig. 9c).

Collectively, these results confirm that C1GALT1 and Hh signaling (in particular, SMO) regulate EWSR1::FLI1/ERG expression in EWSR1::FLI1/ERG-dependent ES cell lines by the same mechanism that we deduced in A673 cells. Furthermore, the results demonstrate that C1GALT1 is required for soft agar colony formation and tumor growth of EWSR1::FLI1/ERG-dependent ES cell lines. We thank the reviewer for suggesting these experiments, the results of which have substantially enhanced the rigor and clinical relevance of our study.

2. A673 cells seem to be unusually sensitive to ITZ, beyond what would be expected by its inhibition of SMO. The ITZ IC₅₀ of SMO inhibition in NIH-3T3 cells is roughly 800 nM. Cyclopamine has an IC₅₀ of SMO inhibition of 200-300 nM in NIH-3T3 cells. Fig. 2c shows that cyclopamine IC₅₀ of EWS-FLI1 expression is 5 micromolar, ~80x less potent than ITZ (Fig. 1e). ITZ is a very promiscuous drug with numerous targets and there is a concern here that inhibition of other targets besides C1GALT1-SMO may be causing such potent responses. GLI1 and PTCH1 mRNA expression, as a readout of Hh pathway activation, should be assayed here with increasing ITZ concentration to better assess if ITZ's effect is through SMO or through other mechanisms.

In response to the reviewer's comment, we performed qRT-PCR to monitor mRNA levels of *GLI1* and *PTCH1* in A673 cells treated with increasing concentrations of ITZ. We found that treatment of A673 cells with ITZ resulted in a dose-dependent decrease in *GLI1* and *PTCH1* mRNA levels, confirming inhibition of the Hh pathway (Supplementary Fig. 4a). In these experiments, the ITZ IC₅₀ for *GLI1/PTCH1* inhibition was ~80 nM, consistent with what we observed for inhibition of *EWSR1::FLI1* expression (shown in Fig. 1e of the original manuscript and Fig. 2d of the revised manuscript).

To address the reviewer's concern that ITZ may be acting by inhibiting targets other than C1GALT1 and SMO, we performed two additional experiments. First, we treated A673 cells with increasing concentrations of ITZ (20 nM – 2.5 μM) and monitored the activity of other signaling pathways known to be inhibited by ITZ, including AKT/mTOR (Xu et al. 2010, *PNAS* 107:4764; Liu et al. 2014, *Autophagy* 10:1241; Tsubamoto et al. 2017, *Anticancer Res* 37:515; Liang et al. 2017, *Oncotarget* 8:28510) and Wnt (Liang et al. 2017, *Oncotarget* 8:28510; Ueda et al. 2017, *Anticancer Res* 37:3521). The results, presented in Supplementary Fig. 10e, show that at the highest dose of ITZ used in our experiments (100 nM), the levels of SMO and EWSR1::FLI1 were substantially reduced, but the levels of phosphorylated-AKT(Ser473) (a readout of AKT signaling), phosphorylated S6K(Thr389) (a readout of mTOR signaling) and β-catenin (a readout of Wnt signaling) were unaffected. Thus, at the relatively low concentration of ITZ used in our cell culture experiments, it does not inhibit other signaling pathways that we tested. Second, we determined whether ectopic expression of C1GALT1 could rescue TC-71 cells from ITZ-mediated reduction in cell viability. The results, presented in Fig. 7c, show that similar to ectopic expression of EWSR1::FLI1, overexpression of C1GALT1 also rescued TC-71 cell viability. The results of these experiments—together with the addition of several other new experiments showing that the effects we observe with ITZ in EWSR1::FLI1/ERG-dependent cell lines are recapitulated by knockdown of C1GALT1 (Fig. 7f and Supplementary Figs. 7, 8a, 8b, 8g, 9d and 11f; see response to comment #2 above), and that the effects of ITZ on tumor growth are not exacerbated

by knockdown of C1GALT1 (Fig. 7f and Supplementary Fig. 11f; see response to Reviewer 3 comment #4 below)—strongly suggest that in ES cells, ITZ exerts its effects by inhibiting the C1GALT1-SMO axis.

3. As a corollary to the previous comment, there seems to be an incongruity with regards to TC-71. Similar to A673, TC-71 seems to be unusually sensitive to itraconazole (Fig. 5) whereas the LD50 for the SMO inhibitors sonidegib and vismodegib is ~20 micromolar (Ext. Fig. 7c), almost 1000x their IC50 for SMO inhibition in NIH-3T3 cells. If the primary effect of ITZ in TC-71 cells is through C1GALT1 and SMO, then the cells should also be much more sensitive to sonidegib and vismodegib. These SMO antagonists are ~50x more potent than ITZ to inhibit SMO. These results raise significant questions on whether TC-71 cells use a SMO-dependent mechanism for survival, presumably through EWS-FLI1 (since expression of EWS-FLI1 mRNA with increasing concentrations of sonidegib and vismodegib is not shown). Experiments that show levels of PTCH1 and GLI1 mRNA expression (as reporters of Hh pathway activity) with increasing doses of ITZ, sonidegib and vismodegib used in the viability studies would be helpful to distinguish between the effects mediated by SMO inhibition versus inhibition of other targets that ITZ may be affecting. mRNA expression of EWS-FLI1 with increasing doses of sonidegib and vismodegib would also be helpful.

In response to the reviewer's comment, we performed additional qRT-PCR analyses to monitor *GLI1*, *PTCH1* and *EWSR1::FLI1* mRNA levels following treatment of TC-71 cells with increasing concentrations of sonidegib or vismodegib. We found that treatment of cells with the SMO inhibitors resulted in a dose-dependent decrease in *GLI1* and *PTCH1* mRNA levels, confirming inhibition of the Hh pathway, as well as a dose-dependent decrease in *EWSR1::FLI1* mRNA levels (Supplementary Fig. 10d), with kinetics that mirrored the decrease in ES cell viability (see Supplementary Fig. 10c). In conjunction with the ITZ experiments shown in Supplementary Fig. 8d and 8f (monitoring *PTCH1*, *GLI1* and *EWSR1::FLI1*), the results of these experiments reveal that in TC-71 cells the IC50 for ITZ for Hh inhibition is ~60-80 nM, whereas the IC50 for sonidegib or vismodegib for Hh inhibition is ~20 μM, confirming our finding with cyclopamine in A673 cells (Fig. 3e).

We note that the concentration of ITZ used in our cell culture experiments (20-100 nM) is ~10-1000 times less than what other published studies have used when monitoring the effects of ITZ on Hh signaling or viability in other cancer cell types. For example, ITZ concentrations of 1-20 μM were used to examine gastric cancer cell lines (Hu et al. 2017, *J Expt Clin Cancer Res* 36:50), 2.5-100 μM in colon cancer cell lines (Deng et al. 2020, *Cell Death and Disease* 11:539), and 0.13-60 μM in melanoma cell lines (Liang et al. 2017, *Oncotarget* 8:28510). Thus, we agree with the reviewer that ES cell lines appear to be sensitive to ITZ.

As noted by the reviewer, the reported IC50 for SMO inhibitors in NIH 3T3 cells is ~1000 times less than what we observe in ES cells, suggesting that ES cells are also unusually *insensitive* to conventional SMO inhibitors. Prompted by the reviewer's comments, we have raised this point in the Discussion section and mentioned the possibility that in ES cells, C1GALT1-induced O-glycosylation of SMO may block binding of conventional SMO inhibitors—such as cyclopamine, sonidegib and vismodegib—to the SMO drug-binding pocket, thus rendering ES cells relatively insensitive to these drugs.

4. The model of this manuscript is that C1GALT1 stabilizes SMO through glycosylation and that stabilization activates SMO that in turn leads to GLI1 activation and EWS-FLI1 expression. Overexpression of SMO in Hh ligand responsive NIH-3T3 cells without any Hh ligands induce only a modest activation of GLI1 activity. Thus, it's unclear how much SMO driven Hh pathway activity is in the ES cells. An immunoblot of A673 cells and an EWS-FLI1 dependent cell line demonstrating loss of GLI1 and EWS-FLI1 expression treated with and without O-glycosylase would give more direct

evidence that it is the stabilization of SMO in the absence of Hh ligands that leads to EWS-FLI1 expression.

We thank the reviewer for this suggestion. In the experiment shown in Fig. 5a, cell extracts [not intact cells] were treated with recombinant O-glycosidase. This enzyme is primarily used for deglycosylating glycoproteins in their denatured form and specifically removes O-linked disaccharides from glycoproteins. However, this enzyme is not typically designed for direct use in live cell cultures.

Based on the reviewer's suggestion, we performed an experiment in which A673 cells were treated with Ac5GalNTGc, a potent cell-permeable inhibitor of O-glycosylation. Ac5GalNTGc acts by preventing the addition of glycans to GalNAc (Tn antigen) (i.e., the step at which C1GALT1 acts), thereby blocking O-glycan biosynthesis. The results, presented in Supplementary Fig. 5b, show that Ac5GalNTGc treatment resulted in reduced O-glycosylation of SMO, which was accompanied by reduced levels of SMO protein, as well as GLI1 and EWSR1::FLI1. These results are consistent with the new results presented in Fig. 5c showing that ITZ treatment also reduces the levels of GLI1 and EWSR1::FLI1 with kinetics similar to reduced SMO O-glycosylation and reduced SMO protein levels. Collectively, these results bolster the idea that stabilization of SMO, in the absence of Hh ligands, promotes EWSR1::FLI1 expression.

5. In the O-glycosylation studies of Fig. 3 f-j, IP of PNA lectin with IB of SMO in the context of C1GALT1 knock down or knock out in A673 cells and a EWS-FLI1 dependent ES cells would more directly demonstrate that C1GALT1 is required for SMO glycosylation.

We acknowledge that published studies reporting the identification C1GALT1 targets have demonstrated that knockdown of C1GALT1 decreases binding of the protein of interest to PNA lectin (see, for example, the identification of MUC1 as a C1GALT1 target in breast cancer cells [Chou et al. 2015, *Oncotarget* 6:6123]). It is important to note that in these studies the total level of the protein of interest in whole cell lysates is unaffected by knockdown of C1GALT1. By contrast, as noted in our manuscript, we find that SMO levels are undetectable in C1GALT1 knockdown cells (see Fig. 4a), which of course would result in the apparent absence of SMO signal in a PNA lectin pull-down assay.

Nevertheless, in an attempt to address the reviewer's concern, we performed additional PNA lectin pull-down experiments following knockdown of C1GALT1 in A673 cells and TC-71 cells (an EWSR1::FLI1-dependent ES cell line). Rather than performing these experiments in stable C1GALT1 knockdown cell lines, we performed them by transducing cells with concentrated, high titer C1GALT1 shRNA lentivirus in the absence of drug selection, enabling us to monitor an early timepoint following C1GALT1 knockdown. We hoped that under these conditions, we would observe efficient reduction in C1GALT1 while retaining SMO at sufficient levels to be able to detect a notable reduction in SMO O-glycosylation as evidenced by reduced binding to PNA lectin (similar to what we observed upon ITZ treatment; see Fig. 5c). Unfortunately, we found that the reduction in SMO levels in whole cell lysates following C1GALT1 knockdown was comparable to the reduction in SMO signal observed in the PNA pull-down assay, and thus we are unable to conclude whether knockdown of C1GALT1 has an effect on SMO O-glycosylation. Because the results are uninterpretable with respect to the role of C1GALT1, we have not included them in the revised manuscript, but present them in below in Fig. A.

Prompted by the reviewer's comment, we have revised the manuscript to state more clearly that previous large-scale glycoproteomic studies have identified SMO as an O-GalNAc glycoprotein harboring glycans with a core 1 structure, indicating it is a C1GALT1 target.

Figure A. PNA lectin pull-down assay in C1GALT1 knockdown ES cells. Representative immunoblot showing the level of O-glycosylated SMO, detected by either immunoblot (IB) analysis for SMO in a PNA lectin pull-down (PD) assay or IB analysis for PNA in a SMO immunoprecipitate (IP), in A673 (left) or TC-71 (right) cell lysates following treatment with neuraminidase (to remove sialic acids that can hinder binding to PNA beads). The level of total SMO in whole cell lysate (input) is also shown. Cell lysates were prepared 72 hours following transduction of cells with a C1GALT1 shRNA lentivirus.

6. The authors test GLI1 as the only transcription factor downstream of SMO. However, GLI2 is the primary effector of SMO activity and GLI1 is a target gene of GLI2 activity. GLI1 transcription is often used as a readout of Hh pathway activity because it is a target gene of GLI2. Thus, Fig 2b should also include knock-downs for GLI2 with readouts for GLI1 and EWS-FLI1 to make the connection that SMO regulates EWS-FLI1.

As suggested, we performed the experiment shown in Fig. 2b (now Fig. 3b and c) in A673 cells and found that knockdown of GLI2 reduced EWSR1::FLI1 mRNA and protein levels (Fig. 3d). As expected, GLI1 protein levels were reduced in GLI2 knockdown cells. We note that similar results were obtained in TC-71, TC-32 and TC-106 cells (Supplementary Fig. 7c and d). These results bolster our claim that SMO regulates expression of EWSR1::FLI1/ERG, and we are thankful to the reviewer for suggesting the experiment.

7. Similarly, for the CHIP analyses, the GLI consensus binding sequence is the same for both GLI1 and GLI2. Thus, presumably, GLI2 also binds to the promoter and intron 2 regions of EWS-FLI1 to activate its transcription. Demonstration that GLI2 binds to the promoter and intron 2 regions of EWS-FLI1 would further solidify the conclusions of this figure although not absolutely necessary.

As suggested by the reviewer, we performed an additional experiment to monitor binding of GLI2 to *EWSR1*. The results, presented in Fig. 3i, show that GLI2 binding was detected at the *EWSR1* promoter and intron 2 regions, similar to GLI1. Furthermore, we provide new results demonstrating that C1GALT1 knockdown or ITZ treatment abrogates binding of GLI1 and GLI2 to *EWSR1* (Fig. 4d and e, and Supplementary Fig. 8g and h).

8. The western blot of NIH-3T3 cells in Fig. 3d does not show any basal level of SMO with DMSO treatment. This is odd as NIH-3T3 cells respond to SHH and IHH ligands and thus, there must be some baseline levels of SMO to initiate the pathway. Furthermore, since SAG binds to and activates SMO, there must be some baseline level of SMO.

We have replaced the immunoblot (now Fig. 4f in the revised manuscript) with a darker exposure that shows, as expected, that NIH 3T3 cells have low levels of SMO, which are increased following treatment with SAG or ectopic expression of C1GALT1. In response to the reviewer's concern, we have also replaced the immunoblot of Supplementary Fig. 4c to show low levels of SMO in control (empty vector-expressing) NIH 3T3 cells.

9. Page 5 – The description of Hh signaling pathway needs amendments. GLI2, not GLI1, is the primary

effector of the pathway after SMO activation. GLI1 is a target gene of GLI2 and acts as an amplifier of the pathway. Thus, GLI2 knock out mice are embryonically lethal whereas GLI1 knockout mice show no developmental defects and as far as we know, is not required for development. Also, SUFU sequesters GLI1/2 from the nucleus but not in the primary cilia. Without activation of SMO, GLI1/2 are not found in the primary cilia, which is implied in the paragraph.

In the revised manuscript, we have modified the description of the Hh signaling pathway according to the reviewer's helpful suggestions.

Reviewer #3 (Remarks to the Author):

Ewing sarcoma (ES) is an aggressive pediatric bone cancer driven by EWS-FLI1, which is an attractive drug target. Using CRISPR/Cas9 screening, the authors identified C1GALT1 as a critical factor to promote EWS-FLI1 expression. Mechanistically, the authors find that C1GALT1 O-glycosylates and stabilizes SMO to enhance EWS-FLI1 expression and signaling, leading to ES growth. Itraconazole, a known C1GALT1 inhibitor, reduces EWS-FLI1 and suppresses ES tumor growth in mice. Although this finding is interesting, there are still several concerns.

Major concerns

1. In a panel of ES cell lines, the evidence showing that C1GALT1 knockdown or knockout decreases SMO protein levels was only demonstrated in A673 cells. Deng et al (Cell Death and Disease, 2020, 11:539) showed that itraconazole did not decrease SMO protein levels in SW480 and HCT116 colon cancer cell lines. In addition, Hu et al (Journal of Experimental & Clinical Cancer Research, 2017, 36:50) showed that itraconazole did not affect SMO levels in MKN45 and AGS gastric cancer cells. However, itraconazole still inhibits the hedgehog signaling pathway. These results suggest that itraconazole can exert its inhibitory effect without alteration of SMO protein levels.

In response to the reviewer's first comment, we now show that C1GALT1 knockdown reduces SMO protein levels in three additional ES cell lines, TC-71, TC-32 and TC-106 (Supplementary Fig. 8b).

Although the reviewer correctly notes that some reports have shown that itraconazole (ITZ) does not affect SMO levels, there are also reports showing ITZ does reduce the levels of SMO (Kim et al. 2010, *Cancer Cell* 17:388; Choi et al. 2017, *Sci Reports* 7:6552; Freitas et al. 2020, *Front Oncol* 10:563838; Dong et al. 2023, *Heliyon* 9:e19244). Thus, whether ITZ reduces SMO levels appears to be cell-type specific.

We agree with the reviewer that in some cell types, ITZ can exert its inhibitory effect on Hh signaling without reducing SMO protein levels. In this regard, numerous studies have reported non-canonical Hh pathway activation in cancer cells, in which GLI transcription factors are activated by a SMO-independent mechanism. For example, several oncogenic drivers have been shown to promote GLI1/GLI2 protein stability or transcriptional activity, including c-MYC in Burkitt lymphoma (Yoon et al. 2013, *DNA Repair* 34:9) and FOXC1 in breast cancer (Han et al. 2015, *Cell Rep* 13:1046). Notably, as mentioned in our manuscript, in Ewing sarcoma, the EWSR1::FLI1 fusion protein directly activates GLI1 transcription (Beauchamp et al. 2009, *J Biol Chem* 284:9074). In addition, several oncogenic signaling pathways have been shown to enhance GLI1/GLI2 function, including MEK/ERK signaling in melanoma (Stecca et al. 2007, *PNAS* 104:5895), pancreatic cancer (Ji et al. 2007, *J Biol Chem* 282:14048), gastric cancer (Seto et al. 2009, *Mol Carcinog* 48:703) and colon cancer (Mazumdar et al. 2011, *Oncotarget* 2:638), and PI3K/ATK/mTOR in pancreatic and ovarian cancers (Singh et al. 2017, *Oncotarget* 8:833) and non-small cell lung cancer (Mizuarai et al. 2009, *Mol Cancer* 8:44).

In the examples mentioned by the reviewer (Deng et al. 2020 and Hu et al. 2017), ITZ was shown to decrease GLI1 levels without a reduction in SMO. It is possible that in these cancer cell types, GLI1 is activated through a SMO-independent mechanism that is inhibited by ITZ. Indeed, ITZ has been shown to inhibit various signaling pathways, including AKT/mTOR (Liang et al. 2017, *Oncotarget* 8:28510; Tsubamoto et al. 2017, *Anticancer Res* 37:515; Ueda et al. 2017, *Anticancer Res* 37:3521) and ERK (Fan et al. 2022, *Cell Biochem Biophys* 80:331), that could drive SMO-independent activation of GLI in these cells.

2. The mechanism by which C1GALT1 O-glycosylates SMO to stabilize SMO protein remains elusive. The authors showed that FLAG-SMO (T55A) and (T500A) mutants cannot be stably expressed in A673 cells. In C1GALT1 deficient cells, proteins still have Gal-GalNAc (T antigen) or sialyl T expression on

all O-glycosites, which is quite different from the mutants with a complete loss of the O-glycan at certain O-glycosites. Moreover, O-glycosites on the proteins in different cell types may be different. To identify the correct O-glycosites on SMO in Ewing sarcoma, authors need to perform mass spectrometry using SMO from ES cell lines or ES tumors. Overall, although C1GALT1 can O-glycosylate SMO, whether C1GALT1-mediated glycosylation can stabilize SMO protein stability remains unclear.

Our laboratory currently lacks the expertise required to conduct detailed glycoproteomic studies for analyzing O-glycosylation. We acknowledge the importance of comprehensively identifying glycosylation sites that may regulate SMO stability in ES cells. Recognizing the significance of this investigation, we plan to collaborate with experts in the field of glycobiology to address this question in a separate follow-up study. Furthermore, we believe that beyond mere identification of glycosites, it is crucial to elucidate their functional roles. Mechanistic studies involving site-specific mutants are essential, albeit extensive, and we consider them beyond the scope of the current study. As described in more detail below, our initial findings with two glycosites on SMO, T55 and T500, identified from previous glycoproteomic studies, suggest their importance in regulating SMO stability.

Based on the reviewer's comments, we performed several additional experiments to further explore the mechanism by which C1GALT1-mediated O-glycosylation stabilizes SMO. In the revised manuscript we now show that the loss of SMO protein levels following C1GALT1 knockdown can be restored by addition of the proteasomal inhibitor MG132 or the lysosomal inhibitor bafilomycin A1 (Fig. 5e), suggesting that C1GALT1-mediated O-glycosylation stabilizes SMO from proteasomal and lysosomal degradation.

We agree with the reviewer that O-glycosites may be different in different cell types. The two SMO residues that we mutated (T55 and T500) had been previously shown in large-scale proteomics studies to be O-glycosites with a core 1 structure, indicating they are glycosylated by C1GALT1—a finding that we now emphasize in the revised manuscript. In the original manuscript, we showed that mutation of these residues resulted in reduced levels of SMO in A673 cells, which could be restored by addition of the proteasomal inhibitor MG132. In the revised manuscript, we present a new experiment showing that the levels of the SMO mutant proteins can also be restored by addition of the lysosomal inhibitor bafilomycin A1 (Fig. 5f), consistent with the results with endogenous SMO following C1GALT1 knockdown. Furthermore, we performed experiments to show that in A673 cells FLAG-SMO mutants (T55A and T500A) undergo polyubiquitination (Fig. 5g), suggesting that the reduced stability of the O-glycosylation-defective mutants is due, at least in part, to ubiquitin-mediated protein degradation.

3. In ES clinical samples, the quality for C1GALT1 immunohistochemistry is poor. Moreover, there is deficiency in the correlation between C1GALT1 expression and SMO.

Prompted by the reviewer's comment, we sought to obtain higher quality ES clinical samples for immunohistochemistry analysis. We purchased two high-quality, commercially-available bone and cartilage malignant tumor tissue arrays, which collectively included three distinct ES patient tumor samples. We note that because ES is a rare cancer, it is difficult to obtain patient samples. We performed immunohistochemistry for C1GALT1, SMO, GLI1 and FLI. The results with the new samples confirm our original finding that C1GALT1 is highly expressed in ES patient tumor samples and that high C1GALT1 expression correlates with high expression of SMO (Figs. 2c and 3a).

4. The effect of C1GALT1 overexpression and knockdown (or knockout) on tumor growth should be done using multiple ES cell lines. Moreover, the effect of itraconazole on tumor growth using C1GALT1 knockdown or knockout cells should be analyzed. This approach will provide insights into whether itraconazole predominantly exerts its effects by inhibiting C1GALT1 or through alternative pathways not explored in the current study.

In response to the reviewer's first comment, we knocked down C1GALT1 in two additional ES cell lines, TC-71 and TC-32, which require EWSR1::FLI1 for viability and proliferation in cell culture, and monitored the ability of the cells to form tumors in mice following subcutaneous injection. We found that C1GALT1 was required for viability of these cell lines (Supplementary Fig. 8a), precluding us from deriving stable C1GALT1 knockdown derivatives, and therefore these experiments were performed by transducing cells with concentrated, high titer C1GALT1 shRNA lentivirus followed by short-term drug selection (12-16 hours) and recovery in the absence of drug (see Methods section for details); under these conditions we observed efficient reduction of C1GALT1 levels (Supplementary Fig. 8b) without a marked decrease in cell viability (Supplementary Fig. 8c). As expected, C1GALT1 knockdown substantially reduced the ability of TC-71 and TC-32 cells to form tumors in mice (Fig. 7f and Supplementary Fig. 11f). Immunohistochemistry analysis of residual tumors from C1GALT1 knockdown TC-71 cells revealed significantly reduced levels of FLI1, SMO, and GLI1 (Fig. 7g). These results confirm that the observed reduction in EWSR1::FLI1, SMO, and GLI1 expression following C1GALT1 knockdown in cultured cells also occurs in vivo. Additionally, the tumors exhibited decreased Ki-67 staining, indicating reduced proliferation.

Furthermore, as requested, we analyzed the effect of ITZ on growth of tumors derived from C1GALT1 knockdown TC-71 and TC-32 cells. If ITZ exerts its effects primarily through C1GALT1, then the drug would not be expected to further exacerbate the reduction in tumor growth observed upon C1GALT1 knockdown. By contrast, if ITZ exerts its effects by inhibiting other pathways, then the drug may further reduce growth of tumors derived from C1GALT1 knockdown cells. We found that ITZ treatment did not significantly reduce the growth of tumors derived from C1GALT1 knockdown TC-71 (Fig. 7f) or TC-32 (Supplementary Fig. 11f) cells, supporting our model that ITZ reduces ES tumor growth by targeting C1GALT1.

Finally, to determine the effect of C1GALT1 overexpression, we performed an experiment in which we ectopically expressed C1GALT1 in TC-71 cells, treated the cells with increasing concentrations of ITZ, and monitored cell viability. Figure 7c shows that, similar to our results with ectopic expression of EWSR1::FLI1, ectopic expression of C1GALT1 rescued the ITZ-mediated decrease in ES cell viability.

Collectively, the results from these experiments strengthen our model that ITZ exerts its effects on ES cell viability and tumor growth by inhibiting C1GALT1. We thank the reviewer for suggesting these experiments.

5. C1GALT1 has been known to regulate several integrins and receptor tyrosine kinases. How can the authors rule out the possibility that C1GALT1 regulate ES tumor growth through the regulation of these proteins.

In response to the reviewer's comment, we have mentioned in the Discussion the possibility that C1GALT1 regulates ES tumor growth through targets other than SMO, including integrins and receptor tyrosine kinases.

Specific comments:

6. Figure 1. Molecular weight markers should be added to all western blots.

Molecular weight markers have been added to all immunoblots presented in the manuscript.

7. In Figure 3d, SMO, GLI, and ACTB in a mouse cell line should be labeled with Smo, Gli, and Actb, respectively. The relative changes in Smo and Gli1 using qRT-PCR analysis are not comparable to those observed on western blots.

The reviewer is not completely correct. Mouse nomenclature rules specify that protein names should be in all uppercase letters, and therefore we have retained the original labeling of the immunoblots in NIH

3T3 cells. However, mouse gene and RNA names have only the first letter capitalized, and so we have revised the labeling in graphs showing qRT-PCR results in NIH 3T3 cells accordingly.

With regard to the comment about the differences in mRNA and protein levels, in Fig. 4f (formerly Fig. 3d) we have provided new immunoblots for SMO and GLI1, which show low levels of SMO in control DMSO-treated and empty vector-expressing NIH 3T3 cells, more accurately mirroring the relative changes in the accompanying qRT-PCR experiment.

8. In Figure 3f, the molecular weight of SMO was decreased after O-glycanase treatment; however, PNA pull-down SMO did not show the change in molecular weight. In addition to PNA, the authors need to use VVA and jacalin to perform pull-down assays. Moreover, to validate the glycan changes, immunoprecipitation of SMO and then immunoblotting with PNA and VVA also should be done.

In the PNA lectin pull-down assay of Fig. 5a (formerly Fig. 3f), we would not expect to see a change in molecular weight because the assay specifically monitors the *level of O-glycosylated SMO* (i.e., the level of SMO harboring O-Gal-GalNAc glycans) in the cell lysate that is bound to the PNA lectin bead (see schematic). In the absence of O-glycosidase, SMO harbors O-Gal-GalNAc glycans and is therefore bound to PNA-agarose beads and detected in the pull-down. However, in the presence of O-glycosidase, which cleaves the Gal-GalNAc disaccharide, the number of O-Gal-GalNAc glycans is reduced, and therefore the amount of O-glycosylated SMO bound to PNA-agarose beads is reduced. By contrast, the immunoblot of total SMO in whole cell lysates (input) shows a change in molecular weight as a result of the reduced number of O-Gal-GalNAc glycans.

As requested by the reviewer, we performed an additional pull-down assay using jacalin (which, like PNA, also binds the T antigen but unlike PNA, binds to a mono- or disialylated form of this structure). The results, presented in Supplementary Fig. 5a, show that consistent with our results using PNA, SMO also binds to jacalin and that binding is substantially reduced following O-glycosidase treatment.

Moreover, as requested, for the PNA and jacalin binding assays, we performed the reciprocal experiment in which SMO was immunoprecipitated and the immunoprecipitate was analyzed for the presence of T antigen by immunoblotting with biotinylated PNA or jacalin. The results confirmed that SMO bound to PNA (Fig. 5a) or jacalin (Supplementary Fig. 5a), and that binding was reduced following O-glycosidase treatment, thus validating the glycan changes.

With regard to performing the lectin pull-down assay using VVA (which binds to Tn antigen), such experiments are typically performed following knockdown of C1GALT1, which results in increased binding of the protein of interest to VVA (see, for example, the identification of the C1GALT1 targets EGFR in head and neck cancer [Lin et al. 2018; *Oncogene* 37:5780]; FGFR2 in colon cancer [Hung et al. 2014, *Oncotarget* 5:2096]; and integrin α v in pancreatic cancer [Kuo et al. *Oncogene* 40:1242]). It is important to note that in these studies the total level of the protein of interest in whole cell lysates is unaffected by knockdown of C1GALT1. By contrast, as noted in our manuscript, we find that SMO levels are undetectable in C1GALT1 knockdown cells (see Fig. 4a).

Nevertheless, in an attempt to address the reviewer's concern, we performed a VVA pull-down assay following knockdown of C1GALT1 in A673 cells. Rather than performing these experiments in stable C1GALT1 knockdown A673 cells, we performed them by transducing cells with concentrated, high titer C1GALT1 shRNA lentivirus in the absence of drug selection, enabling us to monitor an early timepoint following C1GALT1 knockdown. We hoped that under these conditions, we would observe efficient reduction in C1GALT1 while retaining SMO at or near normal levels, such that an increase in VVA binding could be observed. Unfortunately, we found that the reduction in SMO levels in whole cell lysates following C1GALT1 knockdown was comparable to the reduction in SMO signal observed in the VVA pull-down assay, and thus we are unable to conclude whether knockdown of C1GALT1 has an effect on SMO O-glycosylation. Because the results are uninterpretable with respect to the role of C1GALT1, we have not included them in the revised manuscript, but present them in below in Fig. B.

Figure B. VVA lectin pull-down assay in C1GALT1 knockdown ES cells. Representative immunoblot showing the level of O-GalNAc-containing SMO, detected by either immunoblot (IB) analysis for SMO in a VVA lectin pull-down (PD) assay or IB analysis for PNA or VVA in a SMO immunoprecipitate (IP), in A673 cell lysates following treatment with neuraminidase (to remove sialic acids that can hinder binding to VVA beads). The level of total SMO in whole cell lysate (input) is also shown.

9. In Figure 3j, SMO degradation could be dependent on both proteasome and lysosome, the authors also need to add a lysosome inhibitor and assess the mutant SMO levels. In addition, the mRNA levels of mutant SMO need to be analyzed using qRT-PCR. Ubiquitination of SMO should be checked. To monitor and validate the degradation of SMO mutants, it is recommended to perform immunofluorescence microscopy specifically for FLAG-SMO (T55A) and FLAG-SMO (T5800A) in cells treated with or without proteasome and lysosome inhibitors. This additional experiment will provide valuable insights into the subcellular localization and stability of the SMO mutants under different conditions, helping to elucidate the mechanisms involved in their degradation.

In response to the reviewer's comments, we performed several additional experiments.

First, we repeated the experiment originally shown in Fig. 3j, and monitored the levels of SMO(T55A) and SMO(T500A) mutants in the presence and absence of the lysosome inhibitor bafilomycin A1. The results, now presented in Fig. 5f, show that bafilomycin A1 treatment resulted in increased levels of the mutant proteins, comparable to that obtained with MG132 treatment.

Second, we monitored the mRNA levels of wild-type SMO, SMO(T55A) and SMO(T500A) in the presence of MG132 or bafilomycin A1. In these experiments, we performed qRT-PCR using a primer pair that specifically detected the ectopically-expressed FLAG epitope-tagged version of SMO. The results, presented in Supplementary Fig. 5d, show that the steady-state mRNA levels of the SMO mutants were unaffected by treatment with MG132 or bafilomycin A1.

Third, we performed a ubiquitination assay to monitor ubiquitination of the SMO mutants. In brief, A673 cells stably expressing FLAG-SMO, FLAG-SMO(T55A) or FLAG-SMO(T500A) were transfected with a plasmid expressing HA-tagged ubiquitin (HA-Ub) and treated in the presence or absence of MG132 or bafilomycin A1. Cell lysates were incubated with anti-HA magnetic beads, and pulled down proteins were detected using an anti-FLAG M2 antibody; the reciprocal experiment was also performed in which lysates were incubated with anti-FLAG magnetic beads, and the pulled down proteins were probed using an anti-HA antibody. The results, presented in Fig. 5g, show that

polyubiquitination of the SMO mutants could be detected. These results confirm that the reduced stability of the mutants was due, at least in part, to ubiquitin-mediated protein degradation.

Finally, to validate these results, we performed immunofluorescence in A673 cells expressing FLAG-SMO, FLAG-SMO(T55A) or FLAG-SMO(T500A) and treated in the presence of MG132 or bafilomycin A1. The results of Supplementary Fig. 5c show that FLAG-SMO(T55A) and FLAG-SMO(T500A) were expressed at low or undetectable in A673 cells, and that their expression was restored by treatment with either MG132 or bafilomycin A.

We thank the reviewer for suggesting these experiments, the results of which have allowed us to gain additional insight into the mechanisms involved in ITZ-mediated SMO degradation.

10. The experiments presented in Figure 3 are crucial for gaining insights into the underlying mechanism. To enhance the robustness and generalizability of the findings, it is strongly recommended to conduct experiments in panels 3f and 3j using multiple ES cell lines. Additionally, considering the space constraints in the main figures, the data from these experiments involving multiple cell lines can be appropriately presented in the extended data section.

In response to the reviewer's recommendation, we performed two key experiments from Fig. 3 (Fig. 5 in the revised manuscript) in an additional ES cell line, TC-71. First, we monitored SMO O-glycosylation in TC-71 cells following ITZ treatment. Consistent with our findings in A673 cells, we found that in TC-71 cells, SMO bound to PNA lectin beads and that binding was reduced following treatment with ITZ (Supplementary Fig. 9b), indicating that SMO is O-glycosylated in TC-71 cells and that O-glycosylation is reduced upon inhibition of C1GALT1. Second, we expressed SMO(T55A) and SMO(T500A) mutants in TC-71 cells and found, consistent with the results in A673 cells, that SMO(T55A) and SMO(T500A) mutants were expressed at low or undetectable levels, which could be restored by addition of MG132 or bafilomycin A1 (Supplementary Fig. 9c).

11. In Figure 4a, the C1GALT1 staining should be localized in the Golgi rather than being observed throughout the entire cell. Additionally, it will be of importance to stain SMO, EWSFLI, and GLI1, and then analyze the correlation of C1GALT1 expression with the expression of these proteins in ES clinical samples.

Using the new, high-quality ES patient tumor samples, we performed IHC for C1GALT1, SMO, GLI1 and FLI. The results, shown in Fig. 2c and 3a, reveal that high C1GALT1 expression correlates with high levels of SMO, GLI1 and FLI1 in ES clinical samples.

With regard to the reviewer's comment about the sub-cellular localization of C1GALT1, we attempted to discern C1GALT1 subcellular localization in ES tumor sections using confocal microscopy after immunofluorescence staining with an anti-C1GALT1 antibody and a Golgi marker. Unfortunately, due to high background signal from the tissue sections, we could not obtain clear images showing C1GALT1 localization in the Golgi of tumor cells. However, we performed a similar experiment in A673 ES cells and demonstrated that C1GALT1 is localized to the Golgi (see Fig. C below).

Figure C. Immunofluorescence staining of C1GALT1 and a Golgi marker (GM130) in A673 cells.

12. In Figure 4f, it is necessary to transfect FLAG-SMO (T55A) and FLAG-SMO (T500A) into NIH3T3 cells and analyze their effect on tumor growth. Additionally, to establish a more causal relationship, it is recommended to knock down the endogenous Smo in C1GALT1 overexpressing NIH3T3 cells and then perform tumor xenograft assays using these cell lines.

We transfected NIH 3T3 cells with a plasmid expressing FLAG-SMO, FLAG-SMO(T55A) or FLAG-SMO(T500A) and found that the mutant proteins were expressed at moderate levels relative to wild-type FLAG-SMO (Supplementary Fig. 6d). Accordingly, GLI1 protein levels were reduced in NIH 3T3 cells expressing FLAG-SMO(T55A) or FLAG-SMO(T500A) relative to that obtained following ectopic expression of wild-type FLAG-SMO, indicating the mutant SMO proteins had reduced ability to induce Hh signaling (Supplementary Fig. 6d). Consistent with this result, the ability of FLAG-SMO(T55A) and FLAG-SMO(T500A) mutants to promote tumor growth of NIH 3T3 cells in mice was also reduced compared to wild-type FLAG-SMO (Supplementary Fig. 6e).

In addition, we performed the second experiment requested by the reviewer and knocked down SMO in C1GALT1 overexpressing NIH 3T3 cells, and monitored the ability of the cells to form tumors in mice. The results of Fig. 6f show that SMO knockdown substantially reduced the ability of C1GALT1-overexpressing NIH 3T3 cells to form tumors. Immunohistochemistry analysis confirmed reduced proliferation (as assessed by reduced Ki67 staining) and reduced Hh signaling (as assessed by reduced GLI1 staining) in tumors derived from SMO knockdown, C1GALT1-overexpressing NIH 3T3 cells. Thus, the ability of ectopic C1GALT1 expression to transform NIH 3T3 cells is SMO dependent.

13. Figure 5. In addition to A673, it is necessary to analyze the effect of itraconazole on C1GALT1 levels and SMO mRNA and protein levels in all EC cell lines used in this study, including SK-N-MC, TC-71, TC-106, and TC-32.

As requested, we performed additional experiments to analyze the ability of ITZ to reduce C1GALT1 protein levels, and SMO mRNA and protein levels in TC-32, TC-71 and TC-106 cells. Consistent with the results we obtained in A673 cells, we found that ITZ treatment reduced C1GALT1 and SMO protein levels in other ES cell lines in a dose-dependent manner (Supplementary Fig. 8f), and also reduced SMO mRNA levels (Supplementary Fig. 8e). We note that SK-N-MC cells were not analyzed in the manuscript other than the immunoblot of Supplementary Fig. 3c monitoring C1GALT1 and SMO levels in ES cell lines.

14. In Figure 5e, C1GALT1 knockdown TC-71 cells should be included in the animal model. Moreover, the effect of itraconazole treatment on the tumor growth of C1GALT1 knockdown TC-71 cells need to be assessed. The tumors should be harvested to analyze C1GALT1, SMO, and FLI1 expression using immunohistochemistry.

We performed the requested experiment in which we monitored the ability of C1GALT1 knockdown TC-71 or TC-32 cells to form tumors in mice in the presence or absence of ITZ treatment. As expected, C1GALT1 knockdown substantially reduced the ability of TC-71 cells (Fig. 7f) and TC-32 cells (Supplementary Fig. 11f) to form tumors in mice. Furthermore, the ability of ITZ to suppress growth of TC-71 or TC-32 xenografts was not further significantly exacerbated by C1GALT1 knockdown (Fig. 7f and Supplementary Fig. 11f), indicating that ITZ exerts its effects on ES tumor growth by inhibiting C1GALT1. Immunohistochemistry analysis of these tumors confirmed the reduction of C1GALT1, SMO, GLI1 and FLI1 following treatment of mice with ITZ or upon knockdown of C1GALT1 (Fig. 7g).

15. Figure 6. The sugars are expected to be located in the extracellular region of SMO. However, it is crucial to note that the authors currently only possess evidence indicating the presence of disaccharides Gal-GalNAc, as determined by PNA lectin binding, instead of pentasaccharides.

We thank the reviewer for alerting us to the errors in the model (shown in Fig. 6 in the original manuscript and Fig. 8 in the revised manuscript), and have revised the model accordingly to show that the O-glycan (now shown as a Gal-GalNAc disaccharide) is located in the extracellular region of SMO. In addition, we have revised the manuscript to state that the results of the PNA lectin pull-down assay show that SMO harbors the Gal-GalNAc disaccharide, and have revised the schematic of the PNA lectin pull-down assay (Fig. 5a) to show the disaccharide.

16. In Extended Data Figure 3c, the label C1GALTC1 should be corrected to C1GALT1C1.

We have corrected this labeling error in the revised figure, which is now shown as Supplementary Fig. 3a.

17. In Extended Data Figure 5a, SMO expression should also be analyzed in all cell lines.

As requested, we have provided an immunoblot showing SMO levels in the ES cell lines (now shown in Supplementary Fig. 3c). The results show, as expected, that ES cell lines expressing high levels of C1GALT1 also have elevated expression of SMO.

18. In Extended Data Figure 5b, C1GALT1 staining is not localized in the Golgi and the quality of tumor samples is poor. Nothing can be concluded from these images.

In the original manuscript, Extended Data Fig. 5b showed C1GALT1 staining in osteosarcoma patient samples. As described above (see Reviewer 1, comment #1), in the original manuscript, we used osteosarcoma patient tumor samples as a negative control but in the revised manuscript we have removed the osteosarcoma samples and now show staining in normal bone and cartilage samples (see Fig. 2c), which serve as a more appropriate control for these experiments.

19. Extended Data Figure 7. The animal model used in this study lacks the specific manipulations of C1GALT1, such as knockdown or knockout. Incorporating C1GALT1 knockdown or knockout in the animal model is essential for a more targeted investigation into the role of C1GALT1 in the observed outcomes. This experimental modification will provide a clearer understanding of the direct contribution of C1GALT1 to the studied processes in the context of the animal model.

In response to the reviewer's comment, we knocked down C1GALT1 in the ES cell lines TC-71 and TC-32 (which were shown in Extended Data Fig. 7 of the original manuscript), and monitored the ability of the cells to form tumors in mice following subcutaneous injection. We found that C1GALT1 was required for viability of these cell lines (Supplementary Fig. 8a), precluding us from deriving stable C1GALT1 knockdown derivatives, and therefore these experiments were performed by transducing cells with concentrated, high titer C1GALT1 shRNA lentivirus followed by short-term drug selection (12-16 hours) and recovery in the absence of drug (see Methods section for details); under these conditions, we observed efficient reduction of C1GALT1 levels (Supplementary Fig. 8b) without a marked decrease in cell viability (Supplementary Fig. 8c). As expected, C1GALT1 knockdown substantially reduced the ability of TC-71 (Fig. 7f) and TC-32 (Supplementary Fig. 11f) cells to form tumors in mice.

20. Extended Data Figure 8. All representative western blots should be shown. This result is recommended to be described in the Result section instead of Discussion section.

As requested, we have provided immunoblots for the experiments shown in this figure, now Supplementary Fig. 12b. The results show that knockdown of C1GALT1 reduces SMO and GLI1 protein levels in ovarian and pancreatic cell lines, consistent with the qRT-PCR results. In addition, as requested, we now describe the experiments in the Results section rather than the Discussion section.

21. In Discussion section, the authors should consider and discuss the possibility that C1GALT1 may

exert its functional effects through other protein substrates such as integrins and receptor tyrosine kinases (RTKs), which are well known C1GALT1 substrates.

As suggested, in the revised Discussion section we mention the possibility that C1GALT1 may exert its effects through other C1GALT1 substrates, such as integrins and receptor tyrosine kinases.

22. Other minor concerns:

Line 104: References should include the original papers instead of one review article.

We have now provided references for the original papers in the sentence discussing C1GALT1 overexpression in various cancers.

Line 230 and 232: six instead of seven ES samples.

In the revised manuscript, this typo has been corrected to accurately state the number of ES patient samples analyzed.

Line 330: For pancreatic cancer, *Oncogene*. 2021 Feb;40:1242-1254 should be cited.

This sentence has been deleted from the revised manuscript, but the Kuo et al. 2021 (*Oncogene* 40: 1242) reference is cited elsewhere in the manuscript to support the idea that C1GALT1 is highly expressed in pancreatic cancer and that it promotes cell invasiveness in pancreatic cancer by targeting integrin αv .

REVIEWER COMMENTS

Reviewer #1 (Remarks to the Author):

The authors have significantly improved the manuscript and the paper became much stronger. There are only two remaining minor concerns of my previous concerns that should be addressed:

1. While I agree that bone and cartilage may be a better control than osteosarcoma (Fig. 2c), the new image on IHC for C1GALT1 in bone does not show bone like in the corresponding H&E image, but univacuolar fat tissue. The fat tissue should be replaced by bone.

We believe the IHC image did not show univacuolar fat tissue, but rather bone marrow adipose tissue, a type of fat deposit found in the cavities of bone trabeculae. In response to the reviewer's comment, we obtained new normal bone samples and repeated the C1GALT1 immunohistochemistry and H&E staining. The new results confirm that C1GALT1 is undetectable in normal bone.

2. I commend the authors for providing better statistics and more replicates for important experiments. Yet, I disagree that inappropriate t-tests should be employed in n=3 experiments even though this practice has been done in other papers published in the same journal or other high-impact journals. In short: inappropriate statistics cannot be justified by the mistakes of others. Science is not helped by perpetuating inappropriate statistics. In my opinion, statistics should be either appropriate or not provided. The authors could opt in for the latter option as in case of strong and consistently observed differences between groups, analytical statistics is not necessary.

We thank the reviewer for their candid comments. In response, we carefully evaluated each figure panel where we had used a t-test and revised it appropriately:

- For Fig. 3i and Supplementary Figs. 8g and 8h, we removed the statistical analysis because there are additional, independent data to support the conclusion (see Figs. 4d and 4e).
- For Figs. 4b and 4e, we included additional replicates and performed a two-tailed Mann-Whitney test to analyze the differences between the two groups.
- For Fig. 4d, we included the results for a second, independent C1GALT1 shRNA, which allowed us to perform statistical analysis using one-way ANOVA with post-hoc Dunnett's multiple comparisons test.
- For Fig. 4f and Supplementary Fig. 8a, we removed the statistical analysis as the effect is strong and the differences between groups were consistent among replicates.
- For Supplementary Fig. 2a, we reanalyzed the data using one-way ANOVA with post-hoc Dunnett's multiple comparisons test. Furthermore, we removed the statistical analysis for *EWSRI* because there are additional, independent data to support the conclusion (see Figs. 1c and 1d).

Reviewer #2 (Remarks to the Author):

The authors have updated the manuscript with significant new data to address the concerns of previous comments by me and the other reviewers. The manuscript is much improved. However, I still have some major concerns and a minor one that need to be addressed.

1. The presented data, particularly those of SMO shRNA data from Fig. 6 and Supp Fig 7 and SMO glycosylation mutations in Supp Fig 6, suggest that Ewing Sarcoma cells depend on continuous SMO stabilization by C1GALT1 for growth and proliferation, presumably through transcription of FLI1 fusion proteins by activated GLI1/2 since GLI1/2 binds to promoter regions of EWS-FLI1 (Fig 3, 4). As noted in the model Fig 8a and also previous literature, EWS-FLI1 then induces further transcription of GLI1. Given this model, a circular loop of upregulation has been induced between EWS-FLI1 and GLI1 such that this loop should be independent of SMO. Given the model, how can EWS-FLI1 cells still depend on SMO and C1GALT1 esp. if the primary target of C1GALT1 is SMO in ES cells? The reciprocal activation loop between EWS-FLI1 and GLI1 must stop in order for the ES cells to continue to depend on C1GALT1 and SMO. Or perhaps EWS-FLI1 does not induce further GLI1 expression in these cells and thus require GLI1/2 activation by SMO to maintain ESW-FLI1 expression. Data are needed to reconcile this seeming contradiction as currently presented.

With regard to the reviewer's comment "perhaps EWS-FLI1 does not induce further GLI1 expression in these cells," we note that a previous study showed that shRNA-mediated knockdown of EWSR1::FLI1 reduces GLI1 protein levels in A673 cells (Beauchamp et al. 2009, *J Biol Chem* 284:9074), the same cell type used in many of our experiments, indicating that EWSR1::FLI1 does indeed induce GLI1 expression.

However, even though EWSR1::FLI1 directly promotes expression of GLI1, SMO is still continuously required to alleviate repression of GLI1/2 by SUFU, a negative regulator of Hh signaling that acts by sequestering GLI proteins from the nucleus. In response to the reviewer's comment, we have revised the model of Fig. 8 to more clearly show that entry of GLI1/2 into the nucleus is regulated by SMO, such that even the regulatory loop between EWSR1::FLI1 and GLI1/2 is dependent on the presence of SMO.

2. The authors posit that C1GALT1 glycosylation of SMO may be the reason for the insensitivity of ES cells to SMO inhibition by standard SMO inhibitors, cyclopamine, vismodegib and sonidegib, that share the same binding SMO binding pocket whereas ITZ is potent due to its ability to inhibit C1GALT1. Thus, ITZ may be more potent than SMO inhibitors in ES cells than in NIH-3T3 cells where the SMO inhibitors are more potent than ITZ. This hypothesis should be easily testable with the reagents that the authors have already presented. Overexpression of C1GALT1 in NIH-3T3 cells as in Fig. 6E should result in increased glycosylation and protein stabilization of SMO and thus GLI1. The induction of GLI1 protein expression and GLI1 mRNA transcription in the context of C1GALT1 overexpression should be relatively resistant to cyclopamine, vismodegib, sonidegib and perhaps other drugs that bind in the same SMO pocket but sensitive to ITZ. This is important as it would establish upregulation and glycosylation of SMO as a potential new resistance mechanism to the FDA-approved vismodegib/sonidegib and possibly glasdegib as well.

As suggested by the reviewer, we ectopically expressed C1GALT1 (or, as a control, empty vector) in NIH 3T3 cells, treated them with cyclopamine, vismodegib, sonidegib or ITZ, and monitored GLI1 mRNA and protein levels. We first confirmed that ectopic expression of C1GALT1 increased SMO O-glycosylation, as expected (Supplementary Fig. 11a), consistent with the increase in SMO levels we observe (Fig. 4f). The results, presented in Supplementary Fig. 11b-e, show that ectopic expression

of C1GALT1 abrogated the decrease in GLI1 observed upon treatment of NIH 3T3 cells with cyclopamine, vismodegib or sonidegib, strongly supporting the notion that the C1GALT1-mediated increase in SMO O-glycosylation renders cells resistant to the conventional SMO inhibitors. We observed that ectopic expression of C1GALT1 also abrogated the decrease in GLI1 levels observed upon ITZ treatment, consistent with the results presented in Fig. 7c showing that the ITZ-mediated decrease in ES cell viability is rescued by ectopic C1GALT1 expression. We thank the reviewer for suggesting this experiment, which, as the reviewer stated, establishes O-glycosylation-dependent stabilization of SMO as a potential new resistance mechanism to FDA-approved conventional SMO inhibitors. We have mentioned this point in the Discussion of the revised manuscript.

3. In response to my previous comment, the authors note that PNA lectin study with C1GALT1 knock down was difficult due to resultant SMO instability and therefore, used ITZ as a C1GALT1 inhibitor to demonstrate loss of SMO glycosylation. However, as noted previously, ITZ has numerous other effects including glycosylation of other proteins and thus, these new results, while suggestive, are still less definitive. If SMO stability is an issue, then treating cells with proteasome and lysosome inhibitors with C1GALT1 shRNA (as was done in Fig. 5e, f) should solve the SMO instability issue and allow for PNA lectin pull down.

We performed the experiment requested by the reviewer. In brief, A673 cells stably expressing a NS or C1GALT1 shRNA were treated with MG132 or bafilomycin A1 and 8 hours later cell lysates were subjected to a PNA lectin pull-down assay. The results, presented in Fig. 5f, show that SMO O-glycosylation was substantially reduced following shRNA-mediated knockdown of C1GALT1. We thank the reviewer for suggesting the experiment, the results of which have allowed us to definitely conclude that loss of C1GALT1 reduces SMO O-glycosylation.

4. In line 360, the authors note that Hh pathway activation drives other cancers such as ovarian, pancreatic, and lung cancers. However, this definitive statement is incorrect. Canonical Hh signaling pathway in tumors of epithelia derived from the endoderm (e.g. lung, pancreas, colon, prostate, bladder) have been shown to be tumor suppressive through Hh pathway activation in stroma and Hh pathway inhibition, genetically and pharmacologically, accelerated tumor growth. These preclinical results are consistent with the negative human clinical trials of SMO inhibitors in these tumor types. Thus, antagonizing the Hh pathway (at least with therapeutic SMO inhibitors) for these tumor types has been abandoned. Whether canonical Hh pathway activation in these tumors does indeed drive tumor growth is controversial in light of these data demonstrating tumor suppression by paracrine Hh pathway. Perhaps, as the authors note, subsets of these cancers with high C1GALT1 may be amenable to SMO inhibition by C1GALT1 inhibitors esp. if these inhibitors do not lead to stromal SMO antagonism. However, this would not explain the failure of SMO inhibitors in clinical trials of colon, pancreas and other solid tumors without Hh pathway mutations unless the vast majority of these tumors had overexpression of C1GALT1 or other reasons to cause SMO to be resistant to vismodegib and sonidegib.

In the previous version of the manuscript, line 360 stated “Aberrant activation of Hh signaling is involved in many cancers...”, a phrase we used (also in line 152) to avoid saying that Hh pathway activation drives these cancers. In response to the reviewer’s comment, we have revised the wording to state that “Aberrant activation of Hh signaling is associated with many cancers.” Furthermore, prompted by the reviewer’s comment, we have removed from the Discussion the text stating that our results may provide an explanation for why conventional SMO inhibitors have had disappointing results in clinical trials for solid tumors.

Based on the reviewer’s comment, we have also removed the results of Supplementary Fig. 12, which showed that knockdown of C1GALT1 reduced SMO and GLI1 mRNA and protein levels

in human ovarian and pancreatic cancer cell lines. Our intention in presenting these data was solely to demonstrate the generality of our findings that C1GALT1 inhibition reduces SMO and GLI1 levels, thereby suppressing Hh signaling. We did not intend to suggest that inhibition of SMO and Hh signaling would yield therapeutic benefits in these contexts. We agree with the reviewer that the role of the Hh pathway in cancers such as ovarian and pancreatic is complex and highly context-dependent; the mixed outcomes of SMO inhibitors in clinical trials underscore this complexity. Further mechanistic studies, which are beyond the scope of our study, are needed to determine whether C1GALT1-mediated SMO inhibition could indeed offer therapeutic advantages in these cancers.

5. Minor: Line 147-148: Activation of SMO indeed induces GLI1/2 activation. However, SMO activation suppresses GLI3R, not activate it as is implied in the current text.

In the revised manuscript, for simplicity we have removed mention of GLI3, which is not relevant to our study.

Reviewer #3 (Remarks to the Author):

The revised manuscript has been significantly improved. However, there are still some concerns that should be resolved.

1. The immunohistochemistry (IHC) of C1GALT1 in the whole manuscript did not show specific staining signals. This is not acceptable. These IHC data include Fig. 2c, 6c, 6g, 7e, 7g, Supplementary Fig. 6f, Supplementary Fig. 11c and 11d. In addition, the IHC of FLAG in the Supplementary Fig. 6f indicates nuclear staining patterns as hematoxylin counterstain, which is not the expected localization of FLAG-SMO, FLAG-SMO(T55A), and FLAG-SMO(T500A).

As requested, we have repeated all of the C1GALT1 IHC experiments presented in the manuscript. In trouble-shooting these experiments, we tested different dilutions of the C1GALT1 antibody and selected the one that optimized a specific signal. We believe that the new C1GALT1 IHC images presented in the revised manuscript are greatly improved. We note that the new results are consistent with the original conclusions presented in the manuscript.

Regarding the reviewer's concern about the FLAG IHC in Supplementary Fig. 6f, we again tested different dilutions of the FLAG antibody and selected the one that optimized a specific signal. The new images show that the FLAG (SMO) signal is localized to the cytoplasm.

2. Fig. 6f: It is necessary to show C1GALT1, SMO, and GLI1 levels in the stable lines of NIH3T3 cells in the same western blot because the IHC data are not convincing. There are four groups in Fig. 6b and 6f. However, only three groups of IHC data were shown in Fig. 6c and 6g.

As requested, we now provide immunoblots showing C1GALT1, SMO and GLI1 levels in the stable NIH 3T3 cell lines shown in Fig. 6f. The results, presented in Supplementary Fig. 6d, show, as expected, that in NIH 3T3 cells ectopically expressing C1GALT1, the levels of C1GALT1, SMO and GLI1 are elevated, consistent with the results of Fig. 4f. Furthermore, in cells expressing C1GALT1 and a SMO shRNA, the levels of SMO and GLI1 are reduced. Finally, in NIH 3T3 cells expressing SMO, the levels of SMO and GLI1 are increased (but not C1GALT1). The immunoblot results are consistent with the IHC data shown in Fig. 6g. As mentioned above, we have also repeated the C1GALT1 IHC experiments in Fig. 6g, and the results have been improved.

With regard to the reviewer's comments about the discrepancy between the number of groups in Fig. 6b versus Fig. 6c, and in Fig. 6f versus Fig. 6g, we now provide the IHC results for tumors derived from EWSR1 knockdown cells in Fig. 6c. We note that in Fig. 6f, NIH 3T3 cells expressing empty vector do not form tumors and therefore it is not possible to present IHC results for that group.

3. In several figures, such as Fig. 4a, 4b, 4f, and Supplementary Fig. 8e, C1GALT1 or ITZ can significantly regulate SMO mRNA levels. From these results, it appears that most of SMO expression is determined by its transcriptional regulation. However, in the Fig. 5e (left panel), either a proteasome inhibitor or lysosome inhibitor can almost completely reverse the SMO expression to the normal level. Do the authors have explanations for this phenomenon? Which is the major mechanism by which C1GALT1 regulates SMO levels?

Unlike *PTCH1* and *GLI1*, *SMO* is not widely considered to be a transcriptional target of Hh signaling. However, previous studies have suggested the possibility that decreased Hh signaling may lead to reduced *SMO* mRNA levels. For example, siRNA-mediated knockdown of *GLI1* has been shown to reduce *SMO* mRNA levels in a neuroblastoma cell line (Diao et al., 2014, *BMC Cancer* 14:600). Furthermore, treatment of lung adenocarcinoma cells with the SMO inhibitor vismodegib has been shown to reduce expression of *SMO* at the mRNA level (Fan et al., 2019, *Cell Death Dis* 10:626). In our study, we found that C1GALT1 knockdown or ITZ treatment had a more robust effect on SMO

protein levels than *SMO* mRNA levels. We therefore conclude that the major mechanism by which C1GALT1 regulates SMO levels is through O-glycosylation and stabilization of SMO protein.

4. Fig. 8: In the left panel, the multiple transmembrane domains of SMO should be located on the Golgi membrane instead inside the Golgi.

We have revised the schematic of Fig. 8 to show SMO on the Golgi membrane rather than inside the Golgi.

Response to the Reviewers Comments:

Reviewer #1 (Remarks to the Author)

The authors have adequately addressed all my concerns.

We sincerely thank the reviewer for reviewing our manuscript and providing valuable suggestions. We are pleased to know that we have satisfactorily addressed all the concerns.

Reviewer #2 (Remarks to the Author)

The authors have responded well to my previous comments. I have one further suggestion that should be added to the Discussion section as noted below. Otherwise, the manuscript is good to be published.

Comment:

In response to my concern that the previous model did not explain how SMO can affect the positive loop between GLI1 and EWS-FLI1, the authors noted in their rebuttal that SUFU must be inactivated by SMO in order for the GLI transcription factors to be translocated to the nucleus. However, without experimental evidence in EWS-FLI1 cells, this explanation is unsatisfactory. GLI1/GLI1 amplifications are a known mechanisms for sonidegib and vismodegib resistance (Buonamici et al., *Sci Transl Med*, 2010; Sharpe et al., *Cancer Cell*, 2015; Atwood et al., *Cancer Cell*, 2015). aPKC- ι/λ is another vismodegib resistance mechanism through phosphorylation and activation of GLI1 (Atwood et al., *Nature*, 2013). In both of these scenarios, according to the reasoning presented in the rebuttal, overexpression of GLI1/2 and, in particular, activation of physiological levels of GLI1 by aPKC- ι/λ should not be able to overcome SMO inhibitors since SMO is inactivated by the SMO antagonists and SUFU should still active to sequester GLI1/2 from the nucleus.

At this point, I believe it would be more accurate to note in the Discussion that the reason for the dependency of EWS-FLI1 cells on SMO activation, despite the positive feedback loop between GLI1 and EWS-FLI1, is not clear and may be due to the inactivation of SUFU to allow translocation of the GLI transcription factors to the nucleus given the previously noted examples of resistance mechanisms of SMO antagonists.

We are delighted that the reviewer considers our manuscript suitable for publication. We thank the reviewer for their final suggestion and, in response, we have noted in the Discussion that the reason for the dependency of ES cells on continuous SMO activation is not clear but may be due to the requirement for SMO to inactivate SUFU to allow translocation of GLI transcription factors to the nucleus. We note that there is indeed experimental evidence for a role for SUFU in ES cells, as previous studies have shown that ectopic expression of SUFU in ES cell lines reduces anchorage-independent growth (Joo et al. 2009, *PLoS One* 4:e7608), suggesting SUFU restrains GLI activity in ES cells, consistent with its demonstrated role in other cell types.

Reviewer #3 (Remarks to the Author)

In the revised figures, including Figures 2c, 6c, 6g, 7e, and 7g, as well as Supplementary Figures 6g, 12c, and 12d, the IHC staining for C1GALT1 did not display a specific Golgi staining pattern. Additionally, the background signals vary considerably within the same image data set.

Regarding the reviewer's comment on the lack of a specific Golgi staining pattern, we would like to clarify that these IHC images were acquired at 20x magnification, which limits the ability to clearly resolve the localization of cellular organelles, such as the Golgi, within tissues. We emphasize that the primary goal of these IHC experiments was to demonstrate C1GALT1 expression in ES human

tumors and monitor C1GALT1 levels in mouse xenograft tumors under various experimental conditions. These observations are further supported by additional data presented in the manuscript. Moreover, in response to a similar concern raised by the reviewer during the first revision, we provided confocal microscopy data from the ES cell line A673, which conclusively demonstrated C1GALT1 localization in the Golgi.

Regarding the reviewer's comment on the variability of the background signal within the same image dataset, we acknowledge that differences in background signal may occur across multiple tumor tissue sections, potentially due to the processing of each tumor into FFPE sections. However, the antibody dilutions and staining conditions were kept consistent throughout. As requested by the editorial staff, we have provided the results of our experiments to optimize the C1GALT1 antibody dilution in IHC experiments for a representative human ES tumor (Supplementary Fig. 3c) and mouse xenograft tumor derived from A673 cells expressing a NS or C1GALT1 shRNA (Supplementary Fig. 6b), the latter of which confirms the specificity of C1GALT1 staining.